# Provable Benefit of Random Permutations over Uniform Sampling in Stochastic Coordinate Descent

Donghwa Kim [1]   Jaewook Lee [1]   Chulhee Yun [1]

## Abstract

We analyze the convergence rates of two popular variants of coordinate descent (CD): random CD (RCD), in which the coordinates are sampled uniformly at random, and random-permutation CD (RPCD), in which random permutations are used to select the update indices. Despite abundant empirical evidence that RPCD outperforms RCD in various tasks, the theoretical gap between the two algorithms' performance has remained elusive. Even for the benign case of positive-definite quadratic functions with permutation-invariant Hessians, previous efforts have failed to demonstrate a provable performance gap between RCD and RPCD. To this end, we present novel results showing that, for a class of quadratics with permutation-invariant structures, the contraction rate upper bound for RPCD is always strictly smaller than the contraction rate lower bound for RCD for every individual problem instance. Furthermore, we conjecture that this function class contains the worst-case examples of RPCD among all positive-definite quadratics. Combined with our RCD lower bound, this conjecture extends our results to the general class of positive-definite quadratic functions.

## 1. Introduction

We consider the minimization problem:

$$\min_{\boldsymbol{x} \in \mathbb{R}^n} f(\boldsymbol{x}), \tag{1}$$

where $f : \mathbb{R}^n \to \mathbb{R}$ is a smooth and convex function.

The *coordinate descent* (CD) algorithm has been proposed and widely used for solving problem (1) arising in modern optimization and machine-learning problems as it can significantly reduce the computational overhead of high-dimensional or large-scale problems by updating only a single coordinate (or sometimes a small subset, referred to as blocks) instead of the whole set of parameters. Stemming from extensive studies on different types of CD algorithms (Beck & Tetruashvili, 2013; Lee & Sidford, 2013; Wright, 2015; Nesterov & Stich, 2017), *randomized* versions of CD have been especially popularized since Nesterov (2012); Richtárik & Takác (2011). Usually referred to as *random coordinate descent* (RCD), these types of algorithms choose the coordinates to update i.i.d. randomly from a certain distribution, typically uniform and sometimes specifically chosen according to the problem geometry. The introduction of randomness has been demonstrated in Sun & Ye (2016) to outperform deterministic algorithms like *cyclic coordinate descent* (CCD) in terms of worst-case performance.

Meanwhile, for stochastic algorithms under finite-sum minimization settings, utilizing *random permutations* has been common for a long time, based on observations that taking a full pass among the component gradient updates in a randomly permuted order (commonly referred to as SGD with Random Reshuffling) shows faster convergence speed than its ordinary, i.i.d. with-replacement-sampling counterpart. After empirical observations and some conjectures (Bottou, 2009; Recht & Re, 2012), a recent line of work has theoretically analyzed the exact convergence rates of SGD-RR and demonstrated the benefits over ordinary SGD (Ahn et al., 2020; Mishchenko et al., 2020; Cha et al., 2023; Liu & Zhou, 2024). Many other stochastic algorithms also incorporate random reshuffling for acceleration, including federated learning (Mishchenko et al., 2022; Yun et al., 2022) and minimax optimization (Das et al., 2022; Cho & Yun, 2023).

A similar variant also exists for CD, often referred to as *random-permutation coordinate descent* (RPCD) (Sun et al., 2020). While RPCD is based on a similar idea that using permutations can accelerate, theoretical analysis of RPCD is even harder because we must focus on the *preconditioning-like effects* of using permutations, which is very different from the case of SGD-RR (where random permutations induce *variance reduction*) and is notoriously difficult to inspect theoretically. However, for strongly convex quadratic functions with *permutation-invariant Hessians*, it is possible to compute the expectation of such matrices over all

[1]KAIST AI, Seoul, South Korea. Correspondence to: Chulhee Yun <chulhee.yun@kaist.ac.kr>.

*Proceedings of the $42^{nd}$ International Conference on Machine Learning*, Vancouver, Canada. PMLR 267, 2025. Copyright 2025 by the author(s).

permutations explicitly. Based on this approach, Lee & Wright (2019) (first appeared in 2016) derive convergence bounds for the expected *function value* for quadratic functions with permutation-invariant Hessians, and a follow-up work Wright & Lee (2017) further extends to Hessians with slight diagonal perturbations.

A natural, important question arises on *whether RPCD beats RCD*, just as in the case of SGD with random permutations Bottou (2009). Many empirical studies have shown this to be true, as we demonstrate in Figure 1, but the theoretical aspects of this phenomenon have been relatively less revealed in the literature. Gürbüzbalaban et al. (2018) demonstrate that it is possible to derive a stronger convergence upper bound for RPCD for permutation-invariant matrices with negative off-diagonal entries. However, they only compare between the *upper bounds* of RCD and RPCD, failing to demonstrate a rigorous gap between the two algorithms. Also, even for this comparison, the paper does not provide a clear analysis (other than numerical experiments) on the assertion that the ratio of contraction upper bounds for RCD to RPCD is larger than 1.

**Summary of Contributions.** Our results contribute to overcoming these limitations by adequately comparing the lower bounds for RCD and the upper bounds for RPCD. We present the following results in the paper.

- In Theorem 3.1, we show a novel convergence lower bound of RCD that holds for general quadratics with positive definite Hessians.

- In Theorem 3.3, we show the convergence upper bound of RPCD for a class of quadratic functions including permutation-invariant Hessians. This upper bound coincides with the RCD lower bound, concluding that RPCD outperforms RCD for *any* problem instance.

- In Theorem 3.4, we show a stronger convergence lower bound of RCD that holds for the same function class with the RPCD upper bounds. This demonstrates the existence of *a wider gap* between RCD and RPCD for all problem instances in this function class.

- In Section 4, we conjecture that our RPCD upper bounds can be extended to the general class of positive definite quadratic functions. We also provide some experiments demonstrating the convergence of RPCD and RCD in practice.

### 1.1. Related Work

**Randomized CD.** Nesterov (2012) presents global non-asymptotic convergence rates of RCD for (strongly) convex, smooth functions, where the probability of sampling each index is proportional to the $\alpha$-th power of the coordinate-wise

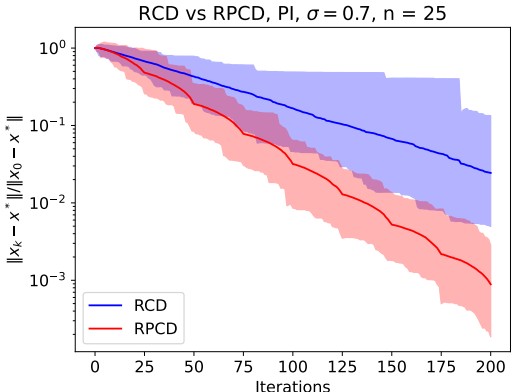

*Figure 1.* Performance comparison of RCD and RPCD. We use the objective function $f(\boldsymbol{x}) = \frac{1}{2}\boldsymbol{x}^\top \boldsymbol{A}\boldsymbol{x}$ with $\boldsymbol{A} = \sigma\boldsymbol{I} + (1-\sigma)\mathbf{1}\mathbf{1}^\top$. For this plot we use $\sigma = 0.7$ and dimension $n = 25$.

Lipschitz constant. (The case $\alpha = 0$ corresponds to uniform sampling). Richtárik & Takáč (2011) further improve the constant factors of the convergence rate and extend to composite minimization problems. After Beck & Tetru-ashvili (2013) established global non-asymptotic convergence rates of block CCD with gradient-based updates, Sun & Ye (2016) show that RCD outperforms CCD for quadratics $f(\boldsymbol{x}) = \frac{1}{2}\boldsymbol{x}^\top \boldsymbol{A}\boldsymbol{x}$ with *permutation-invariant* Hessians of the form $\boldsymbol{A} = \sigma\boldsymbol{I} + (1-\sigma)\mathbf{1}\mathbf{1}^\top$ for some $\sigma \in (0, 1]$, having positive off-diagonals[1]. They also show that this is the worst-case instance of CCD via a (up-to-constant) matching lower bound. Gürbüzbalaban et al. (2017) demonstrate a different case analysis on when CCD outperforms RCD, the Hessians of which they refer to as *2-cyclic matrices*.

**Random Permutations.** One of the earliest works on RPCD by Sun et al. (2020) (first appeared in 2015) analyzes the convergence of (block-)RPCD. While the results cover all positive definite quadratic functions and also the more general method of *alternating direction method of multipliers* (ADMM), expected *iterate* convergence is a weaker guarantee than the expected *iterate norm* or *function value*. Another work by Lee & Wright (2019) (first appeared in public in 2016) derived convergence bounds for the expected *function value* for quadratic functions with permutation-invariant Hessians, and a follow-up work Wright & Lee (2017) further extends to Hessians with subtle diagonal perturbations, which turns out to be equivalent to $\boldsymbol{A} = \sigma\boldsymbol{I} + (1-\sigma)\boldsymbol{u}\boldsymbol{u}^\top$ where the entries of $\boldsymbol{u}$ are not too far from 1. Gürbüzbalaban et al. (2018) compares the contraction ratio upper bound for RCD and RPCD to demonstrate that RPCD has better convergence guarantees for permutation-invariant matrices with negative off-diagonal entries. (We later provide a more detailed, quantitative comparison in Table 2.)

---
[1] See Section 2 for definitions and notations.

| Reference | Algorithm | Function Class | Bounds | |
|---|---|---|---|---|
| Theorem 3.1 | RCD | All Quadratics | $\max\left\{\left(1-\frac{1}{n}\right)^n, \left(1-\frac{\sigma}{n}\right)^{2n}\right\}$ | (LB) |
| Theorem 3.3 | RPCD | Hessian in $\mathcal{A}_\sigma$ | $\max\left\{\left(1-\frac{1}{n}\right)^n, \left(1-\frac{\sigma}{n}\right)^{2n}\right\}$ | (UB) |
| Theorem 3.4 | RCD | Hessian in $\mathcal{A}_\sigma$ | $\left(1-\frac{1}{n}+\frac{(1-\sigma)^2}{n}\right)^n$ | (LB) |
| Appendix D | RPCD | Hessian in $\mathcal{A}_\sigma$ | $1-2\sigma-\frac{2\sigma}{n}+2\sigma^2+\mathcal{O}\left(\frac{\sigma^2}{n}\right)+\mathcal{O}\left(\sigma^3\right)$ | (UB) |

*Table 1.* Summary of our results for (a subclass of) strongly convex, smooth quadratic functions. The rightmost column indicates the upper/lower bounds (UB/LB) of the value $\lim_{K\to\infty}\left(\frac{\mathbb{E}\left[\|\boldsymbol{x}_K\|^2\right]}{\|\boldsymbol{x}_0\|^2}\right)^{1/K}$, where $\boldsymbol{x}_K$ is the algorithm output after either $K$ epochs of RPCD or $T=nK$ iterations of RCD.

## 2. Preliminaries

For a positive integer $N$, we write $[N] := \{1, 2, \ldots, N\}$. We denote the dimension (number of coordinates) by $n \in \mathbb{N}$. We define $\mathbb{S}^n$ ($\mathbb{S}^n_+$) as the collection of all symmetric (positive-definite) matrices of size $n \times n$. We write $\|\cdot\|$ for the Euclidean $\ell_2$-norm for vectors and the spectral norm (*i.e.*, operator norm) for matrices. We also use $\|\cdot\|_\infty$ for the Euclidean $\ell_\infty$-norm in an analogous sense. The symbols $\otimes$ and $\odot$ represent the matrix Kronecker product and the elementwise Hadamard product, respectively.

The spectral radius (*i.e.*, maximum absolute eigenvalue) of a matrix $\boldsymbol{M}$ is denoted by $\rho(\boldsymbol{M})$, and the minimum (maximum) eigenvalue of matrices $\boldsymbol{M}$ (with real eigenvalues) is denoted by $\lambda_{\min}(\boldsymbol{M})$ ($\lambda_{\max}(\boldsymbol{M})$). We denote by $\mathrm{tril}(\boldsymbol{M})$ the lower triangular part of $\boldsymbol{M}$, including the diagonals. We will usually denote the elements of matrices by lower-case letters with subscripts, as in $\boldsymbol{A} = (a_{ij})$, and the elements of vectors via parentheses indicating the index, as in $\boldsymbol{x} = (x(1), \ldots, x(n))$.

We denote by $\boldsymbol{I}_k$ the $k$-dimensional identity matrix, $\boldsymbol{1}_k$ the $k$-dimensional ones vector, and $\mathrm{diag}\{a_1, \ldots, a_n\}$ the diagonal matrix with $(i, i)$-th entry $a_i$, or the similarly constructed block diagonal matrix if the elements are matrices. (We may drop the $k$ subscript in obvious cases, usually when $k = n$.) Also, we denote by $\boldsymbol{e}_i \in \mathbb{R}^n$ the unit vector with the 1 in the $i$-th coordinate and $\boldsymbol{E}_i \in \mathbb{R}^{n \times n}$ by the unit matrix $\boldsymbol{e}_i \boldsymbol{e}_i^\top$.

### 2.1. Problem Settings

Our goal is to minimize the following quadratic function:

$$\min_{\boldsymbol{x}} f(\boldsymbol{x}) := \frac{1}{2}\boldsymbol{x}^\top \boldsymbol{A}\boldsymbol{x},$$

where the Hessian $\boldsymbol{A} \in \mathbb{S}^n_+$ is a *positive definite* matrix. We write $\boldsymbol{A} = (a_{ij})$ for the elements of $\boldsymbol{A}$. The objective will be to find the minimizer $\boldsymbol{x}^\star = \boldsymbol{0}$ of $f$.

For useful purposes, we define the following quantity.

**Definition 2.1.** We define $\sigma = \lambda_{\min}(\boldsymbol{D}^{-1}\boldsymbol{A})$, where $\boldsymbol{D} = \mathrm{diag}(a_{11}, \ldots, a_{nn})$ is the diagonal part of $\boldsymbol{A}$. In particular,

if $\boldsymbol{D} = \boldsymbol{I}$, then we have $\sigma = \lambda_{\min}(\boldsymbol{A})$.

By definition, we must have $\sigma \in (0, 1]$. Later in Assumption 2.2, we justify why we can set $\boldsymbol{D} = \boldsymbol{I}$ without loss of generality. For such cases, $\sigma$ is equivalent to the strong convexity constant of the function $f$, and the convergence bounds we derive will depend on $\sigma$.

*Remark.* Our analysis automatically includes the translated quadratics $f(\boldsymbol{x}) = \frac{1}{2}\boldsymbol{x}^\top \boldsymbol{A}\boldsymbol{x} + \boldsymbol{b}^\top \boldsymbol{x} + c$ for any $\boldsymbol{b} \in \mathbb{R}^n$ and $c \in \mathbb{R}$, as long as the Hessian $\boldsymbol{A}$ is positive definite. We can straightforwardly replace $\boldsymbol{x}$ with $\boldsymbol{x} - \boldsymbol{x}^\star$ for $\boldsymbol{x}^\star = \boldsymbol{A}^{-1}\boldsymbol{b}$, the minimizer of the translated problem Wright & Lee (2017).

### 2.2. Algorithms

Algorithm 1 shows a general framework for coordinate descent (CD) methods. We focus on two versions of CD that differ in how we choose the index $i_t$ at each iteration.

- For *random coordinate descent* (RCD), we choose

$$i_t \sim \mathrm{Unif}([n]). \tag{2}$$

- For *random-permutation coordinate descent* (RPCD), we use $T = nK$ iterations ($K$ is the number of epochs). For each $k = 0, \ldots, K-1$ we choose a permutation $p_k$ of $[n]$ uniformly at random and choose

$$i_t = p_k(\ell + 1), \tag{3}$$

where $t = nk + \ell$ with $\ell \in \{0, \ldots, n-1\}$.

*Remark.* For quadratic objectives $f(\boldsymbol{x}) = \frac{1}{2}\boldsymbol{x}^\top \boldsymbol{A}\boldsymbol{x}$, we can rewrite the $\arg\min$ updates of Algorithm 1 in explicit form:

$$\boldsymbol{x}_{t+1} = \boldsymbol{x}_t - \frac{1}{a_{i_t i_t}}\boldsymbol{E}_{i_t}\boldsymbol{A}\boldsymbol{x}_t = \boldsymbol{x}_t - \frac{1}{a_{i_t i_t}}\boldsymbol{E}_{i_t}\nabla f(\boldsymbol{x}_t)$$

which can also be viewed as a coordinate *gradient* descent method with step size $\eta_{i_t} = \frac{1}{a_{i_t i_t}}$. Such gradient-based methods are often used as a proxy of CD when it is hard to compute the exact $\arg\min$s via line search. In the case of quadratics, the two are essentially equivalent.

**Algorithm 1** Coordinate Descent (CD)

---
**Input:** Number of iterations $T$
**Initialize:** $\boldsymbol{x}_0 \in \mathbb{R}^n$
**for** $t = 0$ **to** $T - 1$ **do**
    Choose the index $i_t \in [n]$ to update
    Update $\boldsymbol{x}_{t+1} = (x_t(1), \ldots, \underbrace{x'}_{i_t\text{-th}}, \ldots, x_t(n))$, where
$$x' = \arg\min_x f(x_t(1), \ldots, \underbrace{x}_{i_t\text{-th}}, \ldots, x_t(n))$$
**end for**
**Output:** $\boldsymbol{x}_T \in \mathbb{R}^n$

---

**RCD.** As shown in Equation (2), RCD randomly chooses a coordinate index per iteration to update.

We can concisely write one iteration of RCD as follows:

$$\boldsymbol{x}_{t+1} = \boldsymbol{T}_{\boldsymbol{A},i}^{\text{RCD}} \boldsymbol{x}_t,$$

where we choose to update index $i_t = i$ and we define

$$\boldsymbol{T}_{\boldsymbol{A},i}^{\text{RCD}} := \boldsymbol{I} - \frac{1}{a_{ii}} \boldsymbol{E}_i \boldsymbol{A}. \tag{4}$$

*Remark.* Nesterov (2012) also considers RCD with indices $i_t$ sampled from a *non-uniform* distribution, primarily when the probabilities are proportional to the Lipschitz constants of each coordinate. However, in our work, we only focus on the uniformly sampled case as our purpose is to make *a fair comparison with RPCD*.

**RPCD.** As shown in Equation (3), RPCD randomly chooses a permutation of indices and then takes coordinate updates in the order of the permutation we chose.

This time, we focus on one *epoch* (containing $n$ iterates) of RPCD for quadratic objectives $f(\boldsymbol{x}) = \frac{1}{2}\boldsymbol{x}^\top \boldsymbol{A}\boldsymbol{x}$. Accordingly, here we denote by $\boldsymbol{x}_{k+1}$ the iterate after applying one epoch (of $n$ updates) to $\boldsymbol{x}_k$. For a better illustration, we follow Sun et al. (2020) and consider the simple case when $n = 3$ and $p_0 = (123)$ is the identity permutation (which can also be viewed as an epoch of cyclic coordinate descent). Observing that we update only the $i$-th coordinate at the $i$-th iteration (which won't change for the rest of the epoch), we can write RPCD in terms of coordinates as follows:

$$x_1(1) = x_0(1) - \tfrac{1}{a_{11}}\left(a_{11}x_0(1) + a_{12}x_0(2) + a_{13}x_0(3)\right)$$
$$x_1(2) = x_0(2) - \tfrac{1}{a_{22}}\left(a_{21}x_1(1) + a_{22}x_0(2) + a_{23}x_0(3)\right)$$
$$x_1(3) = x_0(3) - \tfrac{1}{a_{33}}\left(a_{31}x_1(1) + a_{32}x_1(2) + a_{33}x_0(3)\right)$$

which can be rearranged and written in matrix form as:

$$\begin{bmatrix} a_{11} & 0 & 0 \\ a_{21} & a_{22} & 0 \\ a_{31} & a_{32} & a_{33} \end{bmatrix} \boldsymbol{x}_1 = -\begin{bmatrix} 0 & a_{12} & a_{13} \\ 0 & 0 & a_{23} \\ 0 & 0 & 0 \end{bmatrix} \boldsymbol{x}_0,$$

or more concisely:

$$\boldsymbol{x}_1 = -\boldsymbol{\Gamma}^{-1}(\boldsymbol{A} - \boldsymbol{\Gamma})\boldsymbol{x}_0$$
$$= (\boldsymbol{I} - \boldsymbol{\Gamma}^{-1}\boldsymbol{A})\boldsymbol{x}_0$$

where $\boldsymbol{\Gamma} = \text{tril}(\boldsymbol{A})$. In the general case, we can similarly observe that one epoch of RPCD with permutation $p_k = p$ boils down to

$$\boldsymbol{x}_{k+1} = \boldsymbol{T}_{\boldsymbol{A},p}^{\text{RPCD}} \boldsymbol{x}_k,$$

where we define

$$\boldsymbol{T}_{\boldsymbol{A},p}^{\text{RPCD}} := \boldsymbol{I} - \boldsymbol{P}\boldsymbol{\Gamma}_{\boldsymbol{P}}^{-1}\boldsymbol{P}^\top \boldsymbol{A}. \tag{5}$$

Here, $\boldsymbol{P} \in \{0,1\}^{n \times n}$ is the permutation matrix generated by the permutation $p$ (*i.e.*, $\boldsymbol{P}\boldsymbol{e}_i = \boldsymbol{e}_{p(i)}$ for all $i \in [n]$) and $\boldsymbol{\Gamma}_{\boldsymbol{P}} = \text{tril}(\boldsymbol{P}^\top \boldsymbol{A}\boldsymbol{P})$.

**Unit-diagonal Assumption.** Here we state and justify the following assumption on the Hessian $\boldsymbol{A}$ of $f$.

**Assumption 2.2.** We assume that the Hessian $\boldsymbol{A}$ has unit diagonals, or equivalently, $a_{11} = \cdots = a_{nn} = 1$.

This assumption might seem restrictive, but this is *without loss of generality for any coordinate-descent type methods on quadratics*. For any nonzero diagonal matrix $\boldsymbol{F}$, every iterate of a certain CD algorithm (RCD, RPCD, etc.) applied on $\tilde{f}(\boldsymbol{x}) = \frac{1}{2}\boldsymbol{x}^\top \widetilde{\boldsymbol{A}}\boldsymbol{x}$ for $\widetilde{\boldsymbol{A}} = \boldsymbol{F}^{-1}\boldsymbol{A}\boldsymbol{F}^{-1}$ proceeds as $\tilde{\boldsymbol{x}}_k = \boldsymbol{F}\boldsymbol{x}_k$, where $\boldsymbol{x}_k$ is the trajectory of the same algorithm applied on $f(\boldsymbol{x})$ with the same initialization and choices of $i_k$ (see Appendix A of Wright & Lee (2017)). We may therefore assume that $\boldsymbol{D} = \boldsymbol{I}$ by choosing $\boldsymbol{F} = \boldsymbol{D}^{\frac{1}{2}}$. (Note that $\boldsymbol{D}$ is defined as the diagonal part of $\boldsymbol{A}$.)

**Sign-flip Invariance.** Another particularly useful case of the diagonal transformation is when $\boldsymbol{F} = \text{diag}(\boldsymbol{v})$ with $\boldsymbol{v} \in \{\pm 1\}^n$. In this setting, we have $\widetilde{\boldsymbol{A}} = \boldsymbol{F}^{-1}\boldsymbol{A}\boldsymbol{F}^{-1} = \boldsymbol{A} \odot \boldsymbol{v}\boldsymbol{v}^\top$, which corresponds to flipping the signs of specific rows and columns of $\boldsymbol{A}$.

### 2.3. Matrix Operators

Suppose that $f(\boldsymbol{x}) = \frac{1}{2}\boldsymbol{x}^\top \boldsymbol{A}\boldsymbol{x}$ and we have some iteration that proceeds according to the following update:

$$\boldsymbol{x}_{k+1} = \boldsymbol{T}_{\boldsymbol{A}}\boldsymbol{x}_k$$

where the iteration matrix $\boldsymbol{T}_{\boldsymbol{A}} \in \mathbb{R}^{n \times n}$ is i.i.d. random and independent with the iterate $\boldsymbol{x}_k$. Let us define a linear matrix operator $\mathcal{M}_{\boldsymbol{A}} : \mathbb{R}^{n \times n} \to \mathbb{R}^{n \times n}$ as

$$\mathcal{M}_{\boldsymbol{A}}(\boldsymbol{X}) := \mathbb{E}\left[\boldsymbol{T}_{\boldsymbol{A}}^\top \boldsymbol{X}\boldsymbol{T}_{\boldsymbol{A}}\right].$$

Note that we can also write $\mathcal{M}_{\boldsymbol{A}}$ in matrix form by vectorizing the input and output matrices:

$$\text{vec}(\mathcal{M}_{\boldsymbol{A}}(\boldsymbol{X})) = \mathbb{E}\left[\boldsymbol{T}_{\boldsymbol{A}}^\top \otimes \boldsymbol{T}_{\boldsymbol{A}}^\top\right]\text{vec}(\boldsymbol{X}).$$

We denote the matrix operator for RCD and RPCD, generated by the random matrix iterations $\boldsymbol{T}_{\boldsymbol{A}}^{\text{RCD}}$ and $\boldsymbol{T}_{\boldsymbol{A}}^{\text{RPCD}}$ by $\mathcal{M}_{\boldsymbol{A}}^{\text{RCD}}$ and $\mathcal{M}_{\boldsymbol{A}}^{\text{RPCD}}$, respectively.

For the case of RCD, we can plug in $\boldsymbol{T}_{\boldsymbol{A},i} = \boldsymbol{I} - \boldsymbol{E}_i\boldsymbol{A}$ (as in (4) with $a_{ii} = 1$) to explicitly compute

$$
\mathcal{M}_{\boldsymbol{A}}(\boldsymbol{X}) = \mathbb{E}_i\left[(\boldsymbol{I} - \boldsymbol{E}_i\boldsymbol{A})^\top \boldsymbol{X}(\boldsymbol{I} - \boldsymbol{E}_i\boldsymbol{A})\right]
$$

$$
= \left(\boldsymbol{I} - \frac{\boldsymbol{A}}{n}\right)^\top \boldsymbol{X}\left(\boldsymbol{I} - \frac{\boldsymbol{A}}{n}\right)
$$

$$
+ \frac{\boldsymbol{A}^\top(n \cdot \text{diag}(\boldsymbol{X}) - \boldsymbol{X})\boldsymbol{A}}{n^2}.
$$

To motivate the use of such matrix operators, suppose that we define the convergence measure as the quadratic form below, where $\boldsymbol{\Theta} \in \mathbb{S}_+^n$ is a positive definite matrix that might depend on $\boldsymbol{A}$ but not on $\boldsymbol{x}_k$:

$$
\Psi_k = \frac{1}{2}\boldsymbol{x}_k^\top \boldsymbol{\Theta}\boldsymbol{x}_k. \tag{6}
$$

This (up to scaling) includes typical choices like the *squared Euclidean norm* $\|\boldsymbol{x}_k\|^2$ (using $\boldsymbol{\Theta} = \boldsymbol{I}$) and the *function value* $f(\boldsymbol{x}_k)$ (using $\boldsymbol{\Theta} = \boldsymbol{A}$). Then we can observe that

$$
\frac{\mathbb{E}\left[\Psi_k\right]}{\Psi_0} = \frac{\boldsymbol{y}_0^\top\left(\boldsymbol{\Theta}^{-1/2}\mathcal{M}_{\boldsymbol{A}}^k(\boldsymbol{\Theta})\boldsymbol{\Theta}^{-1/2}\right)\boldsymbol{y}_0}{\|\boldsymbol{y}_0\|^2} \tag{7}
$$

where $\boldsymbol{y}_0 := \boldsymbol{\Theta}^{1/2}\boldsymbol{x}_0$ and the expectation above is conditioned by $\boldsymbol{x}_0$. Similarly, we omit the conditional notation part of the expectations if clear by context.

**Diagonalizability.** If a linear matrix operator $\mathcal{M}_{\boldsymbol{A}}$ is diagonalizable, then we can view $\mathcal{M}_{\boldsymbol{A}}^k(\boldsymbol{\Theta})$ in (7) as a power iteration. The limit as $k \to \infty$ will be dominated by the eigenmatrix of $\mathcal{M}_{\boldsymbol{A}}$ with the largest absolute eigenvalue among nonzero components of the eigendecomposition of $\boldsymbol{\Theta}$. Hence we can understand the spectral radius of the operator $\rho(\mathcal{M}_{\boldsymbol{A}})$ as an upper bound of the contraction for convergence measures.

**Lemma 2.3.** *The matrix operators $\mathcal{M}_{\boldsymbol{A}}^{\text{RCD}}$ and $\mathcal{M}_{\boldsymbol{A}}^{\text{RPCD}}$ are both diagonalizable.*

While we defer the proof of Lemma 2.3 to Appendix A.1, the main idea is to define a similar operator:

$$
\widetilde{\mathcal{M}}_{\boldsymbol{A}}(\boldsymbol{X}) = \boldsymbol{A}^{-\frac{1}{2}}\mathcal{M}_{\boldsymbol{A}}(\boldsymbol{A}^{\frac{1}{2}}\boldsymbol{X}\boldsymbol{A}^{\frac{1}{2}})\boldsymbol{A}^{-\frac{1}{2}} \tag{8}
$$

which can be shown to have a *symmetric matrix form*

$$
\widetilde{\boldsymbol{T}}_{\boldsymbol{A}}^\top \otimes \widetilde{\boldsymbol{T}}_{\boldsymbol{A}}^\top, \quad \widetilde{\boldsymbol{T}}_{\boldsymbol{A}} := \boldsymbol{A}^{\frac{1}{2}}\boldsymbol{T}_{\boldsymbol{A}}\boldsymbol{A}^{-\frac{1}{2}} \tag{9}
$$

for both $\mathcal{M}_{\boldsymbol{A}}^{\text{RCD}}$ and $\mathcal{M}_{\boldsymbol{A}}^{\text{RPCD}}$, therefore is diagonalizable.

Furthermore, if we can show that $\mathcal{M}_{\boldsymbol{A}}$ is closed inside a subspace $\mathcal{S}$ which contains positive definite matrices, then choosing any $\boldsymbol{\Theta} \in \mathcal{S}$, it suffices to show an upper bound of $\rho(\mathcal{M}_{\boldsymbol{A}}|_{\mathcal{S}})$, the spectral radius of $\mathcal{M}_{\boldsymbol{A}}$ restricted at $\mathcal{S}$.

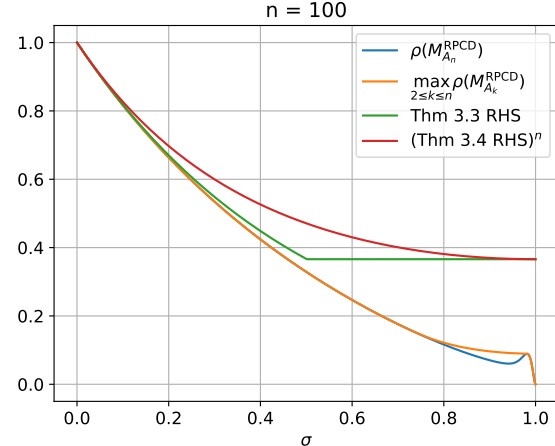

n = 100

**Figure 2.** Plots of $\rho(\mathcal{M}_{\boldsymbol{A}_n}^{\text{RPCD}}|_{\mathcal{S}})$ (blue), $\max_{2 \leq k \leq n} \rho(\mathcal{M}_{\boldsymbol{A}_k}^{\text{RPCD}}|_{\mathcal{S}})$ (yellow), the RPCD upper bound in Theorem 3.3 (green), and the $n$-th power (for fair comparison) of the stronger RCD lower bound for $\boldsymbol{A} \in \mathcal{A}_\sigma$ in Theorem 3.4 (red).

## 3. RCD vs RPCD

In this section, we compare the convergence rates of RCD and RPCD, under the problem setting defined in Section 2.

### 3.1. RCD Lower Bounds

Theorem 3.1 provides the convergence lower bound of RCD for the expected iterate norm. Note that this convergence measure is equivalent to choosing $\boldsymbol{\Theta} = \boldsymbol{I}$ in (6).

**Theorem 3.1.** *For an initial point $\boldsymbol{x}_0 \in \mathbb{R}^n$, let $\boldsymbol{x}_T$ be the output of RCD after $T$ iterates. Then, except for a Lebesgue measure zero set of initial points,*

$$
\lim_{T\to\infty}\left(\frac{\mathbb{E}\left[\|\boldsymbol{x}_T\|^2\right]}{\|\boldsymbol{x}_0\|^2}\right)^{\frac{1}{T}} \geq \max\left\{\left(1 - \frac{1}{n}\right), \left(1 - \frac{\sigma}{n}\right)^2\right\}.
$$

The green graph in Figure 2 coincides with the ($n$-th power of the) RCD lower bound. We can observe that the former term of the max dominates for large $\sigma$ and the latter for small $\sigma$, where the transition happens near $\sigma \approx \frac{1}{2}$.

We defer the proof of Theorem 3.1 to Appendix B.1.

*Remark.* For general convergence lower bounds for a certain function class, the *existence* of a bad function and initialization usually suffices. Our results show an even stronger lower bound argument that holds for *almost all initialization points* and, altogether with Theorem 3.3, shows a performance gap for *every specific instance* in the function class.

### 3.2. RPCD Upper Bounds

In this section, we focus on the case when $\boldsymbol{A}$ is either permutation-invariant or is closely related to such a matrix.

In particular, we focus on the following class of Hessians.

**Definition 3.2.** We define the class of Hessians $\mathcal{A}_\sigma^{\mathrm{PI}}, \mathcal{A}_\sigma$ as

$$\mathcal{A}_\sigma^{\mathrm{PI}} := \left\{ \mathrm{diag}\{\sigma \boldsymbol{I}_k + (1-\sigma)\boldsymbol{1}_k\boldsymbol{1}_k^\top, \boldsymbol{I}_{n-k}\} : 2 \le k \le n \right\},$$
$$\mathcal{A}_\sigma := \left\{ \boldsymbol{A} = \boldsymbol{A}^{\mathrm{PI}} \odot \boldsymbol{v}\boldsymbol{v}^\top : \boldsymbol{A}^{\mathrm{PI}} \in \mathcal{A}_\sigma^{\mathrm{PI}}, \boldsymbol{v} \in \{\pm 1\}^n \right\}$$

for given $\sigma \in (0, 1]$.

Note that $\sigma \boldsymbol{I}_k + (1-\sigma)\boldsymbol{1}_k\boldsymbol{1}_k^\top$ is a unit-diagonal, permutation-invariant matrix with a $\lambda_{\min}$ of $\sigma$, and that the sign-flip invariance of CD (see the end of Section 2.2) allows us to extend any type of convergence analysis from $\mathcal{A}_\sigma^{\mathrm{PI}}$ to $\mathcal{A}_\sigma$.

Theorem 3.3 provides the convergence upper bound of RPCD for the expected iterate norm.

**Theorem 3.3.** *For an initial point $\boldsymbol{x}_0 \in \mathbb{R}^n$, let $\boldsymbol{x}_K$ be the output of RPCD after $K$ epochs. If $\boldsymbol{A} \in \mathcal{A}_\sigma$ and $\boldsymbol{x}_0 \ne 0$,*

$$\lim_{K \to \infty} \left( \frac{\mathbb{E}\left[\|\boldsymbol{x}_K\|^2\right]}{\|\boldsymbol{x}_0\|^2} \right)^{\frac{1}{K}} \le \max\left\{ \left(1 - \frac{1}{n}\right)^n, \left(1 - \frac{\sigma}{n}\right)^{2n} \right\}.$$

Summarizing our results in Theorem 3.1 and Theorem 3.3, we can show that RPCD outperforms RCD for *all* instances of the form $\boldsymbol{A} = \mathrm{diag}\{\sigma\boldsymbol{I}_k + (1-\sigma)\boldsymbol{1}_k\boldsymbol{1}_k^\top, \boldsymbol{I}_{n-k}\}$. The threshold between the two bounds is

$$\max\left\{ \left(1 - \frac{1}{n}\right)^n, \left(1 - \frac{\sigma}{n}\right)^{2n} \right\}, \tag{10}$$

where we recall that $n$ iterations of RCD are equivalent to one epoch of RPCD for a fair comparison.

**Proof Sketch.** The proof starts from the observation that if $\boldsymbol{A} \in \mathcal{S} := \mathrm{span}\{\boldsymbol{I}, \boldsymbol{1}\boldsymbol{1}^\top\}$, or considering Assumption 2.2:

$$\boldsymbol{A} = \sigma\boldsymbol{I} + (1-\sigma)\boldsymbol{1}\boldsymbol{1}^\top, \tag{11}$$

then $\mathcal{M}_{\boldsymbol{A}}^{\mathrm{RPCD}}$ is closed in $\mathcal{S} = \mathrm{span}\{\boldsymbol{I}, \boldsymbol{1}\boldsymbol{1}^\top\}$. This allows us to convert the *restricted matrix operator*

$$\mathcal{M}_{\boldsymbol{A}}^{\mathrm{RPCD}}\big|_{\mathcal{S}}$$

into a $2 \times 2$ matrix $\boldsymbol{M}_{\boldsymbol{A}}$.

Therefore, if we choose any $\boldsymbol{\Theta} \in \mathcal{S}$ with $\boldsymbol{\Theta} \succ 0$ to define the convergence measure (we choose $\boldsymbol{\Theta} = \boldsymbol{I}$ for Theorem 3.3), the matrix $\left(\mathcal{M}_{\boldsymbol{A}}^{\mathrm{RPCD}}\right)^k (\boldsymbol{\Theta})$ will stay in $\mathrm{span}\{\boldsymbol{I}, \boldsymbol{1}\boldsymbol{1}^\top\}$ and it suffices to find an upper bound of $\rho(\mathcal{M}_{\boldsymbol{A}}^{\mathrm{RPCD}}\big|_{\mathcal{S}}) = \rho(\boldsymbol{M}_{\boldsymbol{A}})$ for this $2 \times 2$ matrix. We also show that the matrix

$$\boldsymbol{A} = \mathrm{diag}\{\sigma\boldsymbol{I}_k + (1-\sigma)\boldsymbol{1}_k\boldsymbol{1}_k^\top, \boldsymbol{I}_{n-k}\}$$

for all $2 \le k \le n$ has the same value of $\rho(\mathcal{M}_{\boldsymbol{A}}^{\mathrm{RPCD}})$ with that of the $k \times k$ matrix $\boldsymbol{A}_k = \sigma\boldsymbol{I}_k + (1-\sigma)\boldsymbol{1}_k\boldsymbol{1}_k^\top$.

After this, we upper bound $\rho(\boldsymbol{M}_{\boldsymbol{A}_k})$ with $\|\boldsymbol{M}_{\boldsymbol{A}_k}\|_\infty$, which can be expressed as the maximum of two polynomials of

$\sigma$. We then proceed to divide the range of $\sigma$ into three regions: $(0, 0.6]$, $[0.6, 0.8]$, and $[0.8, 1.0]$. Perhaps interestingly, we can find a (nearly) sign-alternating trend in the coefficients of both the polynomials from $\|\boldsymbol{M}_{\boldsymbol{A}_k}\|_\infty$ and the upper bound. This allows us to use lower-order Taylor approximations instead of the true lower and upper bounds and compare these to complete the proof for the cases $(0, 0.6]$ and $[0.6, 0.8]$, where the difference is in which part of the $\max$ of the upper bound we use for comparison. For the remaining region $[0.8, 1.0]$, we can show that $\|\boldsymbol{M}_{\boldsymbol{A}_k}\|_\infty$ is always too small compared to $\left(1 - \frac{1}{n}\right)^n$.

We defer the full proof of Theorem 3.3 to Appendix B.2.

*Remark.* In Figure 2, we can observe that for permutation-invariant instances $\boldsymbol{A}_n$ with large enough $n$, a small bump appears in the graph of $\rho(\mathcal{M}_{\boldsymbol{A}_n}^{\mathrm{RPCD}})$ close to $\sigma = 1$. This breaks monotonicity and makes theoretical analysis harder, motivating us to separately handle the cases of small and large $\sigma$ in Theorem 3.3 and the proof.

### 3.3. Stronger RCD Lower Bound

If we solely focus on the class $\mathcal{A}_\sigma$, we can further show an RCD lower bound stronger than Theorem 3.1.

**Theorem 3.4.** *Let $\boldsymbol{A} \in \mathcal{A}_\sigma$. For an initial point $\boldsymbol{x}_0 \in \mathbb{R}^n$, let $\boldsymbol{x}_T$ be the output of RCD after $T$ iterates. Then, except for a Lebesgue measure zero set of initial points,*

$$\lim_{T \to \infty} \left( \frac{\mathbb{E}\left[\|\boldsymbol{x}_T\|^2\right]}{\|\boldsymbol{x}_0\|^2} \right)^{\frac{1}{T}} \ge 1 - \frac{1}{n} + \frac{(1-\sigma)^2}{n}.$$

Summarizing our results in Theorem 3.3 and Theorem 3.4, we can now show that there exists a clear gap between the convergence rates of RCD and RPCD for $\boldsymbol{A} \in \mathcal{A}_\sigma$. The green line depicted in Figure 2 is equal to the RPCD upper bounds in Theorem 3.3, which is strictly smaller than the red RCD lower bound graph in Theorem 3.4 at $\sigma \in (0, 1)$.

We defer the proof of Theorem 3.4 to Appendix B.3.

*Remark.* While our results are written in terms of the *expected iterate norm* $\mathbb{E}\left[\|\boldsymbol{x}_T\|^2\right]$, for any $f(\boldsymbol{x}) = \frac{1}{2}\boldsymbol{x}^\top\boldsymbol{A}\boldsymbol{x}$ where $\boldsymbol{A} \succ 0$ has unit diagonals and $\lambda_{\min}(\boldsymbol{A}) = \sigma$ we have

$$\frac{\sigma}{2}\|\boldsymbol{x}\|^2 \le f(\boldsymbol{x}) \le \frac{n - (n-1)\sigma}{2}\|\boldsymbol{x}\|^2,$$

and hence we can easily extend the same convergence analyses to *function values* by scaling with a constant factor.

**Non-asymptotic results.** The inequality between the RCD and RPCD bounds which we establish in the asymptotic limit, in fact, already holds after a finite number of epochs. The required number of epochs depends on $\sigma$, and the corresponding non-asymptotic result is given in Appendix G.

*Table 2.* A comparison of existing convergence rate upper bounds and our results for RPCD. Note that Wright & Lee (2017) consider a larger function class and Gürbüzbalaban et al. (2018) consider a different function. In the last row, we use $\alpha = \frac{1-\sigma}{n-1}$.

| Reference | Convergence Rate |
|---|---|
| Lee & Wright (2019) | $\mathcal{O}\left(\left(1 - 2\sigma - \frac{2\sigma}{n} + 2\sigma^2\right)^K\right)$ |
| Ours (Theorem 3.3) | $\mathcal{O}\left(\left(\max\left\{\left(1 - \frac{1}{n}\right)^n, \left(1 - \frac{\sigma}{n}\right)^{2n}\right\}\right)^K\right)$ |
| Ours (Appendix D) | $\mathcal{O}\left(\left(1 - 2\sigma - \frac{2\sigma}{n} + 2\sigma^2\right)^K\right)$ |
| Wright & Lee (2017) | $\mathcal{O}\left(K\left(1 - \frac{7\sigma}{5}\right)^K\right)$ |
| Gürbüzbalaban et al. (2018) | $\mathcal{O}\left(\left(1 - \frac{\sigma}{n}\left(\frac{(1+\alpha)^{2n}-1}{\alpha(\alpha+2)}\right)\right)^K\right)$ |

**Comparison with Previous Work.** Lee & Wright (2019) demonstrate that, if $n \geq 10$ and $\sigma \in (0, 0.4]$, the epoch-wise contraction ratio of RPCD is of order

$$1 - 2\sigma - \frac{2\sigma}{n} + 2\sigma^2 + \mathcal{O}\left(\frac{\sigma^2}{n}\right) + \mathcal{O}\left(\sigma^3\right)$$

for quadratic functions with permutation-invariant Hessians. While our result seems to be weaker in the sense that

$$\left(1 - \frac{\sigma}{n}\right)^{2n} = 1 - 2\sigma - \frac{\sigma^2}{n} + 2\sigma^2 + \mathcal{O}(\sigma^3),$$

we can deduce from the details in our proof that in fact, a tighter exposition of our result can recover the rate of Lee & Wright (2019) (see Appendix D) while also providing a stronger analysis by **(i)** finding the exact bounds with explicit coefficients (*without asymptotic terms*) and **(ii)** covering the whole region of $\sigma \in (0, 1]$ and smaller values of $n$ using the $(1 - \frac{1}{n})^n$ term and novel proof techniques.

Wright & Lee (2017) consider an extension to Hessians of the form $\sigma \boldsymbol{I} + (1 - \sigma)\boldsymbol{u}\boldsymbol{u}^\top$ where the coordinates of $\boldsymbol{u}$ ranges from $\sqrt{\sigma/(\sigma + \epsilon)}$ to 1 for some small $\epsilon$. Their convergence rate upper bound shown in Table 2 is quantitatively weaker, but it is not directly comparable because it applies to a slightly larger function class. Gürbüzbalaban et al. (2018) consider a different function with Hessian of the form $\boldsymbol{A} = (1 + \alpha)\boldsymbol{I} - \alpha \boldsymbol{1}\boldsymbol{1}^\top$, where $\sigma = 1 - (n-1)\alpha$, an instance for which CCD is also shown to outperform RCD (Gürbüzbalaban et al., 2017).

## 4. RPCD on General Quadratics

One limitation of our work is that the RPCD upper bounds in Theorem 3.3 only apply to problem instances in $\boldsymbol{A} \in \mathcal{A}_\sigma$. Here we leave some noteworthy discussions considering ways to generalize to the *entire class of quadratic functions*.

### 4.1. Conjecture on the RPCD Worst Case

Here we propose a conjecture that the RPCD upper bound we have shown for $\boldsymbol{A} \in \mathcal{A}_\sigma$ in Theorem 3.3 could possibly extend to an upper bound for all positive definite quadratics.

**Conjecture 4.1.** *For an initial point $\boldsymbol{x}_0 \in \mathbb{R}^n$, let $\boldsymbol{x}_K$ be the output of* RPCD *after $K$ epochs. If $\sigma \in (0, 1]$, $\boldsymbol{A} \in \mathbb{S}_+^n$ with $\lambda_{\min}(\boldsymbol{A}) = \sigma$, and $\boldsymbol{x}_0 \neq 0$, then*

$$\lim_{K \to \infty} \left(\frac{\mathbb{E}\left[\|\boldsymbol{x}_K\|^2\right]}{\|\boldsymbol{x}_0\|^2}\right)^{\frac{1}{K}} \leq \max\left\{\left(1 - \frac{1}{n}\right)^n, \left(1 - \frac{\sigma}{n}\right)^{2n}\right\}.$$

**Algorithmic Search.** We use the following framework to search for the worst-case example among the larger space of all (unit-diagonal) quadratics.

---

*Step 1.* Start with a (randomly initialized) lower triangular matrix $\boldsymbol{X} \in \mathbb{R}^{n \times n}$ with nonzero diagonals.

*Step 2.* Construct a unit-diagonal, positive semi-definite matrix by computing $\boldsymbol{Y} = \boldsymbol{X}^\top \boldsymbol{X}$ and set unit diagonals by $\boldsymbol{Z} = \boldsymbol{D}_Y^{-\frac{1}{2}} \boldsymbol{Y} \boldsymbol{D}_Y^{-\frac{1}{2}}$, where $\boldsymbol{D}_Y$ is the diagonal part of $\boldsymbol{Y}$.

*Step 3.* Construct a matrix $\boldsymbol{A}$ with $\sigma = \lambda_{\min}(\boldsymbol{A})$ by

$$\boldsymbol{A} = \frac{1 - \sigma}{1 - \mu}\boldsymbol{Z} + \frac{\sigma - \mu}{1 - \mu}\boldsymbol{I},$$

where $\mu = \lambda_{\min}(\boldsymbol{Z})$. Note that $\boldsymbol{A}$ is also unit-diagonal and positive semi-definite.

*Step 4.* Construct an objective function that takes an input $\boldsymbol{X}$ to compute the value of $\rho(\mathcal{M}_{\boldsymbol{A}}^{\text{RPCD}})$ and run a `scipy` optimizer to maximize the objective.

---

We explore dimensions $n = 3, 4, 5, 6$ and $\sigma$ values ranging from 0.1 to 0.9 with increments of 0.1. Note that **Algorithmic Search** finds the $\arg\max$ of $\rho(\mathcal{M}_{\boldsymbol{A}}^{\text{RPCD}})$, which is an *upper bound* of the convergence rates. The results suggest that $\boldsymbol{A} \in \mathcal{A}_\sigma$ are the worst cases concerning this upper bound, as all of our trials of **Algorithmic Search** converged to a problem instance $\boldsymbol{A} \in \mathcal{A}_\sigma$ with various sign flips. We also observed that for any $\boldsymbol{A} \in \mathcal{A}_\sigma$, we always have $\rho(\mathcal{M}_{\boldsymbol{A}}^{\text{RPCD}}) = \rho(\mathcal{M}_{\boldsymbol{A}}^{\text{RPCD}}|_{\mathcal{S}})$ (where $\mathcal{S} = \text{span}\{\boldsymbol{I}, \boldsymbol{1}\boldsymbol{1}^\top\}$), which we were able to show in Theorem 3.3 that is smaller than the same upper bound. These observations altogether motivated us to propose Conjecture 4.1 on the worst-case instance for general quadratic functions.

*Remark.* Sun & Ye (2016) show that $\boldsymbol{A} = \sigma \boldsymbol{I} + (1 - \sigma)\boldsymbol{1}\boldsymbol{1}^\top$ is the worst-case example of cyclic coordinate descent (CCD), which partially aligns with the idea of our conjecture that CD algorithms using one update per coordinate in each epoch (including both CCD and RPCD) are slow for similar types of matrices.

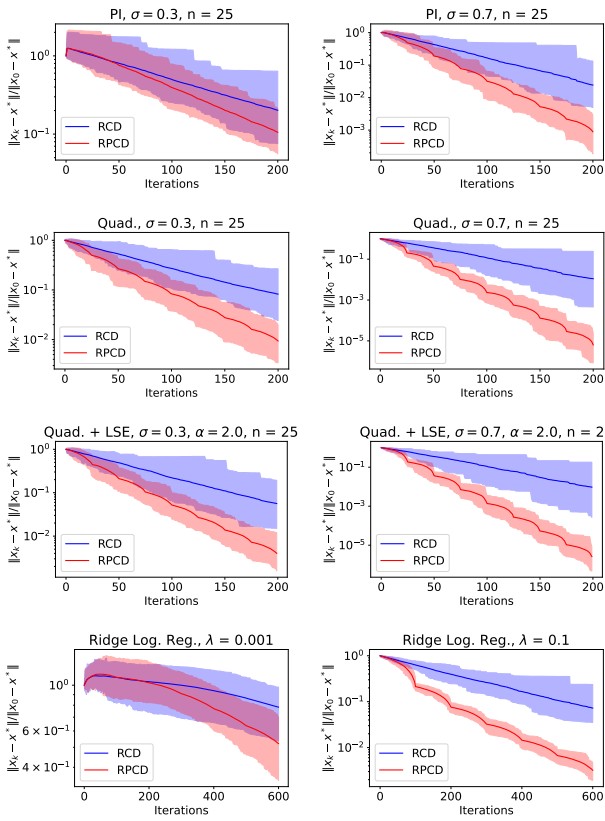

Figure 3. Numerical experiments comparing RCD and RPCD. We plot the mean values and min-max range over multiple trials of RCD/RPCD. For **(i)**-**(iii)**, we use $n = 25$, $\sigma \in \{0.3, 0.7\}$. The experiments are conducted on **(i)** a quadratic function with $\boldsymbol{A} \in \mathcal{A}_\sigma$, **(ii)** a random quadratic with $\lambda_{\min}(\boldsymbol{A}) = \sigma$, **(iii)** a random quadratic + (scaled and transformed) LSE function, and **(iv)** an $\ell_2$-regularized sparse logistic regression objective with $n = 100$.

## 4.2. Experiments

To compare the performance of RCD and RPCD, we conducted experiments on four types of convex and smooth functions: **(i)** quadratic functions with permutation-invariant Hessians, **(ii)** general quadratic functions with positive definite Hessians, **(iii)** general quadratic functions with positive definite Hessians *plus* a scaled log-sum-exp (LSE) term, and **(iv)** $\ell_2$-regularized logistic regression objective functions. Specifically, we use the following form for **(i)**-**(iii)**:

$$f(\boldsymbol{x}) = \frac{1}{2}\boldsymbol{x}^\top \boldsymbol{A}\boldsymbol{x} + \alpha \cdot \text{LSE}(\boldsymbol{Q}\boldsymbol{x}),$$

where $\boldsymbol{A}$ is either $\sigma\boldsymbol{I} + (1-\sigma)\boldsymbol{1}\boldsymbol{1}^\top$ or randomly generated with unit diagonals and $\lambda_{\min}(\boldsymbol{A}) = \sigma$ (following *Steps 1-3* of **Algorithmic Search**), $\boldsymbol{Q}$ is a randomly generated orthogonal matrix we use to avoid coordinate-friendly structures, $\text{LSE}(a_1, \ldots, a_n) = \log(e^{a_1} + \cdots + e^{a_n})$, and $\alpha \geq 0$.

For **(iv)**, we use the $\ell_2$-regularized logistic regression objec-

tive:

$$\min_x \frac{1}{m}\sum_{i=1}^{m} \log(1 + \exp(-b_i \boldsymbol{a}_i^\top \boldsymbol{x})) + \frac{\lambda}{2}\|\boldsymbol{x}\|^2,$$

where $\boldsymbol{a}_i \sim \mathcal{N}(0, \boldsymbol{I}_n)$ and the labels are generated by sampling $\boldsymbol{x}_{\text{true}} \sim \mathcal{N}(0, \boldsymbol{I}_n)$, setting $b_i = \text{sign}(\boldsymbol{a}_i^\top \boldsymbol{x}_{\text{true}})$, and flipping each $b_i$ independently with probability 0.1. This setup follows Nutini et al. (2015).

Our theoretical contribution focuses on establishing the superiority of RPCD over RCD for the specific case of quadratic functions with permutation-invariant Hessians **(i)**. As demonstrated in Figure 3, RPCD does not only show faster convergence than RCD for **(i)** in practice, but also in minimizing both **(ii)** and **(iii)**. Results for **(ii)** particularly support an immediate corollary of Conjecture 4.1; that RPCD is generally faster than RCD for *any problem instance* in the broader class of *quadratic functions*. Readers may refer to Appendix F for more details on the experiments.

### 4.3. Discussions

All previous results considering the convergence of *expected iterate norm* or *function value*[2] of RPCD (Lee & Wright, 2019; Wright & Lee, 2017) rely on the properties of permutation invariant matrices, for which $\mathcal{M}_{\boldsymbol{A}}^{\text{RPCD}}$ is closed in a certain low-dimensional matrix subspace $\mathcal{S}$. In this section, we first discuss a special portion of cases when we might be able to use similar arguments, then proceed to discussions regarding arbitrary positive definite quadratics for which we cannot use such an approach.

**Partially Invariant Hessians.** Suppose that we have a $4 \times 4$ unit-diagonal matrix of the form:

$$\boldsymbol{A} = \begin{bmatrix} 1 & a & a & a \\ a & 1 & b & b \\ a & b & 1 & b \\ a & b & b & 1 \end{bmatrix}.$$

For these types of matrices, there are only 4 possible cases of matrices $\boldsymbol{P}^\top \boldsymbol{A}\boldsymbol{P}$. Then we can define the bases using the following $(1 + (n-1)) \times (1 + (n-1))$ block matrices:

$$\boldsymbol{V}_1 = \begin{bmatrix} 1 & \boldsymbol{0} \\ \boldsymbol{0} & \boldsymbol{0} \end{bmatrix}, \quad \boldsymbol{V}_2 = \begin{bmatrix} 0 & \boldsymbol{1}^\top \\ \boldsymbol{1} & \boldsymbol{0} \end{bmatrix},$$

$$\boldsymbol{V}_3 = \begin{bmatrix} \boldsymbol{0} & \boldsymbol{0} \\ \boldsymbol{0} & \boldsymbol{I} \end{bmatrix}, \quad \boldsymbol{V}_4 = \begin{bmatrix} 0 & \boldsymbol{0} \\ \boldsymbol{0} & \boldsymbol{1}\boldsymbol{1}^\top \end{bmatrix},$$

and show $\mathcal{M}_{\boldsymbol{A}}^{\text{RPCD}}$ is closed in $\mathcal{S}' = \text{span}\{\boldsymbol{V}_1, \boldsymbol{V}_2, \boldsymbol{V}_3, \boldsymbol{V}_4\}$. Then the problem reduces into finding $\rho(\mathcal{M}_{\boldsymbol{A}}^{\text{RPCD}}|_{\mathcal{S}'})$ which

---

[2]Convergence of the expected iterate $\mathbb{E}[\boldsymbol{x}_T]$ is *weaker* than that of the expected iterate norm $\mathbb{E}[\|\boldsymbol{x}_T\|^2]$ or function value $\mathbb{E}[f(\boldsymbol{x}_T)]$ as the latter implies the former (if $f$ is convex) but not vice versa.

requires a fine spectral analysis of an asymmetric $4 \times 4$ matrix with two controllable variables $a, b \in [0, 1]$. See Appendix E.1 for a more detailed discussion.

**General Quadratics.** The only known results that consider RPCD for general quadratics are those by Sun et al. (2020), where they observe that for each permutation $p$,

$$
\begin{aligned}
\widetilde{T}_{A,p}^{\text{RPCD}} &= A^{\frac{1}{2}} T_{A,p}^{\text{RPCD}} A^{-\frac{1}{2}} \\
&= I - A^{\frac{1}{2}} P \Gamma_P^{-1} P^\top A^{\frac{1}{2}} = Z_{p(n)} \cdots Z_{p(1)},
\end{aligned}
\tag{12}
$$

where $Z_i = I - v_i v_i^\top$ is a projection matrix with $v_i$ being the $i$-th column of $A^{\frac{1}{2}}$. They then show *expected iterate*[2] convergence from $\rho(T_{A,p}^{\text{RPCD}}) = \rho(\widetilde{T}_{A,p}^{\text{RPCD}}) \leq 1 - \frac{\sigma}{n}$. The resulting iteration complexity upper bound is $n$ times loose compared to both our upper bounds and previous results for $A \in \mathcal{A}_\sigma$ that involves the $n$-th exponent, $(1 - \frac{\sigma}{n})^n$, instead.

For our goal, it is sufficient to show

$$
\begin{aligned}
\rho(\mathcal{M}_A^{\text{RPCD}}) &= \rho \left( \mathbb{E} \left[ T_A^{\text{RPCD}\top} \otimes T_A^{\text{RPCD}\top} \right] \right) \\
&\leq \max \left\{ \left( 1 - \frac{\sigma}{n} \right)^{2n}, \left( 1 - \frac{1}{n} \right)^n \right\}.
\end{aligned}
\tag{13}
$$

While there are no well-known analyses on the *upper bounds* of the *spectral radius* of the expectation of *Kronecker powers* of matrices over *permutations*, we can use (12) and $\rho(\mathcal{M}_A^{\text{RPCD}}) = \rho(\widetilde{\mathcal{M}}_A^{\text{RPCD}})$ (see (8)) to equivalently write[3]

$$
\begin{aligned}
\rho(\widetilde{\mathcal{M}}_A^{\text{RPCD}}) &= \rho \left( \mathbb{E} \left[ \widetilde{T}_A^{\text{RPCD}\top} \otimes \widetilde{T}_A^{\text{RPCD}\top} \right] \right) \\
&= \rho \left( \mathbb{E} \left[ \left( Z_{p(n)} \cdots Z_{p(1)} \right)^{\top \otimes 2} \right] \right).
\end{aligned}
$$

One possible strategy could be to use

$$
\rho(\mathcal{M}_A^{\text{RPCD}}) \leq \left\| A^{-1/2} \mathcal{M}_A^{\text{RPCD}}(A) A^{-1/2} \right\|
\tag{14}
$$

to circumvent the need to compute the sum of the Kronecker powers. We can use (12) to express the RHS of (14) as

$$
\mathbb{E} \left[ Z_{p(1)} \cdots Z_{p(n)} \cdots Z_{p(1)} \right].
$$

We have numerically checked that (14) is quite tight enough to be useful when $\sigma \leq \frac{1}{2}$, but unfortunately gets suddenly loose for $\sigma > \frac{1}{2}$ if $n$ gets large, as shown in Figure 4. See Appendix E.2 for proofs of some of the statements above and a more detailed discussion.

## 5. Conclusion

We have shown convergence lower bounds for RCD and convergence upper bounds for RPCD for positive definite quadratic functions with permutation-invariant structures.

---

[3]The notation $(\cdot)^{\otimes 2}$ is for the Kronecker product with itself.

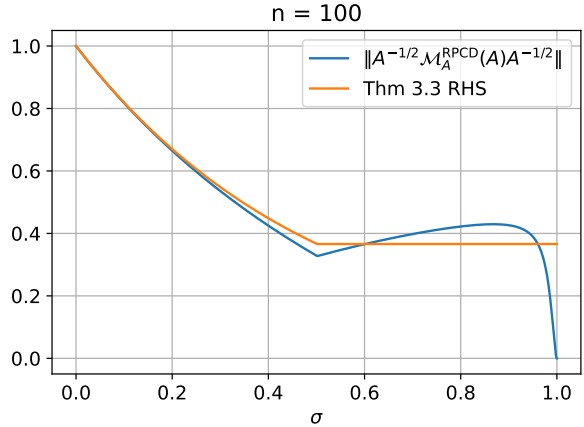

*Figure 4.* Plot of $\left\| A^{-\frac{1}{2}} \mathcal{M}_A(A) A^{-\frac{1}{2}} \right\|$ ($A = \sigma I + (1-\sigma)\mathbf{1}\mathbf{1}^\top$) and the RPCD upper bound from Theorem 3.3 for $n = 100$.

We obtain results demonstrating that RPCD outperforms RCD for any problem instance in $A \in \mathcal{A}_\sigma$ (Theorems 3.1 and 3.3), show a stronger RCD lower bound that induces a clear gap between RCD and RPCD (Theorem 3.4), and conjecture that RPCD will also outperform RCD for general quadratic functions (Conjecture 4.1).

An immediate future direction would be to prove (or disprove) Conjecture 4.1 for general quadratic functions, and possibly further analyze the benefits of random permutations for even broader function/algorithm classes, such as *non-quadratic* (strongly) convex functions or *block* coordinate descent algorithms.

## Acknowledgements

This work was supported by a National Research Foundation of Korea (NRF) grant funded by the Korean government (MSIT) (No. RS-2023-00211352) and an Institute for Information & communications Technology Planning & Evaluation (IITP) grant funded by the Korean government (MSIT) (No. RS-2019-II190075, Artificial Intelligence Graduate School Program (KAIST)).

## Impact Statement

This paper presents the convergence analyses of randomized coordinate descent algorithms. We do not have any specific ethical/societal impact or consequences of our work, other than trivial implications related to the advancement of the field of Machine Learning.

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

# A. Proofs in Section 2

## A.1. Proof of Lemma 2.3

Here we prove Lemma 2.3, restated below for the sake of readability.

**Lemma 2.3.** *The matrix operators $\mathcal{M}_{\boldsymbol{A}}^{\mathrm{RCD}}$ and $\mathcal{M}_{\boldsymbol{A}}^{\mathrm{RPCD}}$ are both diagonalizable.*

*Proof.* For $\mathcal{M}_{\boldsymbol{A}}^{\mathrm{RCD}}$, consider the matrix operator:

$$\widetilde{\mathcal{M}}_{\boldsymbol{A}}^{\mathrm{RCD}}(\boldsymbol{X}) = \boldsymbol{A}^{-\frac{1}{2}} \mathcal{M}_{\boldsymbol{A}}^{\mathrm{RCD}}(\boldsymbol{A}^{\frac{1}{2}} \boldsymbol{X} \boldsymbol{A}^{\frac{1}{2}}) \boldsymbol{A}^{-\frac{1}{2}},$$

a similar operator to $\mathcal{M}_{\boldsymbol{A}}^{\mathrm{RCD}}$ as it maps $\boldsymbol{A}^{-\frac{1}{2}} \boldsymbol{X} \boldsymbol{A}^{-\frac{1}{2}}$ to $\boldsymbol{A}^{-\frac{1}{2}} \mathcal{M}_{\boldsymbol{A}}^{\mathrm{RCD}}(\boldsymbol{X}) \boldsymbol{A}^{-\frac{1}{2}}$. This can be rewritten in the form of

$$\widetilde{\mathcal{M}}_{\boldsymbol{A}}^{\mathrm{RCD}}(\boldsymbol{X}) = \mathbb{E}\left[\widetilde{\boldsymbol{T}}_{\boldsymbol{A}}^{\mathrm{RCD}\top} \boldsymbol{X} \widetilde{\boldsymbol{T}}_{\boldsymbol{A}}^{\mathrm{RCD}}\right],$$

where the expectation is over *symmetric* matrices

$$\widetilde{\boldsymbol{T}}_{\boldsymbol{A},i}^{\mathrm{RCD}} := \boldsymbol{I} - \frac{1}{a_{ii}} \boldsymbol{A}^{\frac{1}{2}} \boldsymbol{E}_i \boldsymbol{A}^{\frac{1}{2}}.$$

Therefore the matrix form of $\widetilde{\mathcal{M}}_{\boldsymbol{A}}^{\mathrm{RCD}}$,

$$\mathbb{E}\left[\widetilde{\boldsymbol{T}}_{\boldsymbol{A},i}^{\mathrm{RCD}\top} \otimes \widetilde{\boldsymbol{T}}_{\boldsymbol{A},i}^{\mathrm{RCD}\top}\right]$$

must also be symmetric, which implies that $\widetilde{\mathcal{M}}_{\boldsymbol{A}}^{\mathrm{RCD}}$ is diagonalizable. Hence $\mathcal{M}_{\boldsymbol{A}}^{\mathrm{RCD}}$ must also be diagonalizable by similarity.

For $\mathcal{M}_{\boldsymbol{A}}^{\mathrm{RPCD}}$, we also consider the matrix operator:

$$\widetilde{\mathcal{M}}_{\boldsymbol{A}}^{\mathrm{RPCD}}(\boldsymbol{X}) = \boldsymbol{A}^{-\frac{1}{2}} \mathcal{M}_{\boldsymbol{A}}^{\mathrm{RPCD}}(\boldsymbol{A}^{\frac{1}{2}} \boldsymbol{X} \boldsymbol{A}^{\frac{1}{2}}) \boldsymbol{A}^{-\frac{1}{2}},$$

which is also similar to $\mathcal{M}_{\boldsymbol{A}}^{\mathrm{RPCD}}$ and can be rewritten as

$$\widetilde{\mathcal{M}}_{\boldsymbol{A}}^{\mathrm{RPCD}}(\boldsymbol{X}) = \mathbb{E}\left[\widetilde{\boldsymbol{T}}_{\boldsymbol{A}}^{\mathrm{RPCD}\top} \boldsymbol{X} \widetilde{\boldsymbol{T}}_{\boldsymbol{A}}^{\mathrm{RPCD}}\right],$$

where the expectation is over the matrices

$$\widetilde{\boldsymbol{T}}_{\boldsymbol{A},p}^{\mathrm{RPCD}} := \boldsymbol{I} - \boldsymbol{A}^{\frac{1}{2}} \boldsymbol{P} \boldsymbol{\Gamma}_{\boldsymbol{P}}^{-1} \boldsymbol{P}^{\top} \boldsymbol{A}^{\frac{1}{2}}.$$

For each permutation $p$, we consider a complementary permutation $p'$ defined as:

$$p'(i) = (n + 1) - p(i), \quad \forall i \in [n].$$

The permutation matrix $\boldsymbol{P}'$ generated by $p'$ satisfies $\boldsymbol{P}' \boldsymbol{\Gamma}_{\boldsymbol{P}'} \boldsymbol{P}'^{\top} = \left(\boldsymbol{P} \boldsymbol{\Gamma}_{\boldsymbol{P}} \boldsymbol{P}^{\top}\right)^{\top}$ (Sun et al., 2020). This is because

$$\boldsymbol{P} \boldsymbol{\Gamma}_{\boldsymbol{P}} \boldsymbol{P}^{\top} = \boldsymbol{P} \mathrm{tril}(\boldsymbol{P}^{\top} \boldsymbol{A} \boldsymbol{P}) \boldsymbol{P}^{\top} = \boldsymbol{A} \odot \boldsymbol{W}_{\boldsymbol{P}},$$

where $\boldsymbol{W}_{\boldsymbol{P}}$ is a binary matrix with

$$W_{\boldsymbol{P}ij} = \begin{cases} 1, & p(i) \geq p(j), \\ 0, & \text{otherwise.} \end{cases}$$

Since $p(i) \geq p(j)$ if and only if $p'(i) \geq p'(j)$, we have $\boldsymbol{W}_{\boldsymbol{P}'} = \boldsymbol{W}_{\boldsymbol{P}}^{\top}$ and $\boldsymbol{P} \boldsymbol{\Gamma}_{\boldsymbol{P}'} \boldsymbol{P}'^{\top} = \left(\boldsymbol{P} \boldsymbol{\Gamma}_{\boldsymbol{P}} \boldsymbol{P}^{\top}\right)^{\top}$.

Therefore we have $\widetilde{\boldsymbol{T}}_{\boldsymbol{A},p'}^{\mathrm{RPCD}} = (\widetilde{\boldsymbol{T}}_{\boldsymbol{A},p}^{\mathrm{RPCD}})^{\top}$, and the following sum:

$$\widetilde{\boldsymbol{T}}_{\boldsymbol{A},p}^{\mathrm{RPCD}\top} \otimes \widetilde{\boldsymbol{T}}_{\boldsymbol{A},p}^{\mathrm{RPCD}\top} + \widetilde{\boldsymbol{T}}_{\boldsymbol{A},p'}^{\mathrm{RPCD}\top} \otimes \widetilde{\boldsymbol{T}}_{\boldsymbol{A},p'}^{\mathrm{RPCD}\top}$$

is also symmetric as a sum of transposes. As the set of all permutations can be paired with its unique complementary permutation, the matrix form of $\widetilde{\mathcal{M}}_{\boldsymbol{A}}^{\mathrm{RPCD}}$,

$$\mathbb{E}\left[\widetilde{\boldsymbol{T}}_{\boldsymbol{A}}^{\mathrm{RPCD}\top} \otimes \widetilde{\boldsymbol{T}}_{\boldsymbol{A}}^{\mathrm{RPCD}\top}\right]$$

must be symmetric, which implies that $\widetilde{\mathcal{M}}_{\boldsymbol{A}}^{\mathrm{RPCD}}$ and $\mathcal{M}_{\boldsymbol{A}}^{\mathrm{RPCD}}$ must also be diagonalizable. □

# B. Proofs in Section 3

## B.1. Proof of Theorem 3.1

Here we prove Theorem 3.1, restated below for the sake of readability.

**Theorem 3.1.** *For an initial point $\boldsymbol{x}_0 \in \mathbb{R}^n$, let $\boldsymbol{x}_T$ be the output of RCD after $T$ iterates. Then, except for a Lebesgue measure zero set of initial points,*

$$\lim_{T \to \infty} \left( \frac{\mathbb{E}\left[\|\boldsymbol{x}_T\|^2\right]}{\|\boldsymbol{x}_0\|^2} \right)^{\frac{1}{T}} \geq \max\left\{ \left(1 - \frac{1}{n}\right), \left(1 - \frac{\sigma}{n}\right)^2 \right\}.$$

*Proof.* We will prove the following two inequalities:

$$\lim_{T \to \infty} \left( \frac{\mathbb{E}\left[\|\boldsymbol{x}_T\|^2 \mid \boldsymbol{x}_0\right]}{\|\boldsymbol{x}_0\|^2} \right)^{1/T} \geq 1 - \frac{1}{n},$$

$$\lim_{T \to \infty} \left( \frac{\mathbb{E}\left[\|\boldsymbol{x}_T\|^2 \mid \boldsymbol{x}_0\right]}{\|\boldsymbol{x}_0\|^2} \right)^{1/T} \geq \left(1 - \frac{\sigma}{n}\right)^2.$$

To show the first inequality, we use the fact that $\mathcal{M}_{\boldsymbol{A}}^{\mathrm{RCD}}(\boldsymbol{I}) = \boldsymbol{I} - \frac{2\boldsymbol{A}}{n} + \frac{\boldsymbol{A}^2}{n}$. Then,

$$\mathbb{E}\left[\boldsymbol{x}_k^\top \boldsymbol{x}_k \mid \boldsymbol{x}_{k-1}\right] = \boldsymbol{x}_{k-1}^\top \mathcal{M}_{\boldsymbol{A}}^{\mathrm{RCD}}(\boldsymbol{I}) \boldsymbol{x}_{k-1}$$

$$= \boldsymbol{x}_{k-1}^\top \left(\boldsymbol{I} - \frac{2\boldsymbol{A}}{n} + \frac{\boldsymbol{A}^2}{n}\right) \boldsymbol{x}_{k-1}$$

$$\geq \lambda_{\min}\left(\boldsymbol{I} - \frac{2\boldsymbol{A}}{n} + \frac{\boldsymbol{A}^2}{n}\right) \boldsymbol{x}_{k-1}^\top \boldsymbol{x}_{k-1}.$$

Since the eigenvalues of $\boldsymbol{I} - \frac{2\boldsymbol{A}}{n} + \frac{\boldsymbol{A}^2}{n}$ are in the form of $1 - \frac{2\lambda}{n} + \frac{\lambda^2}{n}$, where $\lambda$ is an eigenvalue of $\boldsymbol{A}$, and $1 - \frac{2x}{n} + \frac{x^2}{n} \geq 1 - \frac{1}{n}$ for any $x \in \mathbb{R}$, we have

$$\mathbb{E}\left[\boldsymbol{x}_k^\top \boldsymbol{x}_k \mid \boldsymbol{x}_{k-1}\right] \geq \left(1 - \frac{1}{n}\right) \boldsymbol{x}_{k-1}^\top \boldsymbol{x}_{k-1}.$$

By the law of total expectation,

$$\mathbb{E}\left[\boldsymbol{x}_T^\top \boldsymbol{x}_T \mid \boldsymbol{x}_0\right] = \mathbb{E}\left[\mathbb{E}\left[\boldsymbol{x}_T^\top \boldsymbol{x}_T \mid \boldsymbol{x}_{T-1}\right] \mid \boldsymbol{x}_0\right]$$

$$\geq \left(1 - \frac{1}{n}\right) \mathbb{E}\left[\boldsymbol{x}_{T-1}^\top \boldsymbol{x}_{T-1} \mid \boldsymbol{x}_0\right]$$

$$\vdots$$

$$\geq \left(1 - \frac{1}{n}\right)^T \|\boldsymbol{x}_0\|^2.$$

For the second part, we use the following lemma.

**Lemma B.1** (Nesterov (2012), Lemma 1). *If $\boldsymbol{X} \in \mathbb{R}^{n \times n}$ is a positive semi-definite matrix,*

$$\boldsymbol{X} \preceq n \cdot \mathrm{diag}(\boldsymbol{X}).$$

Now we define the following matrix operator:

$$\widehat{\mathcal{M}}_{\boldsymbol{A}}^{\mathrm{RCD}}(\boldsymbol{X}) := \left(\boldsymbol{I} - \frac{\boldsymbol{A}}{n}\right)^\top \boldsymbol{X} \left(\boldsymbol{I} - \frac{\boldsymbol{A}}{n}\right).$$

According to Lemma B.1, $\widehat{\mathcal{M}}_{\boldsymbol{A}}^{\text{RCD}}(\boldsymbol{X}) \preceq \mathcal{M}_{\boldsymbol{A}}^{\text{RCD}}(\boldsymbol{X})$ for any $\boldsymbol{X} \succeq 0$ because

$$\mathcal{M}_{\boldsymbol{A}}^{\text{RCD}}(\boldsymbol{X}) = \widehat{\mathcal{M}}_{\boldsymbol{A}}^{\text{RCD}}(\boldsymbol{X}) + \frac{\boldsymbol{A}^{\top}(n \cdot \text{diag}(\boldsymbol{X}) - \boldsymbol{X})\boldsymbol{A}}{n^2}.$$

Moreover, if $\boldsymbol{X} \preceq \boldsymbol{Y}$, then $\widehat{\mathcal{M}}_{\boldsymbol{A}}^{\text{RCD}}(\boldsymbol{X}) \preceq \widehat{\mathcal{M}}_{\boldsymbol{A}}^{\text{RCD}}(\boldsymbol{Y})$.

We now aim to show that for any positive integer $k$ and $\boldsymbol{X} \succeq 0$,

$$(\widehat{\mathcal{M}}_{\boldsymbol{A}}^{\text{RCD}})^k(\boldsymbol{X}) \preceq (\mathcal{M}_{\boldsymbol{A}}^{\text{RCD}})^k(\boldsymbol{X}). \tag{15}$$

We prove this by induction on $k$.

**Base Case:** For $k = 1$, we have already shown that $\widehat{\mathcal{M}}_{\boldsymbol{A}}^{\text{RCD}}(\boldsymbol{X}) \preceq \mathcal{M}_{\boldsymbol{A}}^{\text{RCD}}(\boldsymbol{X})$.

**Inductive Step:** Assume that for some positive integer $k$, we have

$$(\widehat{\mathcal{M}}_{\boldsymbol{A}}^{\text{RCD}})^k(\boldsymbol{X}) \preceq (\mathcal{M}_{\boldsymbol{A}}^{\text{RCD}})^k(\boldsymbol{X}).$$

We need to show that $(\widehat{\mathcal{M}}_{\boldsymbol{A}}^{\text{RCD}})^{k+1}(\boldsymbol{X}) \preceq (\mathcal{M}_{\boldsymbol{A}}^{\text{RCD}})^{k+1}(\boldsymbol{X})$. Since

$$
\begin{aligned}
(\widehat{\mathcal{M}}_{\boldsymbol{A}}^{\text{RCD}})^{k+1}(\boldsymbol{X}) &= \widehat{\mathcal{M}}_{\boldsymbol{A}}^{\text{RCD}}((\widehat{\mathcal{M}}_{\boldsymbol{A}}^{\text{RCD}})^k(\boldsymbol{X})) \\
&\preceq \widehat{\mathcal{M}}_{\boldsymbol{A}}^{\text{RCD}}((\mathcal{M}_{\boldsymbol{A}}^{\text{RCD}})^k(\boldsymbol{X})) && (\because (\widehat{\mathcal{M}}_{\boldsymbol{A}}^{\text{RCD}})^k(\boldsymbol{X}) \preceq (\mathcal{M}_{\boldsymbol{A}}^{\text{RCD}})^k(\boldsymbol{X})) \\
&\preceq \mathcal{M}_{\boldsymbol{A}}^{\text{RCD}}((\mathcal{M}_{\boldsymbol{A}}^{\text{RCD}})^k(\boldsymbol{X})) && (\because \widehat{\mathcal{M}}_{\boldsymbol{A}}^{\text{RCD}}(\boldsymbol{X}) \preceq \mathcal{M}_{\boldsymbol{A}}^{\text{RCD}}(\boldsymbol{X}), (\mathcal{M}_{\boldsymbol{A}}^{\text{RCD}})^k(\boldsymbol{X})) \succeq 0) \\
&= (\mathcal{M}_{\boldsymbol{A}}^{\text{RCD}})^{k+1}(\boldsymbol{X}),
\end{aligned}
$$

we have $(\widehat{\mathcal{M}}_{\boldsymbol{A}}^{\text{RCD}})^{k+1}(\boldsymbol{X}) \preceq (\mathcal{M}_{\boldsymbol{A}}^{\text{RCD}})^{k+1}(\boldsymbol{X})$ and Equation (15) holds for any positive integer $k$.

Therefore,

$$
\begin{aligned}
\mathbb{E}\left[\boldsymbol{x}_T^{\top}\boldsymbol{x}_T \mid \boldsymbol{x}_0\right] &= \boldsymbol{x}_0^{\top}(\mathcal{M}_{\boldsymbol{A}}^{\text{RCD}})^T(\boldsymbol{I})\boldsymbol{x}_0 \\
&\geq \boldsymbol{x}_0^{\top}(\widehat{\mathcal{M}}_{\boldsymbol{A}}^{\text{RCD}})^T(\boldsymbol{I})\boldsymbol{x}_0 \\
&= \boldsymbol{x}_0^{\top}\left(\boldsymbol{I} - \frac{\boldsymbol{A}}{n}\right)^{2T}\boldsymbol{x}_0.
\end{aligned}
$$

Let $\boldsymbol{v}_1, \boldsymbol{v}_2, \ldots, \boldsymbol{v}_n$ be the eigenvectors of $\boldsymbol{A}$ corresponding to the eigenvalues $\lambda_1 \leq \lambda_2 \leq \cdots \leq \lambda_n$, respectively. Since $\boldsymbol{A} \succ 0$, the eigenvalues are positive and the eigenvectors form an orthonormal basis for $\mathbb{R}^n$. Therefore, we can express $\boldsymbol{x}_0 \in \mathbb{R}^n$ as a linear combination, $\boldsymbol{x}_0 = c_1\boldsymbol{v}_1 + c_2\boldsymbol{v}_2 + \cdots + c_n\boldsymbol{v}_n$, where $c_i$ are scalar coefficients.

Excluding a measure zero set of vectors $\boldsymbol{x}$ such that $\boldsymbol{v}_1^{\top}\boldsymbol{x} = 0$, we may assume that $c_1 \neq 0$.

Now, let us define

$$d_i = \frac{c_i^2}{\sum_{k=1}^n c_k^2}, \quad \mu_i = 1 - \frac{\lambda_i}{n}.$$

Then, $d_1 \neq 0$, $d_i \geq 0$ for $i \in [n]$, $\sum_{i=1}^n d_i = 1$ and $\mu_1 \geq \mu_2 \geq \cdots \geq \mu_n > 0$. We can express $\frac{\boldsymbol{x}_0^{\top}\left(\boldsymbol{I}-\frac{\boldsymbol{A}}{n}\right)^{2T}\boldsymbol{x}_0}{\boldsymbol{x}_0^{\top}\boldsymbol{x}_0}$ in terms of $c_i$ and $\mu_i$:

$$\frac{\boldsymbol{x}_0^{\top}\left(\boldsymbol{I} - \frac{\boldsymbol{A}}{n}\right)^{2T}\boldsymbol{x}_0}{\boldsymbol{x}_0^{\top}\boldsymbol{x}_0} = \sum_{i=1}^n d_i\mu_i^{2T}.$$

Thus, we have

$$
\begin{aligned}
\left(\frac{\boldsymbol{x}_0^{\top}\left(\boldsymbol{I} - \frac{\boldsymbol{A}}{n}\right)^{2T}\boldsymbol{x}_0}{\boldsymbol{x}_0^{\top}\boldsymbol{x}_0}\right)^{1/T} &= \left(\sum_{i=1}^n d_i\mu_i^{2T}\right)^{1/T} \\
&= \mu_1^2\left(d_1 + d_2\left(\frac{\mu_2}{\mu_1}\right)^{2T} + \cdots + d_n\left(\frac{\mu_n}{\mu_1}\right)^{2T}\right)^{1/T}.
\end{aligned}
$$

Since $0 < \frac{\mu_i}{\mu_1} \le 1$, we have

$$d_1^{1/T} \le \left( d_1 + d_2 \left( \frac{\mu_2}{\mu_1} \right)^{2T} + \cdots + d_n \left( \frac{\mu_n}{\mu_1} \right)^{2T} \right)^{1/T} \le (d_1 + \cdots + d_n)^{1/T} = 1.$$

Thus,

$$\lim_{T \to \infty} \left( d_1 + d_2 \left( \frac{\mu_2}{\mu_1} \right)^{2T} + \cdots + d_n \left( \frac{\mu_n}{\mu_1} \right)^{2T} \right)^{1/T} = 1$$

because $d_1 > 0$. Therefore we can conclude that

$$\lim_{T \to \infty} \left( \frac{\mathbb{E}\left[ \boldsymbol{x}_T^\top \boldsymbol{x}_T \mid \boldsymbol{x}_0 \right]}{\boldsymbol{x}_0^\top \boldsymbol{x}_0} \right)^{1/T} \ge \mu_1^2 = \left( 1 - \frac{\sigma}{n} \right)^2$$

which finishes the proof. $\qquad \square$

### B.2. Proof of Theorem 3.3

Here we prove Theorem 3.3, restated below for the sake of readability.

**Theorem 3.3.** *For an initial point $\boldsymbol{x}_0 \in \mathbb{R}^n$, let $\boldsymbol{x}_K$ be the output of* RPCD *after $K$ epochs. If $\boldsymbol{A} \in \mathcal{A}_\sigma$ and $\boldsymbol{x}_0 \ne 0$,*

$$\lim_{K \to \infty} \left( \frac{\mathbb{E}\left[ \|\boldsymbol{x}_K\|^2 \right]}{\|\boldsymbol{x}_0\|^2} \right)^{\frac{1}{K}} \le \max\left\{ \left( 1 - \frac{1}{n} \right)^n, \left( 1 - \frac{\sigma}{n} \right)^{2n} \right\}.$$

*Proof.* We first assume that $\boldsymbol{A} = \sigma \boldsymbol{I}_n + (1 - \sigma) \mathbf{1}_n \mathbf{1}_n^\top$. Then, $\lambda_{\min}(\boldsymbol{A}) = \sigma$ and $\lambda_{\max}(\boldsymbol{A}) = n - (n-1)\sigma$.

Let $\boldsymbol{\Gamma} = \mathrm{tril}(\boldsymbol{A})$ and $\boldsymbol{C} = \boldsymbol{I} - \boldsymbol{\Gamma}^{-1}\boldsymbol{A}$. Since both $\boldsymbol{I}$ and $\mathbf{1}\mathbf{1}^\top$ are permutation-invariant, we have

$$\mathcal{M}_{\boldsymbol{A}}^{\mathrm{RPCD}}(\boldsymbol{I}) = \mathbb{E}[\boldsymbol{P}\boldsymbol{C}^\top \boldsymbol{C} \boldsymbol{P}^\top],$$
$$\mathcal{M}_{\boldsymbol{A}}^{\mathrm{RPCD}}(\mathbf{1}\mathbf{1}^\top) = \mathbb{E}[\boldsymbol{P}\boldsymbol{C}^\top \mathbf{1}\mathbf{1}^\top \boldsymbol{C} \boldsymbol{P}^\top].$$

We use the following lemma to compute $\mathcal{M}_{\boldsymbol{A}}^{\mathrm{RPCD}}(\boldsymbol{I})$ and $\mathcal{M}_{\boldsymbol{A}}^{\mathrm{RPCD}}(\mathbf{1}\mathbf{1}^\top)$.

**Lemma B.2** (Lee & Wright (2019), Lemma 3.1)**.** *Given any matrix $\boldsymbol{Q} \in \mathbb{R}^{n \times n}$ and permutation matrix $\boldsymbol{P}$ selected uniformly at random from the set of all permutations $\Pi$, we have $\mathbb{E}_{\boldsymbol{P}}[\boldsymbol{P}\boldsymbol{Q}\boldsymbol{P}^\top] = \tau_1 \boldsymbol{I} + \tau_2 \mathbf{1}\mathbf{1}^\top$, where*

$$\tau_2 = \frac{\mathbf{1}^\top \boldsymbol{Q} \mathbf{1} - \mathrm{tr}\,\boldsymbol{Q}}{n(n-1)}, \quad \tau_1 = \frac{\mathrm{tr}\,\boldsymbol{Q}}{n} - \tau_2.$$

As a consequence of this lemma, $\mathcal{M}_{\boldsymbol{A}}^{\mathrm{RPCD}}(\boldsymbol{I})$ and $\mathcal{M}_{\boldsymbol{A}}^{\mathrm{RPCD}}(\mathbf{1}\mathbf{1}^\top)$ can be expressed as

$$\tau_1^{(1)} \boldsymbol{I} + \tau_2^{(1)} \mathbf{1}\mathbf{1}^\top = \mathcal{M}_{\boldsymbol{A}}^{\mathrm{RPCD}}(\boldsymbol{I}),$$
$$\tau_1^{(2)} \boldsymbol{I} + \tau_2^{(2)} \mathbf{1}\mathbf{1}^\top = \mathcal{M}_{\boldsymbol{A}}^{\mathrm{RPCD}}(\mathbf{1}\mathbf{1}^\top),$$

where $\tau_1^{(1)}, \tau_2^{(1)}, \tau_1^{(2)}$ and $\tau_2^{(2)}$ can be computed using $\mathbf{1}^\top \boldsymbol{C}^\top \boldsymbol{C} \mathbf{1}$, $\mathrm{tr}(\boldsymbol{C}^\top \boldsymbol{C})$, $\mathbf{1}^\top \boldsymbol{C}^\top \mathbf{1}\mathbf{1}^\top \boldsymbol{C} \mathbf{1}$, and $\mathrm{tr}(\boldsymbol{C}^\top \mathbf{1}\mathbf{1}^\top \boldsymbol{C})$.

We remark that $\mathrm{span}\{\boldsymbol{I}, \mathbf{1}\mathbf{1}^\top\}$ is invariant under $\mathcal{M}_{\boldsymbol{A}}^{\mathrm{RPCD}}$ and the matrix representation of $\mathcal{M}_{\boldsymbol{A}}^{\mathrm{RPCD}}\big|_{\mathrm{span}\{\boldsymbol{I}, \mathbf{1}\mathbf{1}^\top\}}$ is $\boldsymbol{M}_{\boldsymbol{A}} := \begin{bmatrix} \tau_1^{(1)} & \tau_1^{(2)} \\ \tau_2^{(1)} & \tau_2^{(2)} \end{bmatrix}$. That is, if

$$\boldsymbol{M}_{\boldsymbol{A}} \begin{bmatrix} a \\ b \end{bmatrix} = \begin{bmatrix} a' \\ b' \end{bmatrix},$$

then $\mathcal{M}_A^{\mathrm{RPCD}}(a\boldsymbol{I} + b\boldsymbol{1}\boldsymbol{1}^\top) = a'\boldsymbol{I} + b'\boldsymbol{1}\boldsymbol{1}^\top$.

Let us denote $\boldsymbol{1}^\top \boldsymbol{C}^\top \boldsymbol{C} \boldsymbol{1}$, $\mathrm{tr}(\boldsymbol{C}^\top \boldsymbol{C})$, $\boldsymbol{1}^\top \boldsymbol{C}^\top \boldsymbol{1}\boldsymbol{1}^\top \boldsymbol{C}\boldsymbol{1}$, and $\mathrm{tr}(\boldsymbol{C}^\top \boldsymbol{1}\boldsymbol{1}^\top \boldsymbol{C})$ as $\alpha, \beta, \gamma$ and $\delta$, respectively. Then, by Lemma B.2, we have

$$\boldsymbol{M}_{\boldsymbol{A}} = \frac{1}{n(n-1)} \begin{bmatrix} n\beta - \alpha & n\delta - \gamma \\ \alpha - \beta & \gamma - \delta \end{bmatrix}.$$

To derive the explicit forms, we first calculate $\boldsymbol{C}$, which is given by $\boldsymbol{I} - \boldsymbol{\Gamma}^{-1}\boldsymbol{A}$. We can express $\boldsymbol{\Gamma}^{-1}$ as follows:

$$(\boldsymbol{\Gamma}^{-1})_{ij} = \begin{cases} 0 & \text{if } i < j \\ 1 & \text{if } i = j \\ -(1-\sigma)\sigma^{i-j-1} & \text{if } i > j. \end{cases}$$

Consequently, we have

$$\boldsymbol{C}_{ij} = \begin{cases} -(1-\sigma)\sigma^{i-1} & \text{if } i < j \\ (1-\sigma)(\sigma^{i-j} - \sigma^{i-1}) & \text{if } i \geq j. \end{cases} \tag{16}$$

To proceed, we introduce $\boldsymbol{v} = \boldsymbol{C}\boldsymbol{1}, \boldsymbol{w} = \boldsymbol{C}^\top \boldsymbol{1}$ and $L = \lambda_{\max}(\boldsymbol{A}) = n - (n-1)\sigma$. Then, by Equation (16), we have

$$\boldsymbol{v} = \boldsymbol{1} - L \begin{bmatrix} 1 \\ \sigma \\ \vdots \\ \sigma^{n-1} \end{bmatrix}, \quad \boldsymbol{w} = \begin{bmatrix} 0 \\ \sigma^n - \sigma^{n-1} \\ \vdots \\ \sigma^n - \sigma \end{bmatrix}.$$

With the explicit form of the entries of $\boldsymbol{C}$, we can now calculate the four quantities $\alpha, \beta, \gamma,$ and $\delta$, as follows (when $\sigma \neq 1$):

$$\begin{aligned}
\alpha &= \boldsymbol{v}^\top \boldsymbol{v} = n - 2L\frac{1-\sigma^n}{1-\sigma} + L^2\frac{1-\sigma^{2n}}{1-\sigma^2} \\
\beta &= \sum_{1 \leq i,j \leq n} \boldsymbol{C}_{ij}^2 = \sum_{j=1}^n \left( (1-\sigma)^2\frac{1-\sigma^{2n}}{1-\sigma^2} + (1-\sigma)^2(1-2\sigma^{j-1})\frac{1-\sigma^{2(n-j+1)}}{1-\sigma^2} \right) \\
&= \frac{1-\sigma}{1+\sigma}\left( 2n - n\sigma^{2n} - 2(1-\sigma^{n+1})\frac{1-\sigma^n}{1-\sigma} - \sigma^2\frac{1-\sigma^{2n}}{1-\sigma^2} \right) \\
\gamma &= \left(\boldsymbol{1}^\top \boldsymbol{v}\right)^2 = \left( 1 - \frac{1}{1-\sigma} + \left(n - 1 + \frac{1}{1-\sigma}\right)\sigma^n \right)^2 \\
\delta &= \boldsymbol{w}^\top \boldsymbol{w} = n\sigma^{2n} - 2\sigma^{n+1}\frac{1-\sigma^n}{1-\sigma} + \sigma^2\frac{1-\sigma^{2n}}{1-\sigma^2}.
\end{aligned} \tag{17}$$

Note that all of these values are polynomials in $\sigma$. (If $\sigma = 1$, all four quantities above are equal to zero.)

The explicit form of $\alpha, \beta, \gamma,$ and $\delta$ allows us to establish the following lemma.

**Lemma B.3.** $\tau_1^{(1)}, \tau_1^{(2)}, \tau_2^{(1)}$ and $\tau_2^{(2)}$ are all non-negative.

*Proof.* First, we show that $\tau_1^{(1)}, \tau_1^{(2)} \geq 0$ from the fact that $\mathcal{M}_A^{\mathrm{RPCD}}$ preserves positive semi-definiteness.

Recall that $\mathcal{M}_A^{\mathrm{RPCD}}(\boldsymbol{X}) = \mathbb{E}[\boldsymbol{T}_{A,p}^{\mathrm{RPCD}\top} \boldsymbol{X} \boldsymbol{T}_{A,p}^{\mathrm{RPCD}}]$. Since $\boldsymbol{X} \succeq 0$ implies $\boldsymbol{T}_{A,p}^{\mathrm{RPCD}\top} \boldsymbol{X} \boldsymbol{T}_{A,p}^{\mathrm{RPCD}} \succeq 0$ and the expectation of positive semi-definite matrices is also positive semi-definite, we can conclude that $\mathcal{M}_A^{\mathrm{RPCD}}(\boldsymbol{X}) \succeq 0$ whenever $\boldsymbol{X} \succeq 0$. As both $\boldsymbol{I} \succeq 0$ and $\boldsymbol{1}\boldsymbol{1}^\top \succeq 0$, it follows that $\mathcal{M}_A^{\mathrm{RPCD}}(\boldsymbol{I}) \succeq 0$ and $\mathcal{M}_A^{\mathrm{RPCD}}(\boldsymbol{1}\boldsymbol{1}^\top) \succeq 0$, i.e., their eigenvalues are non-negative. Since $\tau_1^{(1)}$ and $\tau_1^{(2)}$ are eigenvalues of $\mathcal{M}_A^{\mathrm{RPCD}}(\boldsymbol{I}) \succeq 0$ and $\mathcal{M}_A^{\mathrm{RPCD}}(\boldsymbol{1}\boldsymbol{1}^\top) \succeq 0$ respectively, we have $\tau_1^{(1)}, \tau_1^{(2)} \geq 0$.

Next, we show that $\tau_2^{(2)} \geq 0$. For $\boldsymbol{w} = \boldsymbol{C}^\top \mathbf{1}$, we have

$$\gamma = \mathbf{1}^\top \boldsymbol{w}\boldsymbol{w}^\top \mathbf{1} = (\mathbf{1}^\top \boldsymbol{w})^2$$
$$\delta = \text{tr}(\boldsymbol{w}\boldsymbol{w}^\top) = \text{tr}(\boldsymbol{w}^\top \boldsymbol{w}) = \|\boldsymbol{w}\|^2.$$

Since $\tau_2^{(2)} = \gamma - \delta$, we have

$$\tau_2^{(2)} = (\mathbf{1}^\top \boldsymbol{w})^2 - \|\boldsymbol{w}\|^2$$
$$= 2\sum_{i<j} w(i)w(j).$$

Since $w(i) = \sigma^n - \sigma^{n+1-i} \leq 0$ for all $i \in [n]$, we have $w(i)w(j) \geq 0$, and therefore $\tau_2^{(2)} = 2\sum_{i<j} w(i)w(j) \geq 0$.

Finally, to show that $\tau_2^{(1)} \geq 0$, we first show that the entries of $\boldsymbol{C}^\top \boldsymbol{C}$ is non-negative.

In particular, we can prove the following two inequalities:

- If $2 \leq i < j < n$, then $\boldsymbol{C}_i^\top \boldsymbol{C}_j \geq \boldsymbol{C}_i^\top \boldsymbol{C}_{j+1}$,

- If $2 < i < j \leq n$, then $\boldsymbol{C}_i^\top \boldsymbol{C}_j \geq \boldsymbol{C}_{i-1}^\top \boldsymbol{C}_j$,

where $\boldsymbol{C}_i$ denotes the $i$-th column of $\boldsymbol{C}$.

To show the first inequality, we begin by considering the difference $\boldsymbol{C}_j - \boldsymbol{C}_{j+1}$. We have

$$(\boldsymbol{C}_j - \boldsymbol{C}_{j+1})_k = \begin{cases} 0 & \text{if } k < j, \\ 1 - \sigma & \text{if } k = j, \\ (1-\sigma)(\sigma^{k-j} - \sigma^{k-j-1}) & \text{if } k > j. \end{cases} \tag{18}$$

Therefore, if $2 \leq i < j < n$,

$$\boldsymbol{C}_i^\top (\boldsymbol{C}_j - \boldsymbol{C}_{j+1}) = (1-\sigma)^2(\sigma^{j-i} - \sigma^{j-1}) + \sum_{k=j+1}^{n} (1-\sigma)^2(\sigma^{k-i} - \sigma^{k-1})(\sigma^{k-j} - \sigma^{k-j-1})$$

$$= (1-\sigma)^2(\sigma^{-i} - \sigma^{-1})\left(\sigma^j + (\sigma^{-j} - \sigma^{-j-1})\sum_{k=j+1}^{n} \sigma^{2k}\right).$$

Since $i \geq 2$ and $\sigma \in (0, 1]$, we have $(1-\sigma)^2 \geq 0$ and $(\sigma^{-i} - \sigma^{-1}) \geq 0$. Thus, it suffices to show that

$$\sigma^j + (\sigma^{-j} - \sigma^{-j-1})\sum_{k=j+1}^{n} \sigma^{2k} \geq 0.$$

Since

$$\sigma^j + (\sigma^{-j} - \sigma^{-j-1})\sum_{k=j+1}^{n} \sigma^{2k} = \sigma^j + (\sigma^{-j} - \sigma^{-j-1})\sigma^{2j+2} \cdot \frac{1 - \sigma^{2(n-j)}}{1 - \sigma^2}$$

$$= \sigma^j\left(1 - \frac{\sigma}{1+\sigma}(1 - \sigma^{2(n-j)})\right)$$

$$= \frac{\sigma^j}{1+\sigma}(1 + \sigma^{2n-2j+1}),$$

we have $\boldsymbol{C}_i^\top (\boldsymbol{C}_j - \boldsymbol{C}_{j+1}) \geq 0$.

For the second inequality, we will show that $(C_{i-1} - C_i)^\top C_j \leq 0$. Using Equation (18), we obtain

$$(C_{i-1} - C_i)^\top C_j = (1 - \sigma)^2 \left( -\sigma^{i-2} + (\sigma^{-i-1} - \sigma^{-i}) \sum_{k=i}^{j-1} \sigma^{2k} + (\sigma^{-i+1} - \sigma^{-i})(\sigma^{-j} - \sigma^{-1}) \sum_{k=j}^{n} \sigma^{2k} \right).$$

Since

$$(\sigma^{-i+1} - \sigma^{-i})(\sigma^{-j} - \sigma^{-1}) \sum_{k=j}^{n} \sigma^{2k} \leq 0,$$

it is sufficient to show that

$$-\sigma^{i-2} + (\sigma^{-i-1} - \sigma^{-i}) \sum_{k=i}^{j-1} \sigma^{2k} \leq 0.$$

Since

$$-\sigma^{i-2} + (\sigma^{-i-1} - \sigma^{-i}) \sum_{k=i}^{j-1} \sigma^{2k} = -\sigma^{i-2} + (\sigma^{-i-1} - \sigma^{-i}) \sigma^{2i} \cdot \frac{1 - \sigma^{2(j-i)}}{1 - \sigma^2}$$

$$\leq -\sigma^{i-2} + \frac{\sigma^{i-1} - \sigma^i}{1 - \sigma^2}$$

$$= -\frac{\sigma^{i-2}}{\sigma + 1}$$

$$\leq 0,$$

it follows that $(C_i - C_{i-1})^\top C_j \leq 0$.

Using a chain of the inequalities we showed, we can obtain $\min_{2 \leq i < j \leq n} C_i^\top C_j = C_2^\top C_n$. Moreover, since

$$C_2^\top C_n = (1 - \sigma)^2 \left( 1 - (\sigma^{-3} - \sigma^{-2}) \sum_{k=2}^{n-1} \sigma^{2k} + (1 - \sigma^{n-1})(\sigma^{n-2} - \sigma^{n-1}) \right)$$

and

$$1 - (\sigma^{-3} - \sigma^{-2}) \sum_{k=2}^{n-1} \sigma^{2k} + (1 - \sigma^{n-1})(\sigma^{n-2} - \sigma^{n-1}) \geq 1 - (\sigma^{-3} - \sigma^{-2}) \sum_{k=2}^{n-1} \sigma^{2k}$$

$$\geq 1 - (\sigma^{-3} - \sigma^{-2}) \frac{\sigma^4}{1 - \sigma^2}$$

$$\geq 0,$$

we have $C_2^\top C_n \geq 0$. Additionally, since $C_1 = 0$, we have $C_i^\top C_j = 0$ whenever $i = 1$ or $j = 1$. With the fact that $C_i^\top C_j \geq 0$ if $2 \leq i, j \leq n$, we have $\min_{1 \leq i, j \leq n} C_i^\top C_j = \min_{1 \leq i, j \leq n} (C^\top C)_{ij} \geq 0$, i.e., all entries of $C^\top C$ are non-negative. Therefore we can conclude that

$$\tau_2^{(1)} = \mathbf{1}^\top C^\top C \mathbf{1} - \text{tr}(C^\top C)$$

$$= \sum_{i \neq j} (C^\top C)_{ij}$$

$$\geq 0,$$

which completes the proof. □

We now shift our focus to the left-hand side of the inequality in the theorem. Since $(\mathcal{M}_{\boldsymbol{A}}^{\mathrm{RPCD}})^K(\boldsymbol{I})$ is symmetric, we have

$$\frac{\mathbb{E}\left[\|\boldsymbol{x}_K\|^2\right]}{\|\boldsymbol{x}_0\|^2} = \frac{\boldsymbol{x}_0^\top (\mathcal{M}_{\boldsymbol{A}}^{\mathrm{RPCD}})^K(\boldsymbol{I})\boldsymbol{x}_0}{\boldsymbol{x}_0^\top \boldsymbol{x}_0}$$
$$\leq \lambda_{\max}((\mathcal{M}_{\boldsymbol{A}}^{\mathrm{RPCD}})^K(\boldsymbol{I})).$$

Importantly, we have previously shown that $(\mathcal{M}_{\boldsymbol{A}}^{\mathrm{RPCD}})^K(\boldsymbol{I})$ can be computed using only $\boldsymbol{M_A}$. Specifically, if we write $(\mathcal{M}_{\boldsymbol{A}}^{\mathrm{RPCD}})^K(\boldsymbol{I}) = \alpha_K \boldsymbol{I} + \beta_K \boldsymbol{1}\boldsymbol{1}^\top$, then it follows that

$$\boldsymbol{M}_{\boldsymbol{A}}^K \begin{bmatrix} 1 \\ 0 \end{bmatrix} = \begin{bmatrix} \alpha_K \\ \beta_K \end{bmatrix}.$$

Moreover, since the entries of $\boldsymbol{M_A}$ are non-negative by Lemma B.3, we have $\alpha_K, \beta_K \geq 0$ for all $K \geq 0$. Since the eigenvalues of a matrix of the form $a\boldsymbol{I} + b\boldsymbol{1}\boldsymbol{1}^\top$ are $a$ and $a + nb$ (with multiplicity $n-1$), the largest eigenvalue of $(\mathcal{M}_{\boldsymbol{A}}^{\mathrm{RPCD}})^K(\boldsymbol{I}) = \alpha_K \boldsymbol{I} + \beta_K \boldsymbol{1}\boldsymbol{1}^\top$ is given by

$$\alpha_K + n\beta_K = \boldsymbol{y}^\top \boldsymbol{M}_{\boldsymbol{A}}^K \boldsymbol{x},$$

where $\boldsymbol{x} = \begin{bmatrix} 1 \\ 0 \end{bmatrix}$ and $\boldsymbol{y} = \begin{bmatrix} 1 \\ n \end{bmatrix}$.

To proceed, we use the following lemma, known as the Gelfand formula.

**Lemma B.4** (Horn & Johnson (2012), Corollary 5.6.14.). *If $\boldsymbol{A} \in \mathbb{R}^{n \times n}$, then $\rho(\boldsymbol{A}) = \lim_{K \to \infty} \|\boldsymbol{A}^K\|^{1/K}$.*

Since $\lambda_{\max}((\mathcal{M}_{\boldsymbol{A}}^{\mathrm{RPCD}})^K(\boldsymbol{I})) = \boldsymbol{y}^\top \boldsymbol{M}_{\boldsymbol{A}}^K \boldsymbol{x} \geq 0$, we have

$$\lim_{K \to \infty} \left(\lambda_{\max}((\mathcal{M}_{\boldsymbol{A}}^{\mathrm{RPCD}})^K(\boldsymbol{I}))\right)^{1/K} = \lim_{K \to \infty} |\boldsymbol{y}^\top \boldsymbol{M}_{\boldsymbol{A}}^K \boldsymbol{x}|^{1/K}$$
$$\leq \lim_{K \to \infty} (\|\boldsymbol{y}\|\|\boldsymbol{M}_{\boldsymbol{A}}^K\|\|\boldsymbol{x}\|)^{1/K}$$
$$\leq \lim_{K \to \infty} \|\boldsymbol{y}\|^{1/K}\|\boldsymbol{x}\|^{1/K}\|\boldsymbol{M}_{\boldsymbol{A}}^K\|^{1/K}$$
$$= \rho(\boldsymbol{M_A}). \qquad (\because \text{Lemma B.4})$$

Now, consider $\boldsymbol{A} = \mathrm{diag}\{\sigma \boldsymbol{I}_k + (1-\sigma)\boldsymbol{1}_k \boldsymbol{1}_k^\top, \boldsymbol{I}_{n-k}\}$ for an integer $k$ with $2 \leq k \leq n$. We denote the submatrix $\sigma \boldsymbol{I}_k + (1-\sigma)\boldsymbol{1}_k \boldsymbol{1}_k^\top$ by $\boldsymbol{A}_k$. Let $S_n$ be the set of permutations on $[n]$. We first show the following lemma:

**Lemma B.5.** *Let $\boldsymbol{P}$ be an $n \times n$ permutation matrix generated by a permutation $p \in S_n$. Let $q$ be a permutation of $[n]$ such that $q(l) = i_l$ for $l \in [k]$, where $i_1, \ldots, i_k$ is a reordering of $[k]$ satisfying $p^{-1}(i_k) > \cdots > p^{-1}(i_1)$, and $q(l) = l$ for $l > k$. Define $\boldsymbol{Q}$ as an $n \times n$ permutation matrix generated by $q$. Then,*

$$\boldsymbol{P}\boldsymbol{\Gamma}_{\boldsymbol{P}}\boldsymbol{P}^\top = \boldsymbol{Q}\boldsymbol{\Gamma}_{\boldsymbol{Q}}\boldsymbol{Q}^\top.$$

*Proof.* Since

$$\boldsymbol{e}_i^\top \mathrm{tril}(\boldsymbol{A})\boldsymbol{e}_j = \begin{cases} a_{ij} & \text{if } i \geq j \\ 0 & \text{if } i < j \end{cases}$$

and $\boldsymbol{P}^\top = \boldsymbol{P}^{-1}$ for any permutation matrix $\boldsymbol{P}$, we have

$$\boldsymbol{e}_i^\top \boldsymbol{P}\boldsymbol{\Gamma}_{\boldsymbol{P}}\boldsymbol{P}^\top \boldsymbol{e}_j = \boldsymbol{e}_{p^{-1}(i)}^\top \boldsymbol{\Gamma}_{\boldsymbol{P}}\boldsymbol{e}_{p^{-1}(j)}$$
$$= \begin{cases} \boldsymbol{e}_{p^{-1}(i)}^\top (\boldsymbol{P}^\top \boldsymbol{A}\boldsymbol{P})\boldsymbol{e}_{p^{-1}(j)} & \text{if } p^{-1}(i) \geq p^{-1}(j) \\ 0 & \text{if } i < j \end{cases}$$
$$= \begin{cases} a_{ij} & \text{if } p^{-1}(i) \geq p^{-1}(j) \\ 0 & \text{if } p^{-1}(i) < p^{-1}(j). \end{cases}$$

Consider first the case where $i, j \in [k]$. Then,

$$p^{-1}(i) \geq p^{-1}(j) \iff i = i_a, \ j = i_b \text{ for some } a, b \in [k] \text{ such that } a \geq b$$
$$\iff q^{-1}(i_a) = a \geq b = q^{-1}(i_b).$$

Therefore,

$$e_i^\top P \Gamma_P P^\top e_j = e_i^\top Q \Gamma_Q Q^\top e_j$$

for all $i, j \in [k]$.

Now consider the case where $i \in [k]$ and $j \notin [k]$. In this case, we have $a_{ij} = 0$, so the equality $e_i^\top P \Gamma_P P^\top e_j = e_i^\top Q \Gamma_Q Q^\top e_j$ holds trivially. Similarly, the equality holds when $i \notin [k]$ and $j \in [k]$.

Finally, suppose $i, j \notin [k]$. Then both $e_i^\top P \Gamma_P P^\top e_j$ and $e_i^\top Q \Gamma_Q Q^\top e_j$ are equal to 1 if $i = j$, and 0 otherwise.

Therefore, we conclude that

$$e_i^\top P \Gamma_P P^\top e_j = e_i^\top Q \Gamma_Q Q^\top e_j$$

for all $i, j \in [n]$. $\qquad\qquad\qquad\qquad\qquad\qquad\qquad\qquad\qquad\qquad\qquad\qquad\qquad\qquad\square$

Note that for any $p \in S_n$, there exists a unique permutation $q_k \in S_k$ satisfying $q(l) = i_l$ for $l \in [k]$, where $i_1, \ldots, i_k$ is a reordering of $[k]$ such that $p^{-1}(i_k) > \cdots > p^{-1}(i_1)$. Furthermore, there exist $\frac{n!}{k!}$ permutations in $S_n$ that correspond to a given $q_k$.

Let $\mathcal{D}_{k,n-k}$ be the set of block-diagonal matrices consisting of a $k \times k$ block and an $(n-k) \times (n-k)$ block. Then, $\mathcal{D}_{k,n-k}$ is closed under scalar multiplication, matrix multiplication, addition, transposition, and inversion.

Now, we have $T_{A,p}^{\mathrm{RPCD}} = I - P \Gamma_P^{-1} P^\top A = I - Q \Gamma_Q^{-1} Q^\top A$ by Lemma B.5. Also, $I, P, \Gamma_P$ and $A$ are in $\mathcal{D}_{k,n-k}$, so $T_{A,p}^{\mathrm{RPCD}} \in \mathcal{D}_{k,n-k}$. To analyze the structure, let $q_k$ be the restriction of $q$ to $[k]$ and $Q_k$ be the $k \times k$ permutation matrix generated by $q_k$, which is equal to the $(1,1)$-block of $Q$. Then, the $(1,1)$-block of $T_{A,p}^{\mathrm{RPCD}}$ is given by $I_k - Q_k \Gamma_{Q_k}^{-1} Q_k^\top A_k$ and the $(2,2)$-block is $\mathbf{0}_{n-k,n-k}$, meaning that

$$T_{A,p}^{\mathrm{RPCD}} = \begin{bmatrix} T_{A_k,q_k}^{\mathrm{RPCD}} & \mathbf{0}_{k,n-k} \\ \mathbf{0}_{n-k,k} & \mathbf{0}_{n-k,n-k} \end{bmatrix}.$$

Therefore, if $X = \begin{bmatrix} X_{11} & \mathbf{0}_{k,n-k} \\ \mathbf{0}_{n-k,k} & X_{22} \end{bmatrix} \in \mathcal{D}_{k,n-k}$, we have

$$\mathcal{M}_A^{\mathrm{RPCD}}(X) = \frac{1}{n!} \sum_{p \in S_n} (T_{A,p}^{\mathrm{RPCD}\top} X T_{A,p}^{\mathrm{RPCD}})$$

$$= \frac{1}{k!} \sum_{q_k \in S_k} \begin{bmatrix} T_{A_k,q_k}^{\mathrm{RPCD}\top} & \mathbf{0}_{k,n-k} \\ \mathbf{0}_{n-k,k} & \mathbf{0}_{n-k,n-k} \end{bmatrix} \begin{bmatrix} X_{11} & \mathbf{0}_{k,n-k} \\ \mathbf{0}_{n-k,k} & X_{22} \end{bmatrix} \begin{bmatrix} T_{A_k,q_k}^{\mathrm{RPCD}} & \mathbf{0}_{k,n-k} \\ \mathbf{0}_{n-k,k} & \mathbf{0}_{n-k,n-k} \end{bmatrix}$$

$$= \frac{1}{k!} \sum_{q_k \in S_k} \begin{bmatrix} T_{A_k,q_k}^{\mathrm{RPCD}\top} X_{11} T_{A_k,q_k}^{\mathrm{RPCD}} & \mathbf{0}_{k,n-k} \\ \mathbf{0}_{n-k,k} & \mathbf{0}_{n-k,n-k} \end{bmatrix}$$

$$= \begin{bmatrix} \mathcal{M}_{A_k}^{\mathrm{RPCD}}(X_{11}) & \mathbf{0}_{k,n-k} \\ \mathbf{0}_{n-k,k} & \mathbf{0}_{n-k,n-k} \end{bmatrix}.$$

By the above, for all $K \geq 1$, we can see that

$$(\mathcal{M}_A^{\mathrm{RPCD}})^K(I_n) = \begin{bmatrix} (\mathcal{M}_{A_k}^{\mathrm{RPCD}})^K(I_k) & \mathbf{0}_{k,n-k} \\ \mathbf{0}_{n-k,k} & \mathbf{0}_{n-k,n-k} \end{bmatrix}$$

and

$$\lambda_{\max}((\mathcal{M}_A^{\mathrm{RPCD}})^K(I_n)) = \lambda_{\max}((\mathcal{M}_{A_k}^{\mathrm{RPCD}})^K(I_k))$$

because $(\mathcal{M}_{\boldsymbol{A}_k}^{\mathrm{RPCD}})^K(\boldsymbol{I}_k) \succeq 0$.

Since we have already shown that $\lim_{K \to \infty} \lambda_{\max}((\mathcal{M}_{\boldsymbol{A}_k}^{\mathrm{RPCD}})^K(\boldsymbol{I}_k))^{1/K} \leq \rho(\boldsymbol{M}_{\boldsymbol{A}_k})$, it suffices to show that

$$\rho(\boldsymbol{M}_{\boldsymbol{A}_k}) \leq \max\left\{\left(1 - \frac{1}{n}\right)^n, \left(1 - \frac{\sigma}{n}\right)^{2n}\right\}.$$

To proceed, we can use the fact that $\rho(\boldsymbol{M}_{\boldsymbol{A}_k}) \leq \|\boldsymbol{M}_{\boldsymbol{A}_k}\|_\infty$. Since $\|\boldsymbol{M}_{\boldsymbol{A}_k}\|_\infty = \max\{\tau_1^{(1)} + \tau_1^{(2)}, \tau_2^{(1)} + \tau_2^{(2)}\}$ (all being positive values by Lemma B.3), it is sufficient to show that the inequalities

$$\tau_1^{(1)}(k, \sigma) + \tau_1^{(2)}(k, \sigma) \leq \max\left\{\left(1 - \frac{1}{n}\right)^n, \left(1 - \frac{\sigma}{n}\right)^{2n}\right\} \tag{19}$$

$$\tau_2^{(1)}(k, \sigma) + \tau_2^{(2)}(k, \sigma) \leq \max\left\{\left(1 - \frac{1}{n}\right)^n, \left(1 - \frac{\sigma}{n}\right)^{2n}\right\} \tag{20}$$

hold for all integers $k$ such that $2 \leq k \leq n$, where $\tau_i^{(j)}(k, \sigma)$ denotes $(\boldsymbol{M}_{\boldsymbol{A}_k})_{ij}$ for $i = 1, 2$ and $j = 1, 2$. Since both left-hand sides in (19) and (20) are polynomials in $\sigma$, the problem reduces to a comparison of polynomials.

We now focus on finding the coefficient series of $\alpha, \beta, \gamma$, and $\delta$ when each is written as a polynomial of $\sigma$. This is because the left-hand sides of the inequalities can be expressed in terms of these polynomials. We introduce some notations for brevity. Let $s_n = \sum_{i=0}^{n-1} \sigma^i$ and $t_n = \sum_{i=0}^{n-1} \sigma^{2i}$. Then, for $\sigma \in (0, 1)$, we have

$$s_n = \frac{1 - \sigma^n}{1 - \sigma}, \quad t_n = \frac{1 - \sigma^{2n}}{1 - \sigma^2}.$$

Since $\alpha = \beta = \gamma = \delta = 0$ when $\sigma = 1$, we will focus on calculating the coefficients of $\alpha, \beta, \gamma$ and $\delta$ for $\sigma \in (0, 1)$.

COEFFICIENTS OF $\alpha$

We have

$$Ls_n = n + \sum_{i=1}^{n-1} \sigma^i - (n-1)\sigma^n$$

$$L^2 t_n = n^2 - 2n(n-1)\sum_{i=0}^{n-1} \sigma^{2i+1} + (n^2 + (n-1)^2)\sum_{i=0}^{n-1} \sigma^{2i+2} + (n-1)^2\sigma^{2n}.$$

Since $\alpha = n - 2Ls_n + L^2 t_n$, if we write $\alpha = \sum_{k=0}^{2n} a_k \sigma^k$,

$$a_k = \begin{cases} n^2 - n & \text{if } k = 0 \\ -2n^2 + 2n - 2 & \text{if } 1 \leq k \leq n-1 \text{ and } k \text{ odd} \\ 2n^2 - 2n - 1 & \text{if } 1 \leq k \leq n-1 \text{ and } k \text{ even} \\ -2(n-1)^2 & \text{if } k = n \text{ and } n \text{ odd} \\ 2n^2 - 1 & \text{if } k = n \text{ and } n \text{ even} \\ -2n(n-1) & \text{if } n+1 \leq k \leq 2n-1 \text{ and } k \text{ odd} \\ 2n^2 - 2n + 1 & \text{if } n+1 \leq k \leq 2n-1 \text{ and } k \text{ even} \\ (n-1)^2 & \text{if } k = 2n. \end{cases}$$

COEFFICIENTS OF $\beta$

We first expand

$$(1-\sigma)^2 \sum_{j=1}^{n}\left((1 - 2\sigma^{j-1})\sum_{i=0}^{n-j}\sigma^{2i}\right) = (1-\sigma)^2 \sum_{j=1}^{n}\sum_{i=0}^{n-j}\sigma^{2i} - (1-\sigma)^2 \sum_{j=1}^{n}\sum_{i=0}^{n-j}2\sigma^{j-1}\sigma^{2i}.$$

Using the fact that

$$\sum_{j=1}^{n}\sum_{i=0}^{n-j}\sigma^{2i} = \sum_{k=0}^{n-1}(n-k)\sigma^{2k},$$

we can simplify the first term:

$$(1-\sigma)^2 \sum_{j=1}^{n}\sum_{i=0}^{n-j}\sigma^{2i} = n + \sum_{k=1}^{2n}(-1)^k(2n+1-k)\sigma^k$$

For the second term, we have

$$(1-\sigma)^2 \sum_{j=1}^{n}\sum_{i=0}^{n-j}\sigma^{j-1}\sigma^{2i} = -\sigma^n + \sum_{k=0}^{2n}(-\sigma)^k.$$

Now, the remaining part is

$$\sum_{j=0}^{n-1}(1-\sigma)^2 \frac{1-\sigma^{2n}}{1-\sigma^2} = n(1-\sigma)^2\left(1+\sigma^2+\cdots+\sigma^{2n-2}\right)$$

$$= n + n\sigma^{2n} + \sum_{k=1}^{2n-1}2n(-\sigma)^k.$$

Therefore, $\beta = \sum_{k=0}^{2n}b_k\sigma^k$, where

$$b_k = \begin{cases} 2n-2 & \text{if } k=0 \\ -3n+3 & \text{if } k=n \text{ and } n \text{ odd} \\ 3n+1 & \text{if } k=n \text{ and } n \text{ even} \\ n-1 & \text{if } k=2n \\ (-1)^k(4n-1-k) & \text{otherwise.} \end{cases}$$

COEFFICIENTS OF $\gamma$

We use the fact that $\gamma = \left(\mathbf{1}^\top \mathbf{v}\right)^2 = (n - Ls_n)^2$. Since $n - Ls_n = (n-1)\sigma^n - \sum_{i=1}^{n-1}\sigma^i$, we have

$$\gamma = (n - Ls_n)^2$$

$$= \left((n-1)\sigma^n - \sum_{i=1}^{n-1}\sigma^i\right)^2$$

$$= (n-1)^2\sigma^{2n} - 2(n-1)\sigma^n\sum_{i=1}^{n-1}\sigma^i + \sum_{j=2}^{2n-2}(j-1)\sigma^j.$$

Consequently, $\gamma = \sum_{k=0}^{2n}c_k\sigma^k$, where

$$c_k = \begin{cases} 0 & \text{if } k=0 \\ k-1 & \text{if } 1 \le k \le n \\ 1-k & \text{if } n+1 \le k \le 2n-1 \\ (n-1)^2 & \text{if } k=2n. \end{cases}$$

COEFFICIENTS OF $\delta$

Let $\delta = \sum_{k=0}^{2n} d_k \sigma^k$. Since

$$\delta = \sum_{i=1}^{n} (\sigma^n - \sigma^i)^2$$
$$= \sum_{i=1}^{n} (\sigma^{2n} - 2\sigma^{n+i} + \sigma^{2i})^2$$
$$= n\sigma^{2n} - 2(\sigma^{n+1} + \cdots + \sigma^{2n}) + (\sigma^2 + \cdots + \sigma^{2n}),$$

$$d_k = \begin{cases} 0 & \text{if } k = 0 \\ 0 & \text{if } 1 \leq k \leq n \text{ and } k \text{ odd} \\ 1 & \text{if } 1 \leq k \leq n \text{ and } k \text{ even} \\ -2 & \text{if } n+1 \leq k \leq 2n-1 \text{ and } k \text{ odd} \\ -1 & \text{if } n+1 \leq k \leq 2n-1 \text{ and } k \text{ even} \\ n-1 & \text{if } k = 2n. \end{cases}$$

We can now express the left-hand sides of inequalities (19) and (20) using the coefficients of $\alpha, \beta, \gamma$ and $\delta$. Let us define

$$T_1(n, \sigma) = n(\beta + \delta) - (\alpha + \gamma) = \sum_{k=0}^{2n} t_{1,k}\sigma^k$$

$$T_2(n, \sigma) = (\alpha + \gamma) - (\beta + \delta) = \sum_{k=0}^{2n} t_{2,k}\sigma^k.$$

Then, $T_1(n, \sigma) = n(n-1)(\tau_1^{(1)} + \tau_1^{(2)})$, $T_2(n, \sigma) = n(n-1)(\tau_2^{(1)} + \tau_2^{(2)})$,

$$t_{1,k} = \begin{cases} n^2 - n & \text{if } k = 0 \\ (n-1)k - 2n^2 - n + 3 & \text{if } 1 \leq k \leq n \text{ and } k \text{ odd} \\ -(n+1)k + 2n^2 + 2n + 2 & \text{if } 1 \leq k \leq n \text{ and } k \text{ even} \\ (n+1)k - 2n^2 - 3n - 1 & \text{if } n+1 \leq k \leq 2n \text{ and } k \text{ odd} \\ -(n-1)k + 2n^2 - 2 & \text{if } n+1 \leq k \leq 2n \text{ and } k \text{ even} \end{cases}$$

and

$$t_{2,k} = \begin{cases} n^2 - 3n + 2 & \text{if } k = 0 \\ -2n^2 + 6n - 4 & \text{if } 1 \leq k \leq n-1 \text{ and } k \text{ odd} \\ 2k + 2n^2 - 6n - 2 & \text{if } 1 \leq k \leq n-1 \text{ and } k \text{ even} \\ -2n^2 + 8n - 6 & \text{if } k = n \text{ and } n \text{ odd} \\ 2n^2 - 2n - 4 & \text{if } k = n \text{ and } n \text{ even} \\ -2k - 2n^2 + 6n + 2 & \text{if } n+1 \leq k \leq 2n \text{ and } k \text{ odd} \\ 2n^2 - 6n + 4 & \text{if } n+1 \leq k \leq 2n \text{ and } k \text{ even.} \end{cases}$$

We will first address the case where $n$ is sufficiently large. For the remaining cases, including inequalities (19) and (20) for smaller values of $n$ as well as specific polynomial inequalities arising during the proof for sufficiently large $n$, we will utilize Sturm's theorem (Proposition B.6), which provides a method for counting the number of distinct real roots of a polynomial within a given interval.

**Proposition B.6** (Jacobson (1985), Sturm's Theorem). *Let $f(x)$ be any polynomial with coefficients in $\mathbb{R}$ of positive degree. We define the **standard sequence** for $f(x)$ by:*

$$f_0(x) = f(x), \quad f_1(x) = f'(x) \quad \textit{(formal derivative of } f(x))$$
$$f_0(x) = q_1(x)f_1(x) - f_2(x), \quad \deg f_2 < \deg f_1$$
$$\vdots$$
$$f_{i-1}(x) = q_i(x)f_i(x) - f_{i+1}(x), \quad \deg f_{i+1} < \deg f_i$$
$$\vdots$$
$$f_s(x) = q_s(x)f_s(x) \quad \textit{(that is, } f_{s+1}(x) = 0).$$

*Given a sequence $c = (c_1, c_2, \ldots, c_m)$ of elements of $\mathbb{R}$, we define the **number of variations in sign** of c to be the number of variations in sign of the subsequence $c'$ obtained by dropping the 0's in c.*

*Let $f(x)$ be a polynomial of positive degree with coefficients in a real closed field $\mathbb{R}$ and let $\{f_0(x) = f(x), f_1(x) = f'(x), \ldots, f_s(x)\}$ be the standard sequence for $f(x)$. Assume $[a, b]$ is an interval such that $f(a) \neq 0$, $f(b) \neq 0$. Then the number of distinct roots of $f(x)$ in $(a, b)$ is $V_a - V_b$, where $V_c$ denotes the number of variations in sign of $\{f_0(c), f_1(c), \ldots, f_s(c)\}$.*

To apply Proposition B.6 to inequalities, we utilize the following property: for a polynomial $p$, if $p(x) \neq 0$ on $(a, b)$ and $p(a) > 0$, then $p(b) \geq 0$; similarly, if $p(b) > 0$, then $p(a) \geq 0$. By this property, we can verify that $p \geq 0$ on $(a, b], [a, b)$, or $[a, b]$. That is, since $p(x) \neq 0$ can be confirmed using Proposition B.6, we only need to check that $p$ is positive at the specific endpoint of the interval. Once this is established, we can conclude that $p \geq 0$ on the interval $(a, b], [a, b)$, or $[a, b]$, as required.

Prior to the main proof, we establish the following results using Proposition B.6, which address not only the remaining cases for smaller values of $n$ but also certain inequalities that arise during the proof.

- For $i = 1, 2$, $\frac{T_i(m,\sigma)}{m(m-1)} \leq \left(1 - \frac{\sigma}{n}\right)^{2n}$ for all integers $m$ and $n$ where $2 \leq m \leq n \leq 6$ and $\sigma \in (0, 0.6]$.

- $\frac{T_1(m,\sigma)}{m(m-1)} \leq \frac{1}{4}$ for all integers $m$ where $2 \leq m \leq 10$ and $\sigma \in [0.6, 0.8]$.

- $\frac{T_1(m,\sigma)}{m(m-1)} \leq \frac{1}{4}$ for all integers $m$ where $2 \leq m \leq 14$ and $\sigma \in [0.8, 1)$.

- $\frac{T_2(m,\sigma)}{m(m-1)} \leq \frac{1}{4}$ for all integers $m$ where $2 \leq m \leq 10$ and $\sigma \in [0.6, 1)$.

We verify these results in Appendix C using Proposition B.6 implemented in a computer algebra system. See Appendix C for more details on the implementations of the computational verification system.

Back to the main proof, to prove (19) and (20) on the interval $(0, 1]$ for all integers $k$ and $n$ such that $2 \leq k \leq n$, we first divide the interval into three subintervals: $(0, 0.6], [0.6, 0.8]$ and $[0.8, 1]$. We then show these inequalities for each subinterval. In the subsequent proof, for the cases when $\sigma \in (0, 0.6]$ and when $\sigma \in [0.6, 0.8]$, we initially assume that $n \geq 7$ and $n \geq 11$, respectively. The results of our proof can be summarized as follows:

- $\sigma \in (0, 0.6]$: Both inequalities (19) and (20) hold for all integers $k$ and $n$ such that $n \geq 7$ and $2 \leq k \leq n$.

- $\sigma \in [0.6, 0.8]$: Both inequalities (19) and (20) hold for all integers $k$ and $n$ such that $n \geq 11$ and $2 \leq k \leq n$.

- $\sigma \in [0.8, 1]$: If $\sigma \neq 1$, inequality (19) holds for all integers $k$ and $n$ such that $n \geq 15$ and $2 \leq k \leq n$, and (20) holds for all integers $k$ and $n$ such that $n \geq 10$ and $2 \leq k \leq n$. If $\sigma = 1$, since the left-hand side of both (19) and (20) is 0, thus they are trivially satisfied.

CASE I. $\sigma \in (0, 0.6]$

We first show that if $2 \leq i \leq n$, then $t_{1,2i} \geq 0$, $t_{1,2i-1} \leq 0$ and $t_{1,2i-1} + t_{1,2i} \leq 0$. If $4 \leq 2i \leq n$,

$$
\begin{aligned}
t_{1,2i} &= -(n+1)2i + 2n^2 + 2n + 2 \\
&\geq -(n+1)n + 2n^2 + 2n + 2 \\
&= n^2 + n + 2 \\
&\geq 0
\end{aligned}
$$

and

$$
\begin{aligned}
t_{1,2i-1} &= (n-1)(2i-1) - 2n^2 - n + 3 \\
&\leq (n-1)^2 - 2n^2 - n + 3 \\
&= -n^2 - 3n - 5 \\
&\leq 0.
\end{aligned}
$$

Moreover, we have $t_{1,2i-1} + t_{1,2i} = -4i + 6 \leq 0$ because $i \geq 2$.

If $2i = n + 1$,

$$
\begin{aligned}
t_{1,2i} &= -(n-1)(n+1) + 2n^2 - 2 \\
&= n^2 - 1 \\
&\geq 0
\end{aligned}
$$

and

$$
\begin{aligned}
t_{1,2i-1} &= (n-1)n - 2n^2 - n + 3 \\
&= -n^2 - 2n + 3 \\
&\leq 0.
\end{aligned}
$$

Furthermore, we have $t_{1,2i-1} + t_{1,2i} = -2n + 2 \leq 0$.

Finally, if $n < 2i - 1 \leq 2n - 1$,

$$
\begin{aligned}
t_{1,2i} &= -(n-1)2i + 2n^2 - 2 \\
&\geq -2n(n-1) + 2n^2 - 2 \\
&= 2n - 2 \\
&\geq 0
\end{aligned}
$$

and

$$
\begin{aligned}
t_{1,2i-1} &= (n+1)(2i-1) - 2n^2 - 3n - 1 \\
&\leq (2n-1)(n+1) - 2n^2 - 3n - 1 \\
&= -2n - 2 \\
&\leq 0.
\end{aligned}
$$

In this case, we have $t_{1,2i-1} + t_{1,2i} = 4i - 4n - 4 \leq 0$ because $2i \leq 2n$.

Now, if $\sigma \in (0, 1]$,

$$
\begin{aligned}
t_{1,2i}\sigma^{2i} + t_{1,2i-1}\sigma^{2i-1} &= \sigma^{2i-1}(t_{1,2i}\sigma + t_{1,2i-1}) \\
&\leq (t_{1,2i}\sigma + t_{1,2i-1}) \\
&\leq t_{1,2i} + t_{1,2i-1} \qquad (\because t_{1,2i} \geq 0) \\
&\leq 0.
\end{aligned}
$$

Thus, for $2 \leq i \leq n$, we have

$$\sum_{k=0}^{2i} t_{1,k}\sigma^k \geq T_1(n,\sigma). \tag{21}$$

This provides an upper bound for $T_1(n,\sigma)$.

A similar result can be obtained for $T_2(n,\sigma)$. We first show that $t_{2,2i-1} \leq 0$, $t_{2,2i} \geq 0$ for all $i \in [n]$.

Since $2n^2 - 6n + 4, 2n^2 - 2n - 4, 2n^2 - 8n + 6 \geq 0$, we only consider when $k$ is even and $2 \leq k \leq n-1$, or $k$ is odd and $n+1 \leq k \leq 2n$. For the first case,

$$\begin{aligned}
t_{2,k} &= 2k + 2n^2 - 6n - 2 \\
&\geq 2n^2 - 6n + 2 \\
&\geq 0
\end{aligned}$$

and for the second case,

$$\begin{aligned}
t_{2,k} &= -2k - 2n^2 + 6n + 2 \\
&\leq -2(n+1) - 2n^2 + 6n + 2 \\
&= -2n^2 + 4n \\
&\leq 0.
\end{aligned}$$

Now, we show that $t_{2,2i} + t_{2,2i-1} \leq 0$ if $2i - 1 \geq n + 1$. We have

$$\begin{aligned}
t_{2,2i} &= 2n^2 - 6n + 4 \\
&\geq 0
\end{aligned}$$

and

$$\begin{aligned}
t_{2,2i-1} &= -2(2i-1) - 2n^2 + 6n + 2 \\
&\leq -2(n+1) - 2n^2 + 6n + 2 \\
&= -2n^2 + 4n \\
&\leq 0.
\end{aligned}$$

Moreover, $t_{2,2i} + t_{2,2i-1} = -4i + 8 \leq 0$ because $2i - 1 \geq n + 1$.

Using the fact that $t_{2,2i-1} \leq 0$, $t_{2,2i} \geq 0$ for all $i \in [n]$, we have

$$\begin{aligned}
t_{2,2i}\sigma^{2i} + t_{2,2i-1}\sigma^{2i-1} &= \sigma^{2i-1}(t_{2,2i}\sigma + t_{2,2i-1}) \\
&\leq 0
\end{aligned}$$

if $\sigma \in \left[0, -\frac{t_{2,2i-1}}{t_{2,2i}}\right]$. Thus, if

$$\sigma \in \bigcap_{i=1}^{n} \left[0, -\frac{t_{2,2i-1}}{t_{2,2i}}\right] = \left[0, \min_{1 \leq i \leq n} -\frac{t_{2,2i-1}}{t_{2,2i}}\right],$$

we have

$$\sum_{k=0}^{2i-2} t_{2,k}\sigma^k \geq T_2(n,\sigma)$$

for $i \geq 1$.

Now, we focus on the minimum of $-\frac{t_{2,2i-1}}{t_{2,2i}}$. If $n$ is even,

$$-\frac{t_{2,2i-1}}{t_{2,2i}} = \begin{cases} \frac{2n^2-6n+4}{2n^2-6n-2+4i} & \text{if } 1 \le 2i \le n-1 \\ \frac{2n^2-6n+4}{2n^2-2n-4} & \text{if } 2i = n \\ \frac{2n^2-6n-4+4i}{2n^2-6n+4} & \text{if } n+1 \le 2i \le 2n. \end{cases}$$

Therefore, if $n$ is even and $n \ge 4$,

$$\min_{1 \le i \le n} -\frac{t_{2,2i-1}}{t_{2,2i}} = \min\left\{ \frac{2n^2-6n+4}{2n^2-4n+4}, \frac{2n^2-6n+4}{2n^2-2n-4}, \frac{2n^2-4n-2}{2n^2-6n+4} \right\}$$

$$= \frac{n-1}{n+1}.$$

If $n$ is odd and $2i-1 \le n$,

$$-\frac{t_{2,2i-1}}{t_{2,2i}} = \begin{cases} \frac{2n^2-6n+4}{2n^2-6n-2+4i} & \text{if } 1 \le 2i-1 \le n-1 \\ \frac{2n^2-8n+6}{2n^2-6n+4} & \text{if } 2i-1 = n \\ \frac{2n^2-6n-4+4i}{2n^2-6n+4} & \text{if } n+1 \le 2i-1 \le 2n. \end{cases}$$

Therefore, if $n$ is odd and $n \ge 3$,

$$\min_{1 \le i \le n} -\frac{t_{2,2i-1}}{t_{2,2i}} = \min\left\{ \frac{2n^2-8n+6}{2n^2-6n+4}, \frac{2n^2-6n+4}{2n^2-4n-2}, \frac{2n^2-4n}{2n^2-6n+4} \right\}$$

$$= \frac{n-3}{n-2}.$$

Thus,

$$\min_{1 \le i \le n} -\frac{t_{2,2i-1}}{t_{2,2i}} = \min\left\{ \frac{n-1}{n+1}, \frac{n-3}{n-2} \right\}. \tag{22}$$

Since $0.6 \le \frac{n-1}{n+1} \le \frac{n-3}{n-2}$ for $n \ge 5$, for $\sigma \in (0, 0.6]$ and $i \in [n]$, we have

$$\sum_{k=0}^{2i-2} t_{2,k}\sigma^k \ge T_2(n, \sigma). \tag{23}$$

Our next step is to find a lower bound for $\left(1 - \frac{\sigma}{n}\right)^{2n}$. To this end, we first prove the following lemma about binomial coefficients.

**Lemma B.7.** *For positive integers $r$ and $n$ such that $1 \le r < n$,*

$$\binom{2n}{r}\frac{1}{n^r} \ge \binom{2n}{r+1}\frac{1}{n^{r+1}}.$$

*Proof.*

$$\frac{\binom{2n}{r+1}\frac{1}{n^{r+1}}}{\binom{2n}{r}\frac{1}{n^r}} = \frac{2n-r}{n(r+1)}$$

and

$$\frac{2n-r}{n(r+1)} \le 1 \iff n \le (n+1)r.$$

Since $n \le (n+1)r$ for all $r \ge 1$ and both $\binom{2n}{r}\frac{1}{n^r}$ and $\binom{2n}{r+1}\frac{1}{n^{r+1}}$ are positive, we conclude that

$$\binom{2n}{r}\frac{1}{n^r} \ge \binom{2n}{r+1}\frac{1}{n^{r+1}}.$$

$\square$

By the Binomial Theorem,

$$\left(1 - \frac{\sigma}{n}\right)^{2n} = \sum_{r=0}^{2n} \binom{2n}{r} \left(-\frac{1}{n}\right)^r \sigma^r.$$

Additionally, for $i \leq n - 1$, we have

$$
\begin{aligned}
\sum_{r=2i}^{2n} \binom{2n}{r} \left(-\frac{1}{n}\right)^r \sigma^r &= \frac{1}{n^{2n}}\sigma^{2n} + \sum_{k=i}^{n-1}\left(\binom{2n}{2k}\left(-\frac{1}{n}\right)^{2k}\sigma^{2k} + \binom{2n}{2k+1}\left(-\frac{1}{n}\right)^{2k+1}\sigma^{2k+1}\right) \\
&\geq \sum_{k=i}^{n-1}\sigma^{2k}\left(\binom{2n}{2k}\left(-\frac{1}{n}\right)^{2k} + \binom{2n}{2k+1}\left(-\frac{1}{n}\right)^{2k+1}\sigma\right) && \left(\because \frac{1}{n^{2n}}\sigma^{2n} \geq 0\right) \\
&= \sum_{k=i}^{n-1}\sigma^{2k}\left(\binom{2n}{2k}\frac{1}{n^{2k}} - \binom{2n}{2k+1}\frac{1}{n^{2k+1}}\sigma\right) \\
&\geq \sum_{k=i}^{n-1}\sigma^{2k}\left(\binom{2n}{2k}\frac{1}{n^{2k}} - \binom{2n}{2k+1}\frac{1}{n^{2k+1}}\right) && (\because 0 \leq \sigma \leq 1) \\
&\geq 0. && (\because \text{Lemma B.7})
\end{aligned}
$$

Therefore, we obtain

$$\sum_{r=0}^{2i-1} \binom{2n}{r} \left(-\frac{1}{n}\right)^r \sigma^r \leq \left(1 - \frac{\sigma}{n}\right)^{2n}, \tag{24}$$

which provides a lower bound for $\left(1 - \frac{\sigma}{n}\right)^{2n}$.

Let $P_1(n, \sigma) = \sum_{k=0}^{6} t_{1,k}\sigma^k$. By Equation (21), $T_1(n, \sigma) \leq P_1(n, \sigma)$. Moreover, if $m$ is an integer with $2 \leq m \leq n$,

$$\frac{P_1(m+1, \sigma)}{(m+1)m} - \frac{P_1(m, \sigma)}{m(m-1)} = \frac{2}{m(m^2-1)}\left(m(\sigma^6 - \sigma^5 - \sigma^2 + \sigma) + 5\sigma^6 + \sigma^5 + 2\sigma^4 - \sigma^2 - \sigma\right). \tag{25}$$

Let $p_1(\sigma) = \sigma^6 - \sigma^5 - \sigma^2 + \sigma$ and $q_1(\sigma) = 5\sigma^6 + \sigma^5 + 2\sigma^4 - \sigma^2 - \sigma$.

It can be easily verified that $6p_1 + q_1 \geq 0$ holds for $\sigma \in (0, 0.6]$. Also, $p_1(\sigma) = \sigma(\sigma-1)^2(\sigma+1)(\sigma^2+1) \geq 0$ if $\sigma \geq 0$, so $mp_1 + q_1 \geq 0$ for all $m \geq 6$ and $\sigma \in (0, 0.6]$. This implies that

$$\frac{P_1(m+1, \sigma)}{(m+1)m} \geq \frac{P_1(m, \sigma)}{m(m-1)},$$

or equivalently, $\frac{P_1(m,\sigma)}{m(m-1)}$ is increasing in $m$, for all $m \geq 6$.

Now, define $D_1(n, \sigma) = \left( \sum_{r=0}^{5} \binom{2n}{r} \left( -\frac{\sigma}{n} \right)^r \right) - \frac{P_1}{n(n-1)} = \sum_{k=0}^{6} d_{1,k} \sigma^k$. Then,

$$d_{1,0} = 0$$

$$d_{1,1} = \frac{2}{n}$$

$$d_{1,2} = \frac{-3n + 1}{n(n-1)}$$

$$d_{1,3} = \frac{2n^2 + 6n - 2}{3n^2}$$

$$d_{1,4} = \frac{-8n^4 - 4n^3 + 35n^2 - 14n + 3}{6n^3(n-1)}$$

$$d_{1,5} = \frac{26n^4 - 10n^3 - 35n^2 + 25n - 6}{15n^4}$$

$$d_{1,6} = \frac{2(-n^2 + 2n + 2)}{n(n-1)}.$$

Note that $d_{1,1} \geq 0$ and $d_{1,2} \leq 0$. Since $n \geq 7$ implies $0.6 \leq -\frac{d_{1,1}}{d_{1,2}} = \frac{2n-2}{3n-1}$, we have $d_{1,2}\sigma^2 + d_{1,1}\sigma = \sigma(d_{1,2}\sigma + d_{1,1}) \geq 0$ for $\sigma \in (0, 0.6]$.

Now, let $r_1(n, \sigma) = d_{1,6}\sigma^3 + d_{1,5}\sigma^2 + d_{1,4}\sigma + d_{1,3}$. Then, we have $D_1 = \sigma^3 r_1 + d_{1,2}\sigma^2 + d_{1,1}\sigma$. Therefore, it suffices to show that $r_1 \geq 0$.

We proceed by examining the derivative of $r_1$ with respect to $\sigma$ to determine its monotonicity on this interval. The derivative is given by

$$\frac{d}{d\sigma} r_1(n, \sigma) = 3d_{1,6}\sigma^2 + 2d_{1,5}\sigma + d_{1,4}.$$

For $n \geq 7$, we have $3d_{1,6} \leq -4, 2d_{1,5} \leq 4$ and $d_{1,4} \leq -1$. Therefore, for $\sigma \in (0, 0.6]$, we have

$$\frac{d}{d\sigma} r_1(n, \sigma) \leq -4\sigma^2 + 4\sigma - 1$$

$$= -(2\sigma - 1)^2$$

$$\leq 0.$$

Thus, $r_1(n, \sigma)$ is decreasing in $\sigma$ on $(0, 0.6]$. Consequently, $r_1(n, \sigma) \geq r_1(n, 0.6)$ if $\sigma \in (0, 0.6]$. Furthermore, we have

$$r_1(0.6) = \frac{44n^5 + 700n^4 + 823n^3 + 530n^2 - 333n + 108}{750n^4(n-1)} \geq 0.$$

Thus, we can conclude that $D_1(n, \sigma) = \sigma^3 r_1(n, \sigma) + d_{1,2}\sigma^2 + d_{1,1}\sigma \geq 0$ for $\sigma \in (0, 0.6]$ and $n \geq 7$.

Similar to $P_1$, let $P_2 = \sum_{k=0}^{6} t_{2,k} \sigma^k$. Then, $T_2 \leq P_2$ by (23). Furthermore,

$$\frac{P_2(m+1, \sigma)}{(m+1)m} - \frac{P_2(m, \sigma)}{m(m-1)} = \frac{2}{m(m^2-1)} (mp_2(\sigma) + q_2(\sigma)) \tag{26}$$

where $p_2(\sigma) = 2\sigma^6 - 2\sigma^5 + 2\sigma^4 - 2\sigma^3 + 2\sigma^2 - 2\sigma + 1$ and $q_2(\sigma) = -8\sigma^6 + 2\sigma^5 - 4\sigma^4 + 2\sigma^3 + 2\sigma - 1$.

Then, it follows that $2p_1 + q_1 \geq 0$ for $\sigma \in (0, 0.6]$.

Moreover,

$$p_2(\sigma) = 1 - 2\sigma(1 - \sigma + \sigma^2 - \sigma^3 + \sigma^4 - \sigma^5)$$

$$= \frac{1 - \sigma + 2\sigma^7}{1 + \sigma}$$

$$\geq 0,$$

so $mp_2(\sigma) + q_2(\sigma) \geq 0$ for all $m \geq 2$. Therefore, $\frac{P_2(m,\sigma)}{m(m-1)}$ is increasing in $m$.

Now, similar to $D_1$, we define $D_2(n,\sigma) = \left(\sum_{r=0}^{5} \binom{2n}{r} \left(-\frac{\sigma}{n}\right)^r\right) - \frac{P_2(n,\sigma)}{n(n-1)} = \sum_{k=0}^{6} d_{2,k}\sigma^k$. Then,

$$d_{2,0} = \frac{2}{n}$$

$$d_{2,1} = -\frac{4}{n}$$

$$d_{2,2} = \frac{3n-1}{n(n-1)}$$

$$d_{2,3} = \frac{2n^2 - 6n - 2}{3n^2}$$

$$d_{2,4} = \frac{-8n^4 + 20n^3 - 13n^2 - 14n + 3}{6n^3(n-1)}$$

$$d_{2,5} = \frac{26n^4 - 40n^3 - 35n^2 + 25n - 6}{15n^4}$$

$$d_{2,6} = \frac{2(-n^2 + 3n - 5)}{n(n-1)}.$$

Let $r_2(n,\sigma) = d_{2,6}\sigma^3 + d_{2,5}\sigma^2 + d_{2,4}\sigma + d_{2,3}$. Then, we can express $D_2$ as

$$D_2 = \sigma^3 r_2 + d_{2,2}\sigma^2 + d_{2,1}\sigma + d_{2,0}.$$

We will show that the minimum of $D_2$ is non-negative on $(0, 0.6]$. First, we consider $d_{2,2}\sigma^2 + d_{2,1}\sigma + d_{2,0}$. This quadratic attains its minimum value of $\frac{2(n+1)}{n(3n-1)}$ at $\sigma = \frac{2(n-1)}{3n-1}$. Thus, to prove that $D_2 \geq 0$ on $(0, 0.6]$, it suffices to show that

$$\frac{2(n+1)}{n(3n-1)} + \min_{\sigma \in (0,0.6]} \sigma^3 r_2(\sigma) \geq 0.$$

If $\min_{\sigma \in (0,0.6]} r_2(\sigma) \geq 0$, then the inequality holds. On the other hand, if $\min_{\sigma \in (0,0.6]} r_2(\sigma) \leq 0$, then we have

$$\min_{\sigma \in (0,0.6]} \sigma^3 r_2(\sigma) \geq 0.6^3 \min_{\sigma \in (0,0.6]} r_2(\sigma).$$

Taking the derivative, we have

$$\frac{d}{d\sigma} r_2 = 3d_{2,6}\sigma^2 + 2d_{2,5}\sigma + d_{2,4}.$$

We can also observe that $3d_{2,6} \leq -\frac{13}{3}$, $2d_{2,5} \leq \frac{52}{15}$ and $d_{2,4} \leq -\frac{3}{4}$ for all $n$. Therefore, given that $\sigma \geq 0$, we can see that

$$\frac{d}{d\sigma} r_2 \leq -\frac{13}{3}\sigma^2 + \frac{52}{15}\sigma - \frac{3}{4}$$

$$\leq -\frac{17}{300}.$$

Hence, $r_2$ is decreasing in $\sigma$ and we have

$$\min_{\sigma \in (0,0.6]} r_2(\sigma) = r_2(0.6)$$

$$= \frac{44n^5 - 716n^4 - 1505n^3 + 530n^2 - 333n + 108}{750n^4(n-1)}.$$

For $n \geq 19$, we have $r_2(0.6) \geq 0$. In the remaining cases, we use the fact that $\min_{\sigma \in (0,0.6]} \sigma^3 r_2(\sigma) \geq 0.6^3 r_2(0.6)$. Then, for all $n$ such that $7 \leq n \leq 18$, we have

$$\frac{2(n+1)}{n(3n-1)} + \min_{\sigma \in (0,0.6]} \sigma^3 r_2(\sigma) \geq \frac{2(n+1)}{n(3n-1)} + 0.6^3 r_2(0.6)$$

$$= \frac{1188n^6 + 42772n^5 - 34191n^4 - 34645n^3 - 13761n^2 + 5913n - 972}{31250n^4(3n^2 - 4n + 1)}$$

$$\geq 0.$$

Hence, we can conclude that $D_2 \geq 0$ for $n \geq 7$ and $\sigma \in (0, 0.6]$.

Finally, we use the following lemma:

**Lemma B.8.** $\left(1 + \frac{c}{n}\right)^n$ *is increasing if* $n \geq -c$.

*Proof.* Let $a_n = \left(1 + \frac{c}{n}\right)^n$. Then, by the AM-GM inequality,

$$a_n = \left(1 + \frac{c}{n}\right)^n$$

$$= 1 \cdot \left(1 + \frac{c}{n}\right) \cdots \left(1 + \frac{c}{n}\right)$$

$$\leq \left(\frac{n+1+c}{n+1}\right)^{n+1}$$

$$= a_{n+1}.$$

$\square$

For $\sigma \in (0, 0.6]$ and $i = 1, 2$, we have

$$\max_{2 \leq m \leq n} \frac{T_i(m, \sigma)}{m(m-1)} = \max\left\{ \max_{2 \leq m \leq 6} \frac{T_i(m, \sigma)}{m(m-1)}, \max_{7 \leq m \leq n} \frac{T_i(m, \sigma)}{m(m-1)} \right\}$$

$$\leq \max\left\{ \left(1 - \frac{\sigma}{6}\right)^{12}, \max_{7 \leq m \leq n} \frac{T_i(m, \sigma)}{m(m-1)} \right\}$$

$$\leq \max\left\{ \left(1 - \frac{\sigma}{6}\right)^{12}, \max_{7 \leq m \leq n} \frac{P_i(m, \sigma)}{m(m-1)} \right\} \qquad (\because T_i \leq P_i \text{ for } n \geq 7)$$

$$\leq \max\left\{ \left(1 - \frac{\sigma}{6}\right)^{12}, \frac{P_i(n, \sigma)}{n(n-1)} \right\} \qquad \left(\because \frac{P_i(m, \sigma)}{m(m-1)} \text{ is increasing in } m \text{ for } m \geq 7\right)$$

$$\leq \max\left\{ \left(1 - \frac{\sigma}{6}\right)^{12}, \left(1 - \frac{\sigma}{n}\right)^{2n} \right\} \qquad (\because D_i \geq 0 \text{ and } (24))$$

$$\leq \left(1 - \frac{\sigma}{n}\right)^{2n}. \qquad (\because \text{Lemma B.8})$$

Note that we used the fact that $\max_{2 \leq m \leq 6} \frac{T_i(m,\sigma)}{m(m-1)} \leq \left(1 - \frac{\sigma}{6}\right)^{12}$ for the first inequality, which can be verified using Proposition B.6.

CASE II. $\sigma \in [0.6, 0.8]$

Since $\left(1 - \frac{\sigma}{n}\right)^{2n} \leq \left(1 - \frac{1}{n}\right)^n$ for $\sigma \in [0.6, 0.8]$, it is enough to show that

$$\max_{2 \leq m \leq n} \frac{T_i(m, \sigma)}{m(m-1)} \leq \left(1 - \frac{1}{n}\right)^n$$

for $i = 1, 2$.

We start by examining $P_1$, which is an upper bound for $T_1$. By Equation (25), if $mp_1 + q_1 \geq 0$, we have

$$\frac{P_1(m+1, \sigma)}{(m+1)m} - \frac{P_1(m, \sigma)}{m(m-1)} \geq 0.$$

Note that $p_1(\sigma) = \sigma(\sigma - 1)^2(\sigma + 1)(\sigma^2 + 1) \geq 0$ for $\sigma \geq 0$ and $2p_1 + q_1 \geq 0$ on $[0.6, 0.8]$. Therefore, $mp_1 + q_1 \geq 0$ for all $m \geq 2$, and

$$\begin{aligned}
\max_{2 \leq m \leq n} \frac{P_1(m, \sigma)}{m(m-1)} &= \frac{P_1(n, \sigma)}{n(n-1)} \\
&\leq \lim_{n \to \infty} \frac{P_1(n, \sigma)}{n(n-1)} \\
&= 2\sigma^6 - 2\sigma^5 + 2\sigma^4 - 2\sigma^3 + 2\sigma^2 - 2\sigma + 1
\end{aligned}$$

for $\sigma \in [0.6, 0.8]$.

Next, we consider $P_2$. We have already shown that if $\sigma \in \left[0, \min\left\{\frac{n-1}{n+1}, \frac{n-3}{n-2}\right\}\right]$, then (23) holds. Since we assumed that $n \geq 11$ in this case, and $0.8 \leq \frac{n-1}{n+1} \leq \frac{n-3}{n-2}$ for $n \geq 11$, we have $T_2 \leq P_2$ on $[0.6, 0.8]$.

Similar to $P_1$, we first note that $p_2(\sigma) = \frac{1 - \sigma + 2\sigma^7}{1 + \sigma} \geq 0$ on $[0.6, 0.8]$. Using the fact that $11p_2 + q_2 \geq 0$ on $[0.6, 0.8]$, we can conclude that

$$\begin{aligned}
\max_{11 \leq m \leq n} \frac{P_2(m, \sigma)}{m(m-1)} &= \frac{P_2(n, \sigma)}{n(n-1)} \\
&\leq \lim_{n \to \infty} \frac{P_2(n, \sigma)}{n(n-1)} \\
&= 2\sigma^6 - 2\sigma^5 + 2\sigma^4 - 2\sigma^3 + 2\sigma^2 - 2\sigma + 1.
\end{aligned}$$

We also note that $2\sigma^6 - 2\sigma^5 + 2\sigma^4 - 2\sigma^3 + 2\sigma^2 - 2\sigma + 1 \leq \frac{7}{20} \leq \left(1 - \frac{1}{11}\right)^{11}$ on $[0.6, 0.8]$.

Thus, for $i = 1, 2$,

$$\begin{aligned}
\max_{2 \leq m \leq n} \frac{T_i(m, \sigma)}{m(m-1)} &\leq \max\left\{\max_{2 \leq m \leq 10} \frac{T_i(m, \sigma)}{m(m-1)}, \max_{11 \leq m \leq n} \frac{T_i(m, \sigma)}{m(m-1)}\right\} \\
&\leq \max\left\{\frac{1}{4}, \max_{11 \leq m \leq n} \frac{T_i(m, \sigma)}{m(m-1)}\right\} \\
&\leq \max\left\{\frac{1}{4}, \max_{11 \leq m \leq n} \frac{P_i(m, \sigma)}{m(m-1)}\right\} \qquad (\because T_i \leq P_i \text{ for } m \geq 11) \\
&\leq \max\left\{\frac{1}{4}, 2\sigma^6 - 2\sigma^5 + 2\sigma^4 - 2\sigma^3 + 2\sigma^2 - 2\sigma + 1\right\} \\
&\leq \max\left\{\left(1 - \frac{1}{2}\right)^2, \left(1 - \frac{1}{11}\right)^{11}\right\} \\
&\leq \left(1 - \frac{1}{n}\right)^n. \qquad (\because \text{Lemma B.8})
\end{aligned}$$

The second inequality follows from the fact that $\max_{2 \leq m \leq 10} \frac{T_i(m, \sigma)}{m(m-1)} \leq \frac{1}{4}$, which can be confirmed using Proposition B.6.

CASE III. $\sigma \in [0.8, 1]$

If $\sigma = 1$, we have $T_1 = T_2 = 0$, and hence inequalities (19) and (20) are trivially satisfied. Therefore, we proceed to prove the inequalities for $\sigma \in [0.8, 1)$.

We first prove the following three lemmas.

**Lemma B.9.** $\log(1 + x) \geq \frac{x}{1+x}$ for $x > -1$.

*Proof.* The inequality is equivalent to $\log x \geq 1 + \frac{1}{x}$ for $x > 0$. This inequality can be easily verified by considering the function $g(x) = \log x + \frac{1}{x} - 1$. Since $g'(x) = \frac{1}{x} - \frac{1}{x^2}$, $g$ has the a local minimum at $x = 1$. Since $g(1) = 0$, we have $g(x) \geq 0$, which implies $\log x \geq 1 - \frac{1}{x}$. $\qquad\square$

**Lemma B.10.** $\frac{\log(1+x)}{x} \geq \frac{2}{2+x}$ *for* $-1 < x < 0$ *and* $x > 0$.

*Proof.* If $x > 0$, the inequality becomes $(2 + x)\log(1 + x) \geq 2x$. Define $f(x) = (2 + x)\log(1 + x) - 2x$. Then, $f'(x) = \log(1 + x) - \frac{x}{1+x}$ and $f''(x) = \frac{x}{(1+x)^2}$. Since $\lim_{x\downarrow 0} f'(x) = 0$ and $f''(x) > 0$, $f'(x) \geq 0$ on $x > 0$. Now, suppose $-1 < x < 0$. Then, the inequality is equivalent to $f(x) \leq 0$. Then, $\lim_{x\uparrow 0} f'(x) = 0$ and $f''(x) < 0$, so $f'(x) \leq 0$ on $-1 < x < 0$. Since $\lim_{x\to 0} f(x) = 0$, $f(x) \leq 0$ for $-1 < x < 0$. $\qquad\square$

**Lemma B.11.** $\log(1 + x) \geq x - \frac{x^2}{2}$ *for* $x \geq 0$.

*Proof.* Let $f(x) = \log(1 + x) - x + \frac{x^2}{2}$. Then, $f'(x) = \frac{1}{1+x} - 1 + x$ and $f''(x) = 1 - \frac{1}{(1+x)^2}$. Since $f(0) = f'(0) = 0$ and $f''(x) \geq 0$, $f'(x) \geq 0$, so $f(x) \geq 0$ and the inequality is proved. $\qquad\square$

We first find an upper bound for $\frac{\gamma}{n(n-1)}$. Recall that $\boldsymbol{w} = \boldsymbol{C}^\top \mathbf{1}$, $s_n = \sum_{i=0}^{n-1} \sigma^i$ and $L = n - (n - 1)\sigma$. We also define $r_n = n\left(\frac{n}{n+1}\right)^{n-1} - 1$.

We use the following two lemmas:

**Lemma B.12.**

$$0 \leq Ls_n - n \leq r_n.$$

*Proof.* The first inequality holds because $s_n = \frac{1-\sigma^n}{1-\sigma}$, $1 - \sigma \geq 0$ and

$$(1 - \sigma)\left(L\frac{1 - \sigma^n}{1 - \sigma} - n\right) = (n - (n - 1)\sigma)(1 - \sigma^n) - n(1 - \sigma)$$

$$= \sigma - n\sigma^n + (n - 1)\sigma^{n+1}$$

$$= \sigma(1 - \sigma)\left(\sum_{i=0}^{n-2}\sigma^i - (n - 1)\sigma^{n-1}\right)$$

$$\geq 0.$$

For the second inequality, we need to show that

$$(n - (n - 1)\sigma)(1 - \sigma^n) - n(1 - \sigma) + r_n(\sigma - 1) \leq 0.$$

To this end, let

$$g_n(\sigma) = (n - (n - 1)\sigma)(1 - \sigma^n) - n(1 - \sigma) + r_n(\sigma - 1)$$
$$= (n - 1)\sigma^{n+1} - n\sigma^n + \sigma + r_n(\sigma - 1).$$

Then, $g_n'(\sigma) = (n^2 - 1)\sigma^n - n^2\sigma^{n-1} + 1 + r_n$. Since $n^2\sigma^{n-1} - (n^2 - 1)\sigma^n$ has the maximum at $\sigma = \frac{n}{n+1}$ on $[0, 1]$, we have

$$g_n'(\sigma) \geq g_n'\left(\frac{n}{n+1}\right)$$
$$= 0$$

by the definition of $r_n$. Therefore, $g_n$ is increasing on $[0, 1]$ As $g_n(1) = 0$, we have $g_n \leq 0$ on $[0, 1]$, and therefore on $[0.8, 1)$. $\qquad\square$

**Lemma B.13.** $\frac{r_n^2}{n(n-1)} \leq \frac{1}{e^2}$.

*Proof.* Let $x > 1$ and $f(x) = x\left(\frac{x}{x+1}\right)^{x-1}$. By showing that $f'(x) \leq \frac{1}{e}$, we can show that $f(x) \leq \frac{1}{e}(x-1) + 1$ because $f(1) = 1$ and

$$\int_1^x f'(t)\, dt = f(x) - 1 \leq \frac{x-1}{e}.$$

Once the above is shown, it follows that

$$\frac{(f(x)-1)^2}{x(x-1)} \leq \left(\frac{f(x)-1}{x-1}\right)^2$$

$$\leq \frac{1}{e^2}$$

and therefore $\frac{r_n^2}{n(n-1)} \leq \frac{1}{e^2}$. To show that $f'(x) \leq \frac{1}{e}$, we differentiate $f(x)$ and obtain

$$f'(x) = \left(\frac{x}{x+1}\right)^{x-1}\left(\frac{2x}{x+1} - x\log\left(1+\frac{1}{x}\right)\right).$$

By Lemma B.10, $x\log\left(1+\frac{1}{x}\right) \geq \frac{2x}{2x+1}$, so

$$f'(x) \leq \left(\frac{x}{x+1}\right)^{x-1}\left(\frac{2x}{x+1} - \frac{2x}{2x+1}\right)$$

$$= \left(\frac{2x}{2x+1}\right)\left(\frac{x}{x+1}\right)^x.$$

Thus, it suffices to show

$$\left(\frac{2x}{2x+1}\right)\left(\frac{x}{x+1}\right)^x \leq \frac{1}{e}$$

which is equivalent to

$$1 \leq \log\left(1+\frac{1}{2x}\right) + x\log\left(1+\frac{1}{x}\right).$$

We now prove the inequality using Lemma B.10 and B.11. It suffices to show that $1 \leq \frac{1}{2x} - \frac{1}{8x^2} + \frac{2x}{2x+1}$ because $\frac{1}{2x} - \frac{1}{8x^2} \leq \log\left(1+\frac{1}{2x}\right)$ and $\frac{2x}{2x+1} \leq x\log\left(1+\frac{1}{x}\right)$. Multiplying both sides by $(2x+1)8x^2$, we have $8x^2 \leq (4x-1)(2x+1)$, which holds for all $x \geq \frac{1}{2}$ and therefore for $x > 1$. $\square$

By the lemma above, we have

$$\frac{\gamma}{n(n-1)} \leq \frac{1}{e^2} \tag{27}$$

because $\gamma = (n - Ls_n)^2$.

Next, we will focus on deriving an upper bound for $\frac{\delta}{n-1}$ on $[0,1]$. Since $\delta = \sum_{i=1}^{n-1}(\sigma^i - \sigma^n)^2 \leq \sum_{i=1}^{n-1}\max_{0\leq\sigma\leq1}(\sigma^i - \sigma^n)^2$, and $(\sigma^i - \sigma^n)^2$ attains its maximum at $\sigma = \left(\frac{i}{n}\right)^{1/(n-i)}$, we have

$$\delta \leq \sum_{i=1}^{n-1}\left(\left(\frac{i}{n}\right)^{\frac{i}{n}/(1-\frac{i}{n})} - \left(\frac{i}{n}\right)^{1/(1-\frac{i}{n})}\right)^2.$$

Now we use the following lemma:

**Lemma B.14.** $x^{x/(1-x)} - x^{1/(1-x)} \le e^{-(4\log 2)x}$ *if* $0 < x < 1$.

*Proof.* Let $f(x) = x^{x/(1-x)} - x^{1/(1-x)}$. Then, $f(x) = x^{x/(1-x)}(1-x)$ and therefore we need to show that $\log f(x) = \frac{x}{1-x}\log x + \log(1-x) \le -(4\log 2)x$. Let $g(x) = \frac{\log x}{1-x} + \frac{\log(1-x)}{x}$. Then, $g'(x) = \frac{x^2\log x - (1-x)^2\log(1-x)}{x(1-x)}$. To check the change of the sign of $g'$, define $h(x) = x^2\log x$. Since $h'(x) + h'(1-x) = 2x\log x + 2(1-x)\log(1-x) + 1$ and $2x\log x$ is convex, so is $h'(x) + h'(1-x)$ and $\lim_{x\uparrow 1} h'(x) + h'(1-x) = \lim_{x\downarrow 0} h'(x) + h'(1-x) = 1$ and $2h'(\frac{1}{2}) < 0$. Thus, for some $c \in (0, \frac{1}{2})$, $h'(x) + h'(1-x) \ge 0$ is $0 < x \le c$ or $1 - c \le x < 1$ and $h'(x) + h'(1-x) \le 0$ if $c \le x \le 1 - c$. Consequently, $g' \ge 0$ if $0 < x \le \frac{1}{2}$ and $g' \le 0$ if $\frac{1}{2} \le x < 1$. This shows that $g$ has the maximum $g\left(\frac{1}{2}\right) = -4\log 2$ at $x = \frac{1}{2}$ and the lemma has been proved. $\qquad\square$

By the lemma above,

$$\delta \le \sum_{i=1}^{n-1} e^{(-8\log 2)i/n}$$

$$\le \sum_{i=1}^{\infty} e^{(-8\log 2)i/n}$$

$$= \frac{1}{e^{(8\log 2)/n} - 1}$$

and since $\frac{t}{e^t - 1} \le 1$ for $t > 0$,

$$\frac{\delta}{n-1} \le \frac{1}{(n-1)\frac{8\log 2}{n}} \frac{\frac{8\log 2}{n}}{e^{(8\log 2)/n} - 1}$$

$$\le \frac{n}{n-1} \frac{1}{8\log 2}. \tag{28}$$

Next, we consider an upper and lower bounds of $\alpha$. Recall that

$$\alpha = n - 2L\frac{1-\sigma^n}{1-\sigma} + L^2\frac{1-\sigma^{2n}}{1-\sigma^2}.$$

First, we derive an upper bound of $\alpha$. To achieve this, we will establish upper bounds for the terms $-2L\frac{1-\sigma^n}{1-\sigma}$ and $L^2\frac{1-\sigma^{2n}}{1-\sigma^2}$. We start by analyzing $L\frac{1-\sigma^n}{1-\sigma}$. Since $L = n - (n-1)\sigma$ and $s_n = \frac{1-\sigma^n}{1-\sigma}$, by Lemma B.12, we have

$$L\frac{1-\sigma^n}{1-\sigma} \ge n. \tag{29}$$

Next we examine the term $L^2\frac{1-\sigma^{2n}}{1-\sigma^2}$. Since $L = (n-1)(1-\sigma) + 1$, we obtain

$$L^2\frac{1-\sigma^{2n}}{1-\sigma^2} = (1-\sigma^{2n})\left((n-1)^2\frac{1-\sigma}{1+\sigma} + \frac{2(n-1)}{1+\sigma} + \frac{1}{1-\sigma^2}\right)$$

$$\le (n-1)^2\frac{1-\sigma}{1+\sigma} + \frac{2(n-1)}{1+\sigma} + \frac{1-\sigma^{2n}}{1-\sigma^2}$$

$$\le (n-1)^2\frac{1-\sigma}{1+\sigma} + \frac{2(n-1)}{1+\sigma} + n. \tag{30}$$

The first inequality holds because $0 \le 1 - \sigma^{2n} \le 1$, while the second inequality follows from the fact that $\frac{1-\sigma^{2n}}{1-\sigma^2} = \sum_{i=0}^{n-1}\sigma^{2i} \le n$. From (29) and (30), we have

$$\frac{\alpha}{n(n-1)} \le \left(1 - \frac{1}{n}\right)\frac{1-\sigma}{1+\sigma} + \frac{2}{n(1+\sigma)}. \tag{31}$$

Next, we obtain a lower bound of $\alpha$. For any real $c$, we have

$$L\frac{1-\sigma^n}{1-\sigma} \leq nc \iff c(1-\sigma) - (1-\sigma^n)\left(1 - \left(1 - \frac{1}{n}\sigma\right)\right) \geq 0,$$

which motivates us to define $g(\sigma) = c(1-\sigma) - (1-\sigma^n)\left(1 - \left(1 - \frac{1}{n}\sigma\right)\right)$. Then, our objective is to find a value for $c$ such that $g(\sigma) \geq 0$. Now, we analyze the function $g(\sigma)$ to determine a suitable value for $c$. We have $g(1) = 0$ and

$$g'(\sigma) = \left(\frac{1}{n} - n\right)\sigma^n + n\sigma^{n-1} - c + 1 - \frac{1}{n}.$$

Since $\left(\frac{1}{n} - n\right)\sigma^n + n\sigma^{n-1} + 1 - \frac{1}{n}$ has the local maximum $\left(\frac{n}{n+1}\right)^{n-1} + 1 - \frac{1}{n}$ at $\sigma = \frac{n}{n+1}$, if $c = 1 + \frac{1}{e}$, $g'(\sigma) \leq 0$ by the following lemma:

**Lemma B.15.** $\left(\frac{x}{x+1}\right)^{x-1} - \frac{1}{x} \leq \frac{1}{e}$ if $x > 1$.

*Proof.* Let $f(x) = \left(\frac{x}{x+1}\right)^{x-1} - \frac{1}{x}$. We first show that $f$ is increasing on $x > 1$. Taking the derivative, we have

$$f'(x) = \left(\frac{x}{x+1}\right)^{x-1}\left(\log\frac{x}{x+1} + \frac{x-1}{x(x+1)}\right) + \frac{1}{x^2}.$$

To show that $f'(x) \geq 0$, we can use the inequality $\log x \geq 1 - \frac{1}{x}$ for $x > 0$, which is directly derived from Lemma B.9. Applying this inequality to the term $\log\frac{x}{x+1}$, we have

$$f'(x) = \left(\frac{x}{x+1}\right)^{x-1}\left(\log\frac{x}{x+1} + \frac{x-1}{x(x+1)}\right) + \frac{1}{x^2}$$
$$\geq \left(\frac{x}{x+1}\right)^{x-1}\left(-\frac{1}{x} + \frac{x-1}{x(x+1)}\right) + \frac{1}{x^2}$$
$$= -\frac{2}{x(x+1)}\left(\frac{x}{x+1}\right)^{x-1} + \frac{1}{x^2}.$$

Since

$$-\frac{2}{x(x+1)}\left(\frac{x}{x+1}\right)^{x-1} + \frac{1}{x^2} \geq 0 \iff \left(1 + \frac{1}{x}\right)^x \geq 2$$

and $\left(1 + \frac{1}{x}\right)^x \geq 2$ holds for all $x > 1$ by Bernoulli's inequality, it follows that $f'(x) \geq 0$ and $f(x)$ is increasing. Since $\lim_{x\to\infty} f(x) = \frac{1}{e}$ and $f(x)$ is continuous on $x > 1$, we have $f(x) \leq \frac{1}{e}$. $\square$

Therefore, $g(\sigma) \geq 0$ on $[0, 1]$ and

$$L\frac{1-\sigma^n}{1-\sigma} \leq n\left(1 + \frac{1}{e}\right). \tag{32}$$

For the term $L^2\frac{1-\sigma^{2n}}{1-\sigma^2}$, we use the fact that

$$L^2\frac{1-\sigma^{2n}}{1-\sigma^2} = (1-\sigma^{2n})\left((n-1)^2\frac{1-\sigma}{1+\sigma} + \frac{2(n-1)}{1+\sigma} + \frac{1}{1-\sigma^2}\right)$$
$$= (n-1)^2\left(\frac{1-\sigma}{1+\sigma} - \sigma^{2n}\frac{1-\sigma}{1+\sigma}\right) + (1-\sigma^{2n})\left(\frac{2(n-1)}{1+\sigma} + \frac{1}{1-\sigma^2}\right).$$

Since we are considering the case $\sigma \in [0.8, 1)$, we have $\frac{1}{1+\sigma} \leq \frac{5}{9}$. Also, $\sigma^{2n}(1-\sigma)$ has the local maximum at $\sigma = \frac{2n}{2n+1}$, and since $\left(\frac{2n}{2n+1}\right)^{2n}$ is decreasing (the inverse $\left(1 + \frac{1}{2n}\right)^{2n}$ is positive and increasing by Lemma B.8),

$$
\sigma^{2n} \frac{1-\sigma}{1+\sigma} \leq \frac{5}{9} \sigma^{2n}(1-\sigma)
$$

$$
\leq \frac{5}{9} \left(\frac{2n}{2n+1}\right)^{2n} \frac{1}{2n+1}
$$

$$
\leq \frac{256}{1125} \frac{1}{2n+1}.
$$

Next, since

$$
(1 - \sigma^2)\left((1 - \sigma^{2n})\left(\frac{2(n-1)}{1+\sigma} + \frac{1}{1-\sigma^2}\right) - n\right) = (1 - \sigma^{2n})(2(n-1)(1-\sigma) + 1) - n(1 - \sigma^2)
$$

$$
= (1-\sigma)^2 \left(n - 1 + \sum_{i=2}^{2n-1} (i-1)\sigma^i\right)
$$

$$
\geq 0,
$$

we have

$$
(1 - \sigma^{2n})\left(\frac{2(n-1)}{1+\sigma} + \frac{1}{1-\sigma^2}\right) \geq n
$$

because $1 - \sigma^2 \geq 0$. Therefore,

$$
L^2 \frac{1 - \sigma^{2n}}{1 - \sigma^2} \geq (n-1)^2 \left(\frac{1-\sigma}{1+\sigma} - \frac{256}{1125}\frac{1}{2n+1}\right) + n. \tag{33}
$$

By (32) and (33), we obtain

$$
\frac{\alpha}{n(n-1)} \geq \frac{1}{n(n-1)}\left((n-1)^2\left(\frac{1-\sigma}{1+\sigma} - \frac{256}{1125}\frac{1}{2n+1}\right) - \frac{2}{e}n\right). \tag{34}
$$

We now derive an upper bound for $\beta$. Since

$$
\beta = \frac{1-\sigma}{1+\sigma}\left(2n - n\sigma^{2n} - 2(1-\sigma^{n+1})\frac{1-\sigma^n}{1-\sigma} - \sigma^2\frac{1-\sigma^{2n}}{1-\sigma^2}\right)
$$

$$
\leq \frac{1-\sigma}{1+\sigma}\left(2n - 2(1-\sigma^{n+1})\frac{1-\sigma^n}{1-\sigma}\right),
$$

we need to find a lower bound for the term $(1-\sigma^{n+1})\frac{1-\sigma^n}{1-\sigma}$. We claim that $(1-\sigma^{n+1})\frac{1-\sigma^n}{1-\sigma} \geq 20(1-\sigma)$. Since $1-\sigma > 0$, this is equivalent to showing that $\frac{1-\sigma^{n+1}}{1-\sigma}\frac{1-\sigma^n}{1-\sigma} \geq 20$. We can rewrite the inequality as

$$
\left(\sum_{i=0}^{n} \sigma^i\right)\left(\sum_{i=0}^{n-1} \sigma^i\right) \geq 20,
$$

The left-hand side is increasing in $\sigma$ for $\sigma \in (0,1)$. Given that $n \geq 15$, we can verify the inequality holds because

$$
\left(\sum_{i=0}^{n} 0.8^i\right)\left(\sum_{i=0}^{n-1} 0.8^i\right) \geq 20.
$$

Therefore, we have $(1-\sigma^{n+1})\frac{1-\sigma^n}{1-\sigma} \geq 20(1-\sigma)$ as claimed.

Substituting this into the inequality for $\beta$, we obtain

$$
\begin{aligned}
\beta &= \frac{1-\sigma}{1+\sigma}\left(2n - n\sigma^{2n} - 2(1-\sigma^{n+1})\frac{1-\sigma^n}{1-\sigma} - \sigma^2\frac{1-\sigma^{2n}}{1-\sigma^2}\right) \\
&= \frac{1-\sigma}{1+\sigma}\left(2n - 2(1-\sigma^{n+1})\frac{1-\sigma^n}{1-\sigma}\right) \\
&\leq \frac{1-\sigma}{1+\sigma}\left(2n - 40(1-\sigma)\right).
\end{aligned}
\tag{35}
$$

Using the derived bounds for $\alpha, \beta, \gamma$ and $\delta$, we now establish upper bounds for $T_1$ and $T_2$. Combining (28), (34) and (35), we have

$$
\begin{aligned}
\frac{T_1}{n(n-1)} &\leq \frac{1}{n(n-1)}(n\beta - \alpha + \max_{\sigma\in(0,1]} n\delta) \\
&\leq \frac{2n}{n-1}\frac{1-\sigma}{1+\sigma} - \frac{40}{n-1}\frac{(1-\sigma)^2}{1+\sigma} - \frac{1}{n(n-1)}\left((n-1)^2\left(\frac{1-\sigma}{1+\sigma} - \frac{256}{1125}\frac{1}{2n+1}\right) - \frac{2}{e}n\right) \\
&\quad + \frac{1}{8\log 2}\frac{n}{n-1} \\
&= \frac{n^2+2n-1}{n(n-1)}\frac{1-\sigma}{1+\sigma} - \frac{40}{n-1}\frac{(1-\sigma)^2}{1+\sigma} + \frac{256}{1125}\frac{n-1}{n(2n+1)} + \frac{2}{e}\frac{1}{n-1} + \frac{1}{8\log 2}\frac{n}{n-1} \\
&\leq \frac{5}{9}\left(\frac{n^2+2n-1}{n(n-1)}(1-\sigma) - \frac{40}{n-1}(1-\sigma)^2\right) + \frac{256}{1125}\frac{n-1}{n(2n+1)} + \frac{2}{e}\frac{1}{n-1} + \frac{1}{8\log 2}\frac{n}{n-1},
\end{aligned}
$$

where the last inequality holds because $n \geq 15$ implies $\frac{n^2+2n-1}{n(n-1)}(1-\sigma) - \frac{40}{n-1}(1-\sigma)^2 \geq 0$ and $\frac{1}{1+\sigma} \leq \frac{5}{9}$.

Similarly, by (27) and (31), we have

$$
\begin{aligned}
\frac{T_2}{n(n-1)} &\leq \frac{1}{n(n-1)}(\alpha + \max_{\sigma\in(0,1]} \gamma) \\
&\leq \left(1 - \frac{1}{n}\right)\frac{1-\sigma}{1+\sigma} + \frac{2}{n(1+\sigma)} + \frac{1}{e^2} \\
&\leq -\frac{5}{9}\left(1 - \frac{1}{n}\right)\sigma + \frac{5}{9}\left(1 + \frac{1}{n}\right) + \frac{1}{e^2}. && \left(\because \frac{1}{1+\sigma} \leq \frac{5}{9}\right)
\end{aligned}
$$

Now, let

$$
\begin{aligned}
U_1(n,\sigma) &= \frac{5}{9}\left(\frac{n^2+2n-1}{n(n-1)}(1-\sigma) - \frac{40}{n-1}(1-\sigma)^2\right) + \frac{256}{1125}\frac{n-1}{n(2n+1)} + \frac{2}{e}\frac{1}{n-1} + \frac{1}{8\log 2}\frac{n}{n-1} \\
U_2(n,\sigma) &= -\frac{5}{9}\left(1 - \frac{1}{n}\right)\sigma + \frac{5}{9}\left(1 + \frac{1}{n}\right) + \frac{1}{e^2}.
\end{aligned}
$$

We first consider $U_1(n,\sigma)$. Note that $U_1(n,\sigma)$ is a quadratic function of $\sigma$, which attains its maximum at $\sigma = 1 - \frac{n^2+2n-1}{80n}$. Thus, $U_1(n,\sigma)$ has its maximum at $\sigma = 0.8$ on $[0.8, 1)$ because $1 - \frac{n^2+2n-1}{80n} \leq 0.8$ for all $n \geq 15$. Since

$$
\begin{aligned}
U_1(n,\sigma) &\leq U_1(n,0.8) \\
&= \frac{n^2+2n-1}{9n(n-1)} - \frac{8}{9(n-1)} + \frac{256}{1125}\frac{n-1}{n(2n+1)} + \frac{2}{e}\frac{1}{n-1} + \frac{1}{8\log 2}\frac{n}{n-1} \\
&\leq \frac{1}{9}\left(1 + \frac{3}{n} + \frac{2}{n(n-1)}\right) + \left(\frac{128}{1125} + \frac{2}{e} + \frac{1}{8\log 2} - \frac{8}{9}\right)\frac{1}{n-1} && \left(\because \frac{n-1}{n(2n+1)} \leq \frac{1}{2(n-1)}\right)
\end{aligned}
$$

and the last line is decreasing in $n$, substituting $n = 15$ yields

$$U_1(n, \sigma) \leq U_1(n, 0.8)$$
$$\leq \frac{1867}{23625} + \left(\frac{2}{e} + \frac{1}{8 \log 2}\right) \frac{1}{14} + \frac{1}{8 \log 2}$$
$$\leq \left(1 - \frac{1}{15}\right)^{15}$$

for $n \geq 15$ and $\sigma \in [0.8, 1)$.

Similarly, since $U_2$ is decreasing in $\sigma$,

$$U_2(n, \sigma) \leq U_2(n, 0.8)$$
$$= \frac{1}{n} + \frac{1}{9} + \frac{1}{e^2}$$
$$\leq \left(1 - \frac{1}{10}\right)^{10}$$

for $n \geq 10$.

Recall that we can verify that

$$\frac{T_1(m, \sigma)}{m(m-1)} \leq \frac{1}{4}$$

for $2 \leq m \leq 14$ and

$$\frac{T_2(m, \sigma)}{m(m-1)} \leq \frac{1}{4}$$

for $2 \leq m \leq 9$ using Proposition B.6.

Therefore, we have

$$\max_{2 \leq m \leq n} \frac{T_1(m, \sigma)}{m(m-1)} = \max\left\{\max_{2 \leq m \leq 14} \frac{T_1(m, \sigma)}{m(m-1)}, \max_{15 \leq m \leq n} \frac{T_1(m, \sigma)}{m(m-1)}\right\}$$
$$\leq \max\left\{\left(1 - \frac{1}{2}\right)^{2}, \left(1 - \frac{1}{15}\right)^{15}\right\}$$
$$\leq \left(1 - \frac{1}{n}\right)^{n} \qquad (\because \text{Lemma B.8})$$

if $n \geq 15$ and

$$\max_{2 \leq m \leq n} \frac{T_1(m, \sigma)}{m(m-1)} \leq \left(1 - \frac{1}{2}\right)^{2}$$
$$\leq \left(1 - \frac{1}{n}\right)^{n}$$

if $2 \leq n \leq 14$.

Similarly, we have

$$\max_{2 \leq m \leq n} \frac{T_2(m, \sigma)}{m(m-1)} = \max\left\{\max_{2 \leq m \leq 9} \frac{T_2(m, \sigma)}{m(m-1)}, \max_{10 \leq m \leq n} \frac{T_2(m, \sigma)}{m(m-1)}\right\}$$
$$\leq \max\left\{\left(1 - \frac{1}{2}\right)^{2}, \left(1 - \frac{1}{10}\right)^{10}\right\}$$
$$\leq \left(1 - \frac{1}{n}\right)^{n} \qquad (\because \text{Lemma B.8})$$

if $n \geq 10$ and

$$\max_{2 \leq m \leq n} \frac{T_2(m, \sigma)}{m(m-1)} \leq \left(1 - \frac{1}{2}\right)^2$$

$$\leq \left(1 - \frac{1}{n}\right)^n$$

if $2 \leq n \leq 9$.

$\square$

## B.3. Proof of Theorem 3.4

Here we prove Theorem 3.4, restated below for the sake of readability.

**Theorem 3.4.** *Let* $A \in \mathcal{A}_\sigma$. *For an initial point* $x_0 \in \mathbb{R}^n$, *let* $x_T$ *be the output of* RCD *after* $T$ *iterates. Then, except for a Lebesgue measure zero set of initial points,*

$$\lim_{T \to \infty} \left(\frac{\mathbb{E}\left[\|x_T\|^2\right]}{\|x_0\|^2}\right)^{\frac{1}{T}} \geq 1 - \frac{1}{n} + \frac{(1-\sigma)^2}{n}.$$

*Proof.* We first assume that $A = \sigma I_n + (1 - \sigma) \mathbf{1}_n \mathbf{1}_n^\top$. Then,

$$\frac{\mathbb{E}\left[\|x_T\|^2\right]}{\|x_0\|^2} = \frac{x_0^\top (\mathcal{M}_A^{\mathrm{RCD}})^T(I) x_0}{x_0^\top x_0}$$

$$\geq \lambda_{\min}((\mathcal{M}_A^{\mathrm{RCD}})^T(I))$$

because $(\mathcal{M}_A^{\mathrm{RCD}})^T(I)$ is symmetric.

Now, we observe that $\mathrm{span}\{I, \mathbf{1}\mathbf{1}^\top\}$ is invariant under $\mathcal{M}_A^{\mathrm{RCD}}$. To verify this, we compute $\mathcal{M}_A^{\mathrm{RCD}}(I)$ and $\mathcal{M}_A^{\mathrm{RCD}}(\mathbf{1}\mathbf{1}^\top)$. Then, we obtain

$$\mathcal{M}_A^{\mathrm{RCD}}(I) = \frac{n - 1 + (1-\sigma)^2}{n} I + \frac{(1-\sigma)^2(n-2)}{n} \mathbf{1}\mathbf{1}^\top$$

$$\mathcal{M}_A^{\mathrm{RCD}}(\mathbf{1}\mathbf{1}^\top) = \frac{\sigma^2}{n} I + \frac{\sigma^2(n-2)}{n} \mathbf{1}\mathbf{1}^\top,$$

which confirms that both $\mathcal{M}_A^{\mathrm{RCD}}(I)$ and $\mathcal{M}_A^{\mathrm{RCD}}(\mathbf{1}\mathbf{1}^\top)$ are in $\mathrm{span}\{I, \mathbf{1}\mathbf{1}^\top\}$. Consequently, we can only focus on the $2 \times 2$ matrix $M_A$ which represents the restriction of $\mathcal{M}_A^{\mathrm{RCD}}$ to $\mathrm{span}\{I, \mathbf{1}\mathbf{1}^\top\}$, where

$$M_A = \frac{1}{n} \begin{bmatrix} n - 1 + (1-\sigma)^2 & (1-\sigma)^2(n-2) \\ \sigma^2 & \sigma^2(n-2) \end{bmatrix}.$$

That is, if

$$M_A \begin{bmatrix} a \\ b \end{bmatrix} = \begin{bmatrix} a' \\ b' \end{bmatrix},$$

then $\mathcal{M}_A^{\mathrm{RCD}}(aI + b\mathbf{1}\mathbf{1}^\top) = a'I + b'\mathbf{1}\mathbf{1}^\top$.

Now, define $\alpha_T$ and $\beta_T$ as

$$\begin{bmatrix} \alpha_T \\ \beta_T \end{bmatrix} = M_A^T \begin{bmatrix} 1 \\ 0 \end{bmatrix}$$

Then,

$$\lambda_{\min}((\mathcal{M}_A^{\mathrm{RCD}})^T(I)) = \lambda_{\min}(\alpha_T I + \beta_T \mathbf{1}\mathbf{1}^\top)$$

$$= \begin{cases} \alpha_T & \text{if } \beta_T \geq 0 \\ \alpha_T + n\beta_T & \text{if } \beta_T < 0. \end{cases}$$

Since the entries of $M_A$ are all non-negative, we have $\beta_T \geq 0$ for all $T \geq 0$. Therefore, we have

$$\lambda_{\min}((\mathcal{M}_A^{\text{RCD}})^T(I)) = \alpha_T.$$

When $n = 2$, we have

$$M_A = \frac{1}{2} \begin{bmatrix} 1 + (1-\sigma)^2 & 0 \\ \sigma^2 & 0 \end{bmatrix}$$

and

$$M_A^T \begin{bmatrix} 1 \\ 0 \end{bmatrix} = \begin{bmatrix} \left(\frac{1+(1-\sigma)^2}{2}\right)^T \\ \sigma^2 \left(\frac{1+(1-\sigma)^2}{2}\right)^{T-1} \end{bmatrix}.$$

Therefore, $\alpha_T = \left(\frac{1+(1-\sigma)^2}{2}\right)^T$ and we obtain

$$\left(\frac{\mathbb{E}\left[\|x_T\|^2\right]}{\|x_0\|^2}\right)^{1/T} = \alpha_T^{1/T}$$

$$= \frac{1 + (1-\sigma)^2}{2}.$$

This establishes the result for the case $n = 2$.

We now turn to the general case where $n \geq 3$. Let $v$ be the dominant eigenvector (i.e., the eigenvector corresponding to the eigenvalue with the largest magnitude) of $M_A$. We will show that $\begin{bmatrix} 1 & 0 \end{bmatrix}^\top$ is not orthogonal to $v$. Assume for contradiction that $\begin{bmatrix} 1 & 0 \end{bmatrix}^\top$ is orthogonal to $v$. Then, $v$ must be of the form $\begin{bmatrix} 0 & c \end{bmatrix}^\top$ for some nonzero scalar $c$. Since $v$ is an eigenvector of $M_A$, $M_A v$ must also be a multiple of $\begin{bmatrix} 0 & 1 \end{bmatrix}^\top$. This implies that $(M_A)_{12} = 0$. However, it implies $\sigma = 1$. In this case,

$$M_A = \begin{bmatrix} 1 - \frac{1}{n} & 0 \\ \frac{1}{n} & 1 - \frac{2}{n} \end{bmatrix}$$

and $M_A v = \left(1 - \frac{2}{n}\right) v$. Nevertheless, $1 - \frac{2}{n}$ is not the the eigenvalue of $M_A$ with the largest absolute value because the eigenvalues of $M_A$ are $1 - \frac{1}{n}$ and $1 - \frac{2}{n}$. Therefore, $\begin{bmatrix} 1 & 0 \end{bmatrix}^\top$ is not orthogonal to $v$. This implies that

$$\lim_{T \to \infty} \alpha_T^{1/T} = \rho(M_A).$$

Now, let $p(\lambda)$ be the characteristic polynomial of $M_A$, which is defined as $\det(\lambda I - M_A)$. Since $M_A \in \mathbb{R}^{2 \times 2}$, we have

$$p(\lambda) = \lambda^2 - \text{tr}(M_A)\lambda + \det(M_A)$$
$$= \lambda^2 - ((M_A)_{11} + (M_A)_{22})\lambda + ((M_A)_{11}(M_A)_{22} - (M_A)_{12}(M_A)_{21}).$$

Furthermore, since $\text{tr}(M_A) = \frac{1}{n}((n-1)\sigma^2 - 2\sigma + n) \geq 0$, $\det(M_A) = \frac{1}{n^2}\sigma^2(n-1)(n-2) \geq 0$, $p(\lambda)$ does not have a negative root. Thus,

$$p\left(1 - \frac{1}{n} + \frac{(1-\sigma)^2}{n}\right) = p((M_A)_{11})$$
$$= (M_A)_{11}^2 - ((M_A)_{11} + (M_A)_{22})(M_A)_{11} + ((M_A)_{11}(M_A)_{22} - (M_A)_{12}(M_A)_{21})$$
$$= -(M_A)_{12}(M_A)_{21}$$
$$\leq 0,$$

because the entries of $M_A$ are all non-negative.

Therefore, we have

$$1 - \frac{1}{n} + \frac{(1-\sigma)^2}{n} \leq \rho(\boldsymbol{M_A}).$$

Next, consider $\boldsymbol{A} = \mathrm{diag}\{\sigma \boldsymbol{I}_k + (1-\sigma)\mathbf{1}_k \mathbf{1}_k^\top, \boldsymbol{I}_{n-k}\}$ for an integer $k$ with $2 \leq k \leq n$. Since $\sigma = 1$ implies $\boldsymbol{A} = \boldsymbol{I}_n$, which has already been considered, we will assume that $\sigma \in (0,1)$.

We will denote $\sigma \boldsymbol{I}_k + (1-\sigma)\mathbf{1}_k\mathbf{1}_k^\top$ as $\boldsymbol{A}_k$, and the set of of block-diagonal matrices consisting of a $k \times k$ block and an $(n-k) \times (n-k)$ block as $\mathcal{D}_{k,n-k}$. Recall that $\mathcal{D}_{k,n-k}$ is closed under scalar multiplication, matrix multiplication, addition, transposition, and inversion. Thus, if $\boldsymbol{X} \in \mathcal{D}_{k,n-k}$, then $\mathcal{M}_{\boldsymbol{A}}^{\mathrm{RCD}}(\boldsymbol{X}) \in \mathcal{D}_{k,n-k}$ because $\boldsymbol{T}_{\boldsymbol{A},i}^{\mathrm{RCD}} = \boldsymbol{I} - \boldsymbol{E}_i \boldsymbol{A} \in \mathcal{D}_{k,n-k}$. Therefore, we will now compute the $(1,1)$-block and $(2,2)$-block of $\mathcal{M}_{\boldsymbol{A}}^{\mathrm{RCD}}(\boldsymbol{X})$ when $\boldsymbol{X} \in \mathcal{D}_{k,n-k}$.

To analyze the $(1,1)$-block of $\mathcal{M}_{\boldsymbol{A}}^{\mathrm{RCD}}(\boldsymbol{X})$, we consider two cases for the index $i$: $1 \leq i \leq k$ and $k+1 \leq i \leq n$. If $1 \leq i \leq k$, the $(1,1)$-block of $\boldsymbol{T}_{\boldsymbol{A},i}^{\mathrm{RCD}\top} \boldsymbol{X} \boldsymbol{T}_{\boldsymbol{A},i}^{\mathrm{RCD}}$ is $(\boldsymbol{I} - \boldsymbol{E}_{i,k}\boldsymbol{A}_k)^\top \boldsymbol{X}_{11}(\boldsymbol{I} - \boldsymbol{E}_{i,k}\boldsymbol{A}_k)$, where $\boldsymbol{E}_{i,k}$ denotes the $k \times k$ matrix $\boldsymbol{e}_i\boldsymbol{e}_i^\top$. If $k+1 \leq i \leq n$, since the $(1,1)$-block of $\boldsymbol{E}_i$ is zero, the $(1,1)$-block of $\boldsymbol{T}_{\boldsymbol{A},i}^{\mathrm{RCD}\top} \boldsymbol{X} \boldsymbol{T}_{\boldsymbol{A},i}^{\mathrm{RCD}}$ is $\boldsymbol{X}_{11}$.

Therefore, the $(1,1)$-block of $\mathcal{M}_{\boldsymbol{A}}^{\mathrm{RCD}}(\boldsymbol{X})$ is

$$\frac{1}{n}\left(\sum_{i=1}^k (\boldsymbol{I} - \boldsymbol{E}_{i,k}\boldsymbol{A}_k)^\top \boldsymbol{X}_{11}(\boldsymbol{I} - \boldsymbol{E}_{i,k}\boldsymbol{A}_k) + \sum_{i=k+1}^n \boldsymbol{X}_{11}\right) = \frac{1}{n}\left(k\mathcal{M}_{\boldsymbol{A}_k}^{\mathrm{RCD}}(\boldsymbol{X}_{11}) + (n-k)\boldsymbol{X}_{11}\right).$$

We denote the matrix operator $\boldsymbol{X} \mapsto \frac{1}{n}\left(k\mathcal{M}_{\boldsymbol{A}_k}^{\mathrm{RCD}}(\boldsymbol{X}) + (n-k)\boldsymbol{X}\right)$ on $\mathbb{R}^{k \times k}$ as $\mathcal{M}_{\boldsymbol{A}}$. Note that $\mathrm{span}\{\boldsymbol{I}_k, \mathbf{1}_k\mathbf{1}_k^\top\}$ is invariant under $\mathcal{M}_{\boldsymbol{A}}$, and the matrix representation of $\mathcal{M}_{\boldsymbol{A}}|_{\mathrm{span}\{\boldsymbol{I},\mathbf{1}\mathbf{1}^\top\}}$ is

$$\frac{k}{n}\boldsymbol{M}_{\boldsymbol{A}_k} + \left(1 - \frac{k}{n}\right)\boldsymbol{I}_2,$$

where

$$\boldsymbol{M}_{\boldsymbol{A}_k} = \frac{1}{k}\begin{bmatrix} k - 1 + (1-\sigma)^2 & (1-\sigma)^2(k-2) \\ \sigma^2 & \sigma^2(k-2) \end{bmatrix}.$$

Let us now consider the $(2,2)$-block of $\mathcal{M}_{\boldsymbol{A}}^{\mathrm{RCD}}(\boldsymbol{X})$. Given that the $(2,2)$-block of $\boldsymbol{A}$ is $\boldsymbol{I}_{n-k}$, the $(2,2)$-block of can be computed as $\boldsymbol{X}_{22}$ when $1 \leq i \leq k$ and $(\boldsymbol{I}_{n-k} - \boldsymbol{E}_{i-k,n-k})^\top \boldsymbol{X}_{22}(\boldsymbol{I}_{n-k} - \boldsymbol{E}_{i-k,n-k})$ when $k+1 \leq i \leq n$.

Therefore, if we denote the diagonal part of $\boldsymbol{X}_{22}$ as $\boldsymbol{D}_{22}$, the $(1,1)$-block of $\mathcal{M}_{\boldsymbol{A}}^{\mathrm{RCD}}(\boldsymbol{X})$ is

$$\frac{1}{n}\left(\sum_{i=1}^k \boldsymbol{X}_{22} + \sum_{i=k+1}^n (\boldsymbol{I}_{n-k} - \boldsymbol{E}_{i-k,n-k})^\top \boldsymbol{X}_{22}(\boldsymbol{I}_{n-k} - \boldsymbol{E}_{i-k,n-k})\right) = \frac{1}{n}\left(k\boldsymbol{X}_{22} + (n-k-2)\boldsymbol{X}_{22} + \boldsymbol{D}_{22}\right)$$

$$= \left(1 - \frac{2}{n}\right)\boldsymbol{X}_{22} + \frac{1}{n}\boldsymbol{D}_{22}$$

because $\sum_{i=k+1}^n \boldsymbol{E}_{i-k,n-k} = \boldsymbol{I}_{n-k}$ and $\sum_{i=k+1}^n \boldsymbol{E}_{i-k,n-k}^\top \boldsymbol{X}_{22} \boldsymbol{E}_{i-k,n-k} = \boldsymbol{D}_{22}$.

From the above, we have

$$(\mathcal{M}_{\boldsymbol{A}}^{\mathrm{RCD}})^T(\boldsymbol{I}_n) = \begin{bmatrix} (\mathcal{M}_{\boldsymbol{A}_k}^{\mathrm{RCD}})^T(\boldsymbol{I}_k) & \boldsymbol{0}_{k,n-k} \\ \boldsymbol{0}_{n-k,k} & \left(1 - \frac{1}{n}\right)^T \boldsymbol{I}_{n-k} \end{bmatrix}.$$

Now, we define $\alpha_T$ and $\beta_T$ as

$$\begin{bmatrix} \alpha_T \\ \beta_T \end{bmatrix} = \boldsymbol{M}_{\boldsymbol{A}}^T \begin{bmatrix} 1 \\ 0 \end{bmatrix}.$$

We now proceed to examine the limit

$$\lim_{T \to \infty} \left( \frac{x_0^\top (\mathcal{M}_A^{\text{RCD}})^T (I) x_0}{x_0^\top x_0} \right)^{1/T}.$$

To this end, let $\begin{bmatrix} y_0 & z_0 \end{bmatrix}^\top = x_0$, where $y_0 \in \mathbb{R}^k$ and $z_0 \in \mathbb{R}^{n-k}$. Since the set $\{x_0 \in \mathbb{R}^n : y_0 = 0_k\}$ has measure zero in $\mathbb{R}^n$, we assume that $y_0 \neq 0$. Then,

$$x_0^\top (\mathcal{M}_A^{\text{RCD}})^T (I) x_0 = y_0^\top (\alpha_T I + \beta_T \mathbf{1} \mathbf{1}^\top) y_0 + \left( 1 - \frac{1}{n} \right)^T \|z_0\|^2$$

$$\geq \alpha_T \|y_0\|^2 + \left( 1 - \frac{1}{n} \right)^T \|z_0\|^2. \qquad (\because \beta_T \geq 0)$$

By the same argument in the case when $k = n$, we can show that

$$\lim_{T \to \infty} \alpha_T^{1/T} = \rho \left( \frac{k}{n} M_{A_k} + \left( 1 - \frac{k}{n} \right) I_2 \right)$$

$$\geq \frac{k}{n} \left( 1 - \frac{1}{k} + \frac{(1 - \sigma)^2}{k} \right) + 1 - \frac{k}{n}$$

$$= 1 - \frac{1}{n} + \frac{(1 - \sigma)^2}{n}.$$

Since $\sigma \neq 1$ by the assumption, we have

$$\lim_{T \to \infty} \frac{\left( 1 - \frac{1}{n} \right)^T}{\alpha_T} = \lim_{T \to \infty} \left( \frac{\left( 1 - \frac{1}{n} \right)}{\alpha_T^{1/T}} \right)^T$$

$$= 0$$

and

$$\lim_{T \to \infty} \left( \frac{x_0^\top (\mathcal{M}_A^{\text{RCD}})^T (I) x_0}{x_0^\top x_0} \right)^{1/T} = \lim_{T \to \infty} \left( \frac{\alpha_T \|y_0\|^2 + \left( 1 - \frac{1}{n} \right)^T \|z_0\|^2}{\|y_0\|^2 + \|z_0\|^2} \right)^{1/T}$$

$$= \lim_{T \to \infty} \alpha_T^{1/T} \left( \frac{\|y_0\|^2}{\|y_0\|^2 + \|z_0\|^2} + \frac{\left( 1 - \frac{1}{n} \right)^T}{\alpha_T} \left( \frac{\|z_0\|^2}{\|y_0\|^2 + \|z_0\|^2} \right) \right)^{1/T}$$

$$= \lim_{T \to \infty} \alpha_T^{1/T}$$

$$\geq 1 - \frac{1}{n} + \frac{(1 - \sigma)^2}{n}.$$

$\square$

## C. Computational Verification of Inequalities Deferred from Appendix B

In this section, we provide the computer-assisted verification of the inequalities presented in Appendix B, which were stated to be verifiable using Sturm's theorem (Proposition B.6). Recall that for a polynomial $p$, if $p(x) \neq 0$ on $(a, b)$ and $p(a) > 0$, then $p(b) \geq 0$; similarly, if $p(b) > 0$, then $p(a) \geq 0$. Therefore, to verify that $p(x) \geq 0$ on an interval, it suffices to show that $p(x)$ has no real roots in the interval using Proposition B.6 and then check that $p(x)$ is positive at one of the endpoints.

The specific cases that require verification using Proposition B.6 are as follows:

- For $i = 1, 2$, $\frac{T_i(m,\sigma)}{m(m-1)} \leq \left(1 - \frac{\sigma}{n}\right)^{2n}$ for all integers $m$ and $n$ where $2 \leq m \leq n \leq 6$ and $\sigma \in (0, 0.6]$.

- $\frac{T_1(m,\sigma)}{m(m-1)} \leq \frac{1}{4}$ for all integers $m$ where $2 \leq m \leq 10$ and $\sigma \in [0.6, 0.8]$.

- $\frac{T_1(m,\sigma)}{m(m-1)} \leq \frac{1}{4}$ for all integers $m$ where $2 \leq m \leq 14$ and $\sigma \in [0.8, 1)$.

- $\frac{T_2(m,\sigma)}{m(m-1)} \leq \frac{1}{4}$ for all integers $m$ where $2 \leq m \leq 10$ and $\sigma \in [0.6, 1)$.

The following SymPy code can be used to verify these inequalities. Since the coefficients are all rational numbers, the computations are exact and free from floating-point errors.

```python
import sympy as sp

def sturm_sequence(f, x):
    f0 = sp.expand(f)
    f1 = sp.expand(f.diff(x))
    sturm_seq = [f0, f1]
    while sturm_seq[-1] != 0:
        f_prev = sturm_seq[-2]
        f_curr = sturm_seq[-1]
        q, r = sp.div(f_prev, f_curr)
        sturm_seq.append(-r)
    return sturm_seq

def sign_variations(seq):
    seq_nonzero = [val for val in seq if val != 0]
    if not seq_nonzero:
        return 0
    variations = 0
    for i in range(len(seq_nonzero) - 1):
        if (seq_nonzero[i] > 0 and seq_nonzero[i+1] < 0) or \
           (seq_nonzero[i] < 0 and seq_nonzero[i+1] > 0):
            variations += 1
    return variations

def count_real_roots(f, a, b, x):
    seq = sturm_sequence(f, x)
    vals_a = [poly.subs(x, a) for poly in seq]
    vals_b = [poly.subs(x, b) for poly in seq]
    return sign_variations(vals_a) - sign_variations(vals_b)

def tau(n, s):
    L = n - (n - 1) * s
    alpha = 0
    for i in range(n):
        alpha += (1 - L * s**i)**2

    C = sp.Matrix.zeros(n, n)
    for i in range(1, n + 1):
        for j in range(1, n + 1):
            if i < j:
                C[i - 1, j - 1] = -(1 - s) * s**(i - 1)
```

```python
            else:
                C[i - 1, j - 1] = (1 - s) * (s**(i - j) - s**(i - 1))

    beta = sum(C[i, j]**2 for i in range(n) for j in range(n))
    gamma = sum(1 - L * s**i for i in range(n))**2
    delta = sum((s**n - s**i)**2 for i in range(1, n + 1))
    t1 = sp.Rational(1, (n - 1))*(beta + delta) \
        - sp.Rational(1, n*(n - 1))*(alpha + gamma)
    t2 = sp.Rational(1, n*(n - 1))*((alpha + gamma) - (beta + delta))
    return t1, t2

def p(n, s):
    p1_coeffs = []
    p2_coeffs = []
    for k in range(7):
        if k == 0:
            p1_coeffs.append(n**2 - n)
            p2_coeffs.append(n**2 - 3*n + 2)
        else:
            if k % 2 == 1:
                p1_coeffs.append((n-1)*k - 2*n**2 - n + 3)
                p2_coeffs.append(-2*n**2 + 6*n - 4)
            else:
                p1_coeffs.append(-(n+1)*k + 2*n**2+ 2*n + 2)
                p2_coeffs.append(2*k + 2*n**2 - 6*n - 2)

    p1 = sum([coeff * s**i for i, coeff in enumerate(p1_coeffs)])
    p2 = sum([coeff * s**i for i, coeff in enumerate(p2_coeffs)])
    return p1, p2

s = sp.symbols('s')

tau_k = {}
for k in range(2, 15):
    tau_k[k] = tau(k, s)

for n in range(2, 7):
    rcd = (1 - sp.Rational(1, n)*s)**(2*n)
    inequality_verified = True
    for k in range(2, n+1):
        d1 = rcd - tau_k[k][0]
        d2 = rcd - tau_k[k][1]
        roots_1 = count_real_roots(d1, sp.Rational(0), sp.Rational(1), s)
        roots_2 = count_real_roots(d2, sp.Rational(0), sp.Rational(1), s)
        d1_end = d1.subs(s, sp.Rational(1))
        d2_end = d2.subs(s, sp.Rational(1))
        if not all([roots_1 == 0, roots_2 == 0, d1_end > 0, d2_end > 0]):
            inequality_verified = False

        assert all([roots_1 == 0, roots_2 == 0, d1_end > 0, d2_end > 0]), \
            f"Error: Inequality verification failed for n = {n}, k = {k}."

    if inequality_verified:
        print(f"Case 1, n = {n}: All inequalities verified.")
    else:
        print(f"Case 1, n = {n}: Inequality verification failed for some k.")

for m in range(2, 11):
    lhs = sp.Rational(1, m*(m-1)) * tau_k[m][0]
    rhs = sp.Rational(1, 4)
    diff = rhs - lhs
    diff_end = diff.subs(s, sp.Rational(4, 5))
    roots = count_real_roots(diff, sp.Rational(3, 5), sp.Rational(4, 5), s)
    assert roots == 0 and diff_end > 0, \
```

```
        f"Error: Inequality verification failed for Case 2, m = {m}."
    print(f"Case 2, m = {m}: Inequality verified.")

for m in range(2, 15):
    lhs = sp.Rational(1, m*(m-1)) * tau_k[m][0]
    rhs = sp.Rational(1, 4)
    diff = rhs - lhs
    diff_end = diff.subs(s, sp.Rational(1))
    roots = count_real_roots(rhs - lhs, sp.Rational(4, 5), sp.Rational(1), s)
    assert roots == 0 and diff_end > 0, \
        f"Error: Inequality verification failed for Case 3, m = {m}."
    print(f"Case 3, m = {m}: Inequality verified.")

for m in range(2, 11):
    lhs = sp.Rational(1, m*(m-1)) * tau_k[m][1]
    rhs = sp.Rational(1, 4)
    diff = rhs - lhs
    diff_end = diff.subs(s, sp.Rational(1))
    roots = count_real_roots(rhs - lhs, sp.Rational(3, 5), sp.Rational(1), s)
    assert roots == 0 and diff_end > 0, \
        f"Error: Inequality verification failed for Case 4, m = {m}."
    print(f"Case 4, m = {m}: Inequality verified.")
```

We also provide an example demonstrating the computation involved in applying Proposition B.6 to verify the inequality $\frac{1}{4} - \frac{T_2(m,\sigma)}{m(m-1)}$ on $(0.6, 1)$, when $m = 3$. Since

$$\frac{1}{4} - \frac{T_2(m,\sigma)}{m(m-1)} = -\frac{\sigma^6}{9} + \frac{2\sigma^5}{9} - \frac{\sigma^4}{9} - \frac{\sigma^2}{18} + \frac{\sigma}{9} + \frac{7}{36},$$

the Sturm sequence is

$$p_0(\sigma) = -\frac{\sigma^6}{9} + \frac{2\sigma^5}{9} - \frac{\sigma^4}{9} - \frac{\sigma^2}{18} + \frac{\sigma}{9} + \frac{7}{36}$$

$$p_1(\sigma) = -\frac{2\sigma^5}{3} + \frac{10\sigma^4}{9} - \frac{4\sigma^3}{9} - \frac{\sigma}{9} + \frac{1}{9}$$

$$p_2(\sigma) = -\frac{2\sigma^4}{81} + \frac{2\sigma^3}{81} + \frac{\sigma^2}{27} - \frac{7\sigma}{81} - \frac{65}{324}$$

$$p_3(\sigma) = \sigma^3 - 3\sigma^2 - \frac{15\sigma}{4} + \frac{7}{2}$$

$$p_4(\sigma) = \frac{11\sigma^2}{54} + \frac{5\sigma}{27} + \frac{1}{36}$$

$$p_5(\sigma) = \frac{161\sigma}{484} - \frac{488}{121}$$

$$p_6(\sigma) = -\frac{10021099}{311052}$$

$$p_7(\sigma) = 0.$$

Therefore, we have

$$p_0(0.6) = \frac{134329}{562500}, \qquad p_0(1) = \frac{1}{4},$$

$$p_1(0.6) = \frac{1142}{28125}, \qquad p_1(1) = 0,$$

$$p_2(0.6) = -\frac{47993}{202500}, \qquad p_2(1) = -\frac{1}{4},$$

$$p_3(0.6) = \frac{193}{500}, \qquad p_3(1) = -\frac{9}{4},$$

$$p_4(0.6) = \frac{191}{900}, \qquad p_4(1) = \frac{5}{12},$$

$$p_5(0.6) = -\frac{9277}{2420}, \qquad p_5(1) = -\frac{1791}{484},$$

$$p_6(0.6) = -\frac{10021099}{311052}, \qquad p_6(1) = -\frac{10021099}{311052},$$

$$p_7(0.6) = 0, \qquad p_7(1) = 0.$$

We now examine the number of variations in sign for each sequence. Recall that for a sequence $c = (c_1, c_2, \ldots, c_m)$ of real numbers, the number of variations in sign is defined as the number of sign changes in the subsequence obtained by removing any zeros from $c$.

For the sequence evaluated at $\sigma = 0.6$, we observe the signs are given by $(+, +, -, +, +, -, -, 0)$. Removing the zero, the signs of the subsequence are $(+, +, -, +, +, -, -)$. Thus, there are three variations in sign.

Similarly, for the sequence evaluated at $\sigma = 1$, we observe the signs are $(+, 0, -, -, +, -, -, 0)$. Removing the zeros, the signs of the subsequence are $(+, -, -, +, -, -)$. This also has three variations in sign.

Applying Proposition B.6, the number of distinct real roots of $p_0(\sigma)$ in the interval $(0.6, 1)$ is given by $V_{0.6} - V_1 = 3 - 3 = 0$, where $V_a$ denotes the number of variations in sign of the sequence $(p_0(a), p_1(a), \ldots, p_7(a))$. Thus, we conclude that $p_0(\sigma)$ has no roots in the interval $(0.6, 1)$.

## D. Comparison with Previous Work

In this section, we provide a detailed comparison of previous work by Lee & Wright (2019), continuing from the discussion at the end of Section 3. Recall that Lee & Wright (2019) established that, for quadratic functions with permutation-invariant Hessians, the epoch-wise contraction ratio of RPCD is of order

$$1 - 2\sigma - \frac{2\sigma}{n} + 2\sigma^2 + \mathcal{O}\left(\frac{\sigma^2}{n}\right) + \mathcal{O}\left(\sigma^3\right) \tag{36}$$

when $n \geq 10$ and $\sigma \in (0, 0.4]$. In comparison, Theorem 3.3 provides an upper bound of $\max\left\{\left(1 - \frac{\sigma}{n}\right)^{2n}, \left(1 - \frac{1}{n}\right)^n\right\}$, which simplifies to $\left(1 - \frac{\sigma}{n}\right)^{2n}$ in the region $\sigma \in (0, 0.4]$.

As noted earlier, we obtain

$$\left(1 - \frac{\sigma}{n}\right)^{2n} = 1 - 2\sigma - \frac{\sigma^2}{n} + 2\sigma^2 + \mathcal{O}\left(\sigma^3\right).$$

This expansion initially appears to have extra terms compared to Equation (36). However, a closer examination of our proof methodology reveals a more refined upper bound.

We exploit the inequality $\rho(\boldsymbol{M}) \leq \|\boldsymbol{M}\|_\infty$, where $\boldsymbol{M}$ is the iteration matrix. In our analysis, we define the iteration matrix as

$$\boldsymbol{M_A} = \frac{1}{n(n-1)} \begin{bmatrix} n\beta - \alpha & n\delta - \gamma \\ \alpha - \beta & \gamma - \delta \end{bmatrix},$$

where $\alpha, \beta, \gamma$, and $\delta$ are polynomials in $\sigma$ (see (17) in Appendix B for their explicit definitions).

Lemma B.3 establishes that all entries of $\boldsymbol{M_A}$ are non-negative. Consequently, $\|\boldsymbol{M_A}\|_\infty$ simplifies to the maximum of the sums of the first and second rows of $\boldsymbol{M_A}$, both of which are polynomials in $\sigma$. We define $T_1$ and $T_2$ as the sum of the first and second rows of $\begin{bmatrix} n\beta - \alpha & n\delta - \gamma \\ \alpha - \beta & \gamma - \delta \end{bmatrix}$, respectively.

Importantly, our proof explicitly determines the coefficients of the polynomials $T_1$ and $T_2$ in terms of $n$. Furthermore, (21) and (23) establish that the even-degree Taylor approximations of $T_1$ and $T_2$ serve as upper bounds. Specifically, if we consider the 4-th order Taylor approximations of $T_1$ and $T_2$, we have

$$\frac{T_1}{n(n-1)} \leq 1 - \left(2 + \frac{1}{n}\right)\sigma + \frac{2n}{n-1}\sigma^2 - 2\sigma^3 + \frac{2(n^2 - n - 1)}{n(n-1)}\sigma^4$$

$$\frac{T_2}{n(n-1)} \leq 1 - \frac{2}{n} - 2\left(1 - \frac{2}{n}\right)\sigma + \frac{2(n^2 - 3n + 1)}{n(n-1)}\sigma^2 - 2\left(1 - \frac{2}{n}\right)\sigma^3 + \frac{2(n^2 - 3n + 3)}{n(n-1)}\sigma^4.$$

Remarkably, the upper bound for $\frac{T_1}{n(n-1)}$ precisely matches the contraction ratio given in Equation (36). While the upper bound for $\frac{T_2}{n(n-1)}$ does not directly match this expression, we have

$$\frac{T_2}{n(n-1)} \leq 1 - \frac{2}{n} - 2\sigma + \frac{4}{n}\sigma + 2\sigma^2 - \frac{4}{n}\sigma^2 + \mathcal{O}\left(\sigma^3\right)$$

$$\leq 1 - 2\sigma - \frac{2}{n}\sigma + 2\sigma^2 + \frac{\sigma^2}{n} + \mathcal{O}\left(\sigma^3\right),$$

where the last inequality follows from the fact that $2 - 6\sigma + 5\sigma^2 \geq 0$.

Therefore, we conclude that

$$\rho(\boldsymbol{M_A}) \leq \|\boldsymbol{M_A}\|_\infty$$

$$= \max_{i=1,2} \frac{T_i}{n(n-1)}$$

$$= 1 - 2\sigma - \frac{2\sigma}{n} + 2\sigma^2 + \mathcal{O}\left(\frac{\sigma^2}{n}\right) + \mathcal{O}\left(\sigma^3\right).$$

# E. Further Discussions: RPCD on General Quadratics

Here we continue from and elaborate on our discussions in Section 4.3.

## E.1. Partially Invariant Hessians.

Let us revisit the $4 \times 4$ unit-diagonal matrix[4]:

$$A = \begin{bmatrix} 1 & a & a & a \\ a & 1 & b & b \\ a & b & 1 & b \\ a & b & b & 1 \end{bmatrix}. \tag{37}$$

As stated in Section 4.3, we can observe that there are only 4 possible cases of matrices $P^\top A P$:

$$\begin{bmatrix} 1 & a & a & a \\ a & 1 & b & b \\ a & b & 1 & b \\ a & b & b & 1 \end{bmatrix}, \begin{bmatrix} 1 & a & b & b \\ a & 1 & a & a \\ b & a & 1 & b \\ b & a & b & 1 \end{bmatrix}, \begin{bmatrix} 1 & b & a & b \\ b & 1 & a & b \\ a & a & 1 & a \\ b & b & a & 1 \end{bmatrix}, \begin{bmatrix} 1 & b & b & a \\ b & b & 1 & a \\ b & b & b & a \\ a & a & a & 1 \end{bmatrix},$$

one for each $i \in [4]$ with unit diagonals, $a$ in the non-diagonal entries at the $i$-th row and column, and $b$ for the rest of the entries. We can also think of a similar formulation where $A$ is an $n \times n$ unit-diagonal matrix with the non-diagonal elements of the first row and column filled with $a$'s and the rest of the elements filled with $b$'s:

$$A = \begin{bmatrix} 1 & a\mathbf{1}^\top \\ a\mathbf{1} & (1-b)I + b\mathbf{1}\mathbf{1}^\top \end{bmatrix}. \tag{38}$$

Note that the upper diagonal block is a scalar, the lower diagonal block is a $(n-1) \times (n-1)$, permutation-invariant matrix, and the off-diagonal blocks are constant-filled $(n-1)$-dimensional vectors.

We define the bases using the following block matrices with the same dimensions:

$$V_1 = \begin{bmatrix} 1 & \mathbf{0} \\ \mathbf{0} & \mathbf{0} \end{bmatrix}, \quad V_2 = \begin{bmatrix} 0 & \mathbf{1}^\top \\ \mathbf{1} & \mathbf{0} \end{bmatrix}, \quad V_3 = \begin{bmatrix} 0 & \mathbf{0} \\ \mathbf{0} & I \end{bmatrix}, \quad V_4 = \begin{bmatrix} 0 & \mathbf{0} \\ \mathbf{0} & \mathbf{1}\mathbf{1}^\top \end{bmatrix}.$$

**Lemma E.1.** *If $A$ is defined as in* (38), *then* $\mathcal{M}_A^{\mathrm{RPCD}}$ *is closed in* $\mathcal{S}' = \mathrm{span}\{V_1, V_2, V_3, V_4\}$.

*Proof.* First, we observe that we can define

$$C_P = I - \Gamma_P^{-1} P^\top A P = I - \mathrm{tril}(P^\top A P)^{-1} P^\top A P$$

so that $T_{A,p}^{\mathrm{RPCD}} = P C_P P^\top$. Suppose we have any $X \in \mathcal{S}'$. Then we can write

$$\mathcal{M}_A^{\mathrm{RPCD}}(X) = \mathbb{E}\left[ P C_P^\top P^\top X P C_P P^\top \right].$$

Now, we think of partitioning the set of all possible permutations into $n$ sets, denoted by $\Pi_i$ for $i \in [n]$, each containing the set of all permutations such that $p(i) = 1$. For any $i$, each permutation $p \in \Pi_i$ can be uniquely decomposed into two parts:

$$p = p' \circ p_{(1i)}$$

where $p_{(1i)}$ is a permutation that swaps 1 and $i$ and $p'$ is a permutation that satisfies $p(1) = 1$. For each $i \in [n]$ we have $(n-1)!$ different choices of $p'$ and hence $p$ as well. Let us use the same analogy between permutations $p$ and permutation matrices $P$ preserving the subscripts. Using this decomposition, for each *fixed* $i \in [n]$ we have

$$\begin{aligned} C_P &= I - \mathrm{tril}(P^\top A P)^{-1} P^\top A P \\ &= I - \mathrm{tril}(P_{(1i)}^\top P'^\top A P' P_{(1i)})^{-1} P_{(1i)}^\top P'^\top A P' P_{(1i)} \\ &= I - \mathrm{tril}(P_{(1i)}^\top A P_{(1i)})^{-1} P_{(1i)}^\top A P_{(1i)} \end{aligned}$$

---

[4]The *minimal* example would be a $3 \times 3$ version with two $a$'s and one $b$ below the diagonal, but we choose $4 \times 4$ for clearer illustration.

where we use our construction of $A \in \mathcal{S}'$ such that $A$ is $P'$-invariant. Thus, if we fix $i$, then $C_P$ is also a fixed (constant) matrix, allowing us to use the notation $C_P = C_i$ instead. Considering the expectation over $\Pi_i$ for fixed $i$, we have

$$
\mathbb{E}_{p \in \Pi_i} \left[ PC_P^\top P^\top XPC_P P^\top \right] = \mathbb{E}_{p \in \Pi_i} \left[ P'P_{(1i)} C_i^\top P_{(1i)}^\top P'^\top XP'P_{(1i)} C_i P_{(1i)}^\top P'^\top \right]
$$
$$
= \mathbb{E}_{p \in \Pi_i} \left[ P'P_{(1i)} C_i^\top P_{(1i)}^\top XP_{(1i)} C_i P_{(1i)}^\top P'^\top \right]
$$

where we also use the fact that $X \in \mathcal{S}'$ is $P'$-invariant. Therefore we can **(i)** use Lemma B.2 over the permutations $P'$ to conclude that the lower diagonal block is permutation-invariant, and **(ii)** also observe that the off-diagonal block parts are also filled with constants. (This is because the entries in $(1,2), \ldots, (1,n)$ of $P_{(1i)} C_i^\top P_{(1i)}^\top XP_{(1i)} C_i P_{(1i)}^\top$ will be averaged after taking expectations on $P'$, and the same holds for entries in $(2,1), \ldots, (n,1)$. Moreover, the two averaged values must be identical because the matrix $P_{(1i)} C_i^\top P_{(1i)}^\top XP_{(1i)} C_i P_{(1i)}^\top$ is symmetric.)

Therefore we can conclude that for all $i \in [n]$,

$$
\mathbb{E}_{p \in \Pi_i} \left[ PC_P^\top P^\top XPC_P P^\top \right] = \mathbb{E}_{p \in \Pi_i} \left[ P'P_{(1i)} C_i^\top P_{(1i)}^\top XP_{(1i)} C_i P_{(1i)}^\top P'^\top \right] \in \mathcal{S}'.
$$

Taking one final average over $i \in [n]$, we have shown that

$$
\mathcal{M}_A^{\text{RPCD}}(X) = \mathbb{E} \left[ PC_P^\top P^\top XPC_P P^\top \right] \in \mathcal{S}'
$$

as desired. $\qquad \square$

As we have discussed in Section 4.3, we can see that Lemma E.1 allows us to reduce the problem into finding $\rho(\mathcal{M}_A^{\text{RPCD}}|_{\mathcal{S}'})$, which can be written as a $4 \times 4$ matrix using the subspace $\mathcal{S}'$ as the basis. Note that the remaining steps will require a relatively complicated analysis of an asymmetric $4 \times 4$ matrix with two controllable variables $a, b \in [0, 1]$, compared to Theorem 3.3 where we use a $2 \times 2$ matrix with only one variable $\sigma$.

More generally, it is intuitive that a similar dimension-reduction argument can also apply to different block structures. For example, we can think of the following $4 \times 4$ matrix:

$$
A = \begin{bmatrix} 1 & a & c & c \\ a & 1 & c & c \\ c & c & 1 & b \\ c & c & b & 1 \end{bmatrix}. \tag{39}
$$

For this case, we instead have $6 = \frac{4!}{2!2!}$ types of matrices $P^\top AP$. Similar arguments are also always possible for various $n$ and various dimensions of blocks. In general cases, however, it is much more likely that we *cannot* find such a nice block structure that reduces permutation variance and thus a small-dimensional subspace $\mathcal{S}$ on which we can express $\mathcal{M}_A^{\text{RPCD}}|_{\mathcal{S}}$ as a smaller matrix.

### E.2. General Hessians: Matrix AM-GM Inequality

Continuing from our discussion in the second paragraph of Section 4.3, the only results we are aware of that consider RPCD for general quadratics are those by Sun et al. (2020). In particular, their objective was to show that

$$
\rho(\mathbb{E}[T_A^{\text{RPCD}}]) \le 1 - \frac{\sigma}{n}.
$$

This is a direct corollary of a *weak* version of the well-known matrix AM-GM inequality (Recht & Re, 2012; Lai & Lim, 2020; De Sa, 2020; Yun et al., 2021). To elaborate, Sun et al. (2020) (Claim 4.1) observe that:

$$
A^{\frac{1}{2}} T_{A,p}^{\text{RPCD}} A^{-\frac{1}{2}} = I - A^{\frac{1}{2}} P\Gamma_P^{-1} P^\top A^{\frac{1}{2}} = Z_{p(n)} \cdots Z_{p(1)}, \tag{40}
$$

where $Z_i = I - v_i v_i^\top$ is a projection matrix, with $v_i$ being the $i$-th column of $A^{\frac{1}{2}}$. Then they show for the expectation:

$$
\frac{1}{n!} \sum_p Z_{p(n)} \cdots Z_{p(1)} \preceq \frac{1}{n} \sum_{i=1}^n Z_i \quad \left( = I - \frac{A}{n} \right),
$$

which is called *weak* because the RHS is missing the $n$-th exponent from the original matrix AM-GM,

$$\frac{1}{n!}\sum_p \boldsymbol{Z}_{p(n)}\cdots\boldsymbol{Z}_{p(1)} \preceq \left(\frac{1}{n}\sum_{i=1}^n \boldsymbol{Z}_i\right)^n.$$

To obtain a proper comparison with the RCD lower bound results in Theorem 3.1, it suffices to show an upper bound of

$$\rho(\mathcal{M}_{\boldsymbol{A}}^{\text{RPCD}}) = \rho(\mathbb{E}[\boldsymbol{T}_{\boldsymbol{A}}^{\text{RPCD}\top}\otimes\boldsymbol{T}_{\boldsymbol{A}}^{\text{RPCD}\top}]) \le \max\left\{\left(1-\frac{\sigma}{n}\right)^{2n}, \left(1-\frac{1}{n}\right)^n\right\},$$

which is not only stronger than the weak AM-GM, considering that we have an order $n$ exponent added to the upper bound, but also much more difficult in many aspects.

- Analyzing the *upper bounds* of quantities derived from a *sum of Kronecker power* of matrices is presumably the harder direction than the opposite (we typically use the sum of squares *as* an upper bound).

- Even for the case of two matrices $\boldsymbol{A}$ and $\boldsymbol{B}$, the relationship between the spectrum of $\boldsymbol{A}+\boldsymbol{B}$ and each of the matrices $\boldsymbol{A}$ and $\boldsymbol{B}$ (beyond Weyl's inequality (Weyl, 1912)) is highly nontrivial (Knutson & Tao, 2000). In our case, we have a sum of $n!$ matrices, and a fine spectral analysis is extremely hard.

One of the few possible solutions to avoid computing the Kronecker powers could be to use the following lemma.

**Lemma E.2.** *We have*

$$\rho(\mathcal{M}_{\boldsymbol{A}}^{\text{RPCD}}) \le \left\|\boldsymbol{A}^{-1/2}\mathcal{M}_{\boldsymbol{A}}^{\text{RPCD}}(\boldsymbol{A})\boldsymbol{A}^{-1/2}\right\|. \tag{41}$$

*Proof.* The Russo-Dye theorem (Russo & Dye (1966), Theorem 2.3.7 of Bhatia (2007)) states that if $\mathcal{M}$ is a positive linear map,[5] then we have $\|\mathcal{M}\| = \|\mathcal{M}(\boldsymbol{I})\|$, where $\|\cdot\|$ is the operator norm induced by the matrix operator norm $\|\cdot\|$. For our case, we can easily check that $\mathcal{M}_{\boldsymbol{A}}^{\text{RPCD}}$ is positive as it outputs the expectation of positive semi-definite matrices. Moreover, we have defined a similar matrix operator earlier in Section 2:

$$\widetilde{\mathcal{M}}_{\boldsymbol{A}}^{\text{RPCD}}(\boldsymbol{X}) = \boldsymbol{A}^{-\frac{1}{2}}\mathcal{M}_{\boldsymbol{A}}^{\text{RPCD}}(\boldsymbol{A}^{\frac{1}{2}}\boldsymbol{X}\boldsymbol{A}^{\frac{1}{2}})\boldsymbol{A}^{-\frac{1}{2}},$$

which is also a positive linear map (with a symmetric matrix form). Using the Russo-Dye theorem on $\widetilde{\mathcal{M}}_{\boldsymbol{A}}^{\text{RPCD}}$, we have

$$\rho(\mathcal{M}_{\boldsymbol{A}}^{\text{RPCD}}) = \rho(\widetilde{\mathcal{M}}_{\boldsymbol{A}}^{\text{RPCD}}) \le \left\|\left\|\widetilde{\mathcal{M}}_{\boldsymbol{A}}^{\text{RPCD}}\right\|\right\| = \left\|\widetilde{\mathcal{M}}_{\boldsymbol{A}}^{\text{RPCD}}(\boldsymbol{I})\right\| = \left\|\boldsymbol{A}^{-\frac{1}{2}}\mathcal{M}_{\boldsymbol{A}}^{\text{RPCD}}(\boldsymbol{A})\boldsymbol{A}^{-\frac{1}{2}}\right\|$$

which completes the proof. $\qquad\square$

One caveat of the approach is that the RHS of (41) gets too large and exceeds the RCD lower bound for some values of $\sigma \in (\frac{1}{2}, 1)$ after $n$ gets sufficiently large, i.e., (41) is too loose for this case (as shown in Figure 4 in Section 4). The same bound still seems tight enough for the region $\sigma \in (0, \frac{1}{2}]$, where we also have $(1-\frac{\sigma}{n})^{2n} \ge (1-\frac{1}{n})^n$ and hence the upper bound simply reduces to $(1-\frac{\sigma}{n})^{2n}$.

From (40), we can observe the following:

$$\boldsymbol{A}^{-\frac{1}{2}}\mathcal{M}_{\boldsymbol{A}}^{\text{RPCD}}(\boldsymbol{A})\boldsymbol{A}^{-\frac{1}{2}} = \mathbb{E}\left[(\boldsymbol{I}-\boldsymbol{A}^{\frac{1}{2}}\boldsymbol{P}\boldsymbol{\Gamma}_{\boldsymbol{P}}^{-1}\boldsymbol{P}^\top\boldsymbol{A}^{\frac{1}{2}})^\top(\boldsymbol{I}-\boldsymbol{A}^{\frac{1}{2}}\boldsymbol{P}\boldsymbol{\Gamma}_{\boldsymbol{P}}^{-1}\boldsymbol{P}^\top\boldsymbol{A}^{\frac{1}{2}})\right]$$

$$= \frac{1}{n!}\sum_p (\boldsymbol{Z}_{p(n)}\cdots\boldsymbol{Z}_{p(1)})^\top(\boldsymbol{Z}_{p(n)}\cdots\boldsymbol{Z}_{p(1)}).$$

Then by (41), it suffices to show

$$\rho\left(\frac{1}{n!}\sum_p (\boldsymbol{Z}_{p(n)}\cdots\boldsymbol{Z}_{p(1)})^\top(\boldsymbol{Z}_{p(n)}\cdots\boldsymbol{Z}_{p(1)})\right) \le \rho\left(\frac{1}{n}\sum_{i=1}^n \boldsymbol{Z}_i\right)^{2n}$$

to prove Conjecture 4.1 on $\sigma \in (0, \frac{1}{2}]$, and we leave the proof/disproof for future work.

---

[5]A linear map $\mathbb{R}^{n\times n}\to\mathbb{R}^{k\times k}$ is *positive* if it maps positive semi-definite matrices into positive semi-definite matrices.

# F. Details on Experiments

## F.1. Algorithmic Search.

Here we provide detailed information about the algorithmic search used to find the worst-case instances for RPCD. Recall that we use the following framework to search for the worst-case example among the space of all (unit-diagonal) quadratics.

*Step 1.* Start with a (randomly initialized) lower triangular matrix $X \in \mathbb{R}^{n \times n}$ with nonzero diagonals.

*Step 2.* Construct a unit-diagonal, positive semi-definite matrix by computing $Y = X^\top X$ and set unit diagonals by $Z = D_Y^{-\frac{1}{2}} Y D_Y^{-\frac{1}{2}}$, where $D_Y$ is the diagonal part of $Y$.

*Step 3.* Construct a matrix $A$ with $\sigma = \lambda_{\min}(A)$ by

$$A = \frac{1-\sigma}{1-\mu} Z + \frac{\sigma-\mu}{1-\mu} I,$$

where $\mu = \lambda_{\min}(Z)$. Note that $A$ is also unit-diagonal and positive semi-definite.

*Step 4.* Construct an objective function that takes an input $X$ to compute the value of $\rho(\mathcal{M}_A^{\text{RPCD}})$ and run a `scipy` optimizer to maximize the objective.

**Experiment Settings.** In our search for worst-case matrices, we explored the performance of our algorithm for dimensions $n \in \{3, 4, 5, 6\}$ and for minimum eigenvalue $\sigma \in \{0.1, \ldots, 0.9\}$ (with increments of 0.1).

**Initialization.** To initialize the lower-triangular matrix $X$, we generated a vector of length $\frac{n(n+1)}{2}$ with elements drawn from a standard normal distribution and then used this vector to create an $n \times n$ lower-triangular matrix. We used 10 random initial vectors for $n = 3, 4$ and 2 vectors for $n = 5, 6$. The reduced number of initializations for $n = 5$ and $n = 6$ was due to the computational complexity of calculating the matrix representation of $\mathcal{M}_A^{\text{RPCD}}$ and its spectral radius, which is $\mathcal{O}(n^4 \cdot n!)$.

**Construction of $A$.** We computed $A = \frac{1-\sigma}{1-\mu} Z + \frac{\sigma-\mu}{1-\mu} I$, where $\mu = \lambda_{\min}(Z)$. This construction ensures that $A$ is a unit-diagonal matrix and that its eigenvalues are of the form $\frac{1-\sigma}{1-\mu}\lambda + \frac{\sigma-\mu}{1-\mu}$, where $\lambda$ is an eigenvalue of $Z$. Since $Z$ is a unit-diagonal positive semi-definite matrix, its minimum eigenvalue $\mu$ lies in $[0, 1]$, and thus $\frac{1-\sigma}{1-\mu} \geq 0$. This guarantees that the minimum eigenvalue of $A$ is $\frac{1-\sigma}{1-\mu}\mu + \frac{\sigma-\mu}{1-\mu} = \sigma$.

**Optimization.** We used the BFGS method in the `scipy.optimize.minimize` function with default parameters as the optimization algorithm to minimize the function $-\rho(\mathcal{M}_A^{\text{RPCD}})$. To calculate the spectral radius of $\mathcal{M}_A^{\text{RPCD}}$, we utilized its matrix representation, which is given by $\mathbb{E}\left[(T_{A,p}^{\text{RPCD}})^\top \otimes (T_{A,p}^{\text{RPCD}})^\top\right]$. Here, $T_{A,p}^{\text{RPCD}}$ is defined in Equation (5).

**Results.** We present a few examples of the optimized matrices obtained through **Algorithmic Search**. Specifically, for each combination of $n \in \{3, 4, 5, 6\}$ with $\sigma \in \{0.3, 0.7\}$, we selected the matrix that achieved the largest $\rho(\mathcal{M}_A^{\text{RPCD}})$ among the trials. These examples demonstrate that the optimization process converges to matrices in $\mathcal{A}_\sigma$ with various sign flips. Here, $A_{n,\sigma}$ represents the matrix $\sigma I_n + (1-\sigma)\mathbf{1}_n\mathbf{1}_n^\top \in \mathcal{A}_\sigma^{\text{PI}}$. Note that $A_{n,\sigma} \odot vv^\top \in \mathcal{A}_\sigma$ when $v \in \{\pm 1\}^n$.

$$A_{n,\sigma} \odot vv^\top + \Delta, \quad v = \begin{bmatrix} 1 & 1 & 1 \end{bmatrix}^\top, \qquad \|\Delta\|_F \approx 1.2 \cdot 10^{-10} \quad \text{for } n = 3, \sigma = 0.3$$

$$A_{n,\sigma} \odot vv^\top + \Delta, \quad v = \begin{bmatrix} 1 & -1 & -1 \end{bmatrix}^\top, \qquad \|\Delta\|_F \approx 8.7 \cdot 10^{-11} \quad \text{for } n = 3, \sigma = 0.7$$

$$A_{n,\sigma} \odot vv^\top + \Delta, \quad v = \begin{bmatrix} 1 & 1 & -1 & 1 \end{bmatrix}^\top, \qquad \|\Delta\|_F \approx 2.7 \cdot 10^{-10} \quad \text{for } n = 4, \sigma = 0.3$$

$$A_{n,\sigma} \odot vv^\top + \Delta, \quad v = \begin{bmatrix} 1 & -1 & -1 & 1 \end{bmatrix}^\top, \qquad \|\Delta\|_F \approx 2.5 \cdot 10^{-10} \quad \text{for } n = 4, \sigma = 0.7$$

$$A_{n,\sigma} \odot vv^\top + \Delta, \quad v = \begin{bmatrix} 1 & -1 & 1 & -1 & 1 \end{bmatrix}^\top, \qquad \|\Delta\|_F \approx 1.1 \cdot 10^{-8} \quad \text{for } n = 5, \sigma = 0.3$$

$$A_{n,\sigma} \odot vv^\top + \Delta, \quad v = \begin{bmatrix} 1 & -1 & 1 & -1 & 1 \end{bmatrix}^\top, \qquad \|\Delta\|_F \approx 9.3 \cdot 10^{-10} \quad \text{for } n = 5, \sigma = 0.7$$

$$A_{n,\sigma} \odot vv^\top + \Delta, \quad v = \begin{bmatrix} 1 & 1 & -1 & 1 & 1 & -1 \end{bmatrix}^\top, \qquad \|\Delta\|_F \approx 1.9 \cdot 10^{-4} \quad \text{for } n = 6, \sigma = 0.3$$

$$A_{n,\sigma} \odot vv^\top + \Delta, \quad v = \begin{bmatrix} 1 & -1 & 1 & 1 & -1 & -1 \end{bmatrix}^\top, \qquad \|\Delta\|_F \approx 9.4 \cdot 10^{-7} \quad \text{for } n = 6, \sigma = 0.7$$

Furthermore, we empirically observed that for any optimized matrix $\widehat{A}$, it holds that $\rho(\mathcal{M}_{\widehat{A}}^{\text{RPCD}}) \leq \max_{A \in \mathcal{A}_\sigma} \rho(\mathcal{M}_A^{\text{RPCD}})$.

The following Python code implements **Algorithmic Search**, which also empirically validates that $\rho(\mathcal{M}_{\widehat{A}}^{\text{RPCD}}) \leq \max_{A \in \mathcal{A}_\sigma} \rho(\mathcal{M}_A^{\text{RPCD}})$ for all optimized $\widehat{A}$.

```python
import numpy as np
from scipy.optimize import minimize
from itertools import permutations
from numpy.linalg import *
from math import factorial

def rho(M):
    return max(abs(eig(M)[0]))

def unit_diag(M):
    diag_sqrt = np.sqrt(np.diag(M))
    return M / (diag_sqrt * diag_sqrt[:, None])

def mat_rep_rpcd(A):
    n = A.shape[0]
    result = np.zeros((n*n, n*n))
    for perm in list(permutations(np.arange(n))):
        P = np.eye(n)[np.array(perm)]
        Ap = P.T@A@P
        Gp = np.tril(Ap)
        Gxp = P@Gp@P.T
        temp = (np.eye(n)-inv(Gxp)@A).T
        result += np.kron(temp, temp)
    return result/factorial(n)

def rpcd_pi_rho(n, s):
    L = n - (n - 1) * s
    if s != 1.:
        alpha = n - 2 * L * (1 - s**n) / (1 - s) + L**2 * (1 - s**(2*n)) / (1 - s**2)
        beta = ((1 - s) / (1 + s)) * (2 * n - n * s**(2*n) \
            - 2 * (1 - s**(n + 1)) * (1 - s**n) / (1 - s) \
            - s**2 * (1 - s**(2*n)) / (1 - s**2))
        gamma = (1 - 1/(1 - s) + (n - 1 + 1/(1 - s)) * s**n)**2
        delta = n * s**(2 * n) - 2 * s**(n + 1) * (1 - s**n) / (1 - s) \
            + s**2 * (1 - s**(2 * n)) / (1 - s**2)
    else:
        alpha = 0.
        beta = 0.
        gamma = 0.
        delta = 0.
    M = np.array([[n*beta - alpha, n*delta - gamma], \
        [alpha - beta, gamma - delta]])/(n*(n-1))
    return rho(M)

def set_min_eigval(A, s):
    n = A.shape[0]
    mu = min(eigvals(A))
    a = (1-s)/(1-mu)
    b = (s-mu)/(1-mu)
    B = a*A + b*np.eye(n)
    return B

def X_to_A(X, n, s):
    if len(X) != int(n*(n+1)/2):
        raise ValueError("X must have length n(n+1)/2.")
    L = np.zeros((n, n)).astype(float)
    idx = 0
    for i in range(n):
        for j in range(i + 1):
            L[i, j] = X[idx]
            idx += 1
```

```python
    Y = L.T@L
    Z = unit_diag(Y)
    A = set_min_eigval(Z, s)
    return A

#### Optimization ####

def objective(X, n, s):
    A = X_to_A(X, n, s)
    return -rho(mat_rep_rpcd(A))

n_values = [3, 4, 5, 6]
s_values = np.linspace(0.1, 0.9, 9)

sim_num_list = [10, 10, 2, 2]
results = []

# Compute max rho(M_A^RPCD), for A in A_sigma
rho_max_dict = {}
for n in range(n_values):
    for s in s_values:
        temp = []
        for k in range(2, n+1):
            Aks = s*np.eye(k) + (1-s)*np.ones((k, k))
            temp.append(rho(mat_rep_rpcd(Aks)))
        rho_max_dict[(n, s)] = max(temp)

for i, n in enumerate(n_values):
    for s in s_values:
        sim_num = sim_num_list[i]
        for sim in range(sim_num):
            values = []
            X_init = np.random.randn(int(n*(n+1)/2))
            result = minimize(objective, X_init, args=(n, s), method='BFGS')
            X = result.x
            res_A = X_to_A(X, n, s)
            results.append({'n': n, 's': s, 'sim': sim+1, 'A': res_A})
            assert rho(mat_rep_rpcd(res_A)) <= rho_max_dict[(n, s)], f"Counterexample"
```

### F.2. RCD vs RPCD.

**Problem Settings.** We considered four types of convex and smooth optimization problems in $n$-dimensional space. The first three share a common structure of the form

$$f(\boldsymbol{x}) = \frac{1}{2}\boldsymbol{x}^\top \boldsymbol{A}\boldsymbol{x} + \alpha \cdot \mathrm{LSE}(\boldsymbol{Q}\boldsymbol{x}),$$

where LSE is the log-sum-exp function, *i.e.*, $\mathrm{LSE}(a_1, \ldots, a_n) = \log(e^{a_1} + \cdots + e^{a_n})$. These problems differ in the specific construction of $\boldsymbol{A}$ and the presence of the LSE term:

- **(i)** Quadratic functions with permutation-invariant Hessians: $\boldsymbol{A} = \sigma \boldsymbol{I}_n + (1-\sigma)\mathbf{1}_n\mathbf{1}_n^\top$ with $\sigma \in (0, 1]$ and $\alpha = 0$.

- **(ii)** General quadratic functions with positive definite Hessians: $\boldsymbol{A}$ was randomly generated with unit diagonals and $\lambda_{\min}(\boldsymbol{A}) = \sigma \in (0, 1]$, and $\alpha = 0$.

- **(iii)** General quadratic functions with positive definite Hessians and a scaled LSE term: we used the same $\boldsymbol{A}$ as in **(ii)** with an additional term $\alpha \cdot \mathrm{LSE}(\boldsymbol{Q}\boldsymbol{x})$, where $\boldsymbol{Q}$ is a randomly generated orthogonal matrix.

We further consider a logistic regression problem with ridge regularization.

- **(iv)** $\ell_2$-regularized logistic regression: the objective is

$$\min_x \frac{1}{m} \sum_{i=1}^{m} \log(1 + \exp(-b_i \boldsymbol{a}_i^\top \boldsymbol{x})) + \frac{\lambda}{2} \|\boldsymbol{x}\|^2,$$

where each $\boldsymbol{a}_i \in \mathbb{R}^n$ is drawn from $\mathcal{N}(0, \boldsymbol{I}_n)$. We sample a ground-truth vector $\boldsymbol{x}_{\text{true}} \sim \mathcal{N}(0, \boldsymbol{I}_n)$, set $b_i = \text{sign}(\boldsymbol{a}_i^\top \boldsymbol{x}_{\text{true}})$, and flip each $b_i$ independently with probability $0.1$. This setup follows Nutini et al. (2015).

**Matrix Generation.** For **(ii)** and **(iii)**, the matrix $\boldsymbol{A}$ was constructed by first generating an $n \times n$ random matrix $\boldsymbol{X}$ with entries drawn from the standard normal distribution, followed by *Steps 2-3* of **Algorithmic Search** detailed in Appendix F.1. This procedure ensures that the generated $\boldsymbol{A}$ is unit-diagonal and has a minimum eigenvalue $\sigma$. The orthogonal matrix $\boldsymbol{Q}$ in the LSE term was generated by applying QR decomposition to a random matrix with entries drawn from the standard normal distribution.

**Initialization.** For each parameter setting in problems **(i)-(iv)** (i.e., each combination of $n$, $\sigma$, $\alpha$, or $\lambda$), we sampled 10 initial points from the standard normal distribution. For each initial point, both RCD and RPCD were run 10 times, resulting in 100 runs per setting.

**Parameter Settings.** We performed experiments for dimension $n \in \{25, 50\}$ and minimum eigenvalue $\sigma \in \{0.1, \ldots, 0.9\}$ (with increments of $0.1$). For **(iii)**, we considered the scaling factor $\alpha \in \{0.5, 2.0\}$. For each algorithm execution, we ran 200 iterations when $n = 25$ and 300 iterations when $n = 50$. For **(iv)**, we set $n = m = 100$, $\lambda \in \{0.0001, 0.001, 0.01, 0.1, 1.0\}$, and ran 600 iterations.

**Implementation Details.** For the implementation of RCD and RPCD, we computed the $\arg\min$ for the objective function at each coordinate, as described in Algorithm 1. For **(i)** and **(ii)**, the update rule can be computed in closed form:

$$\boldsymbol{x}_{t+1} = \boldsymbol{x}_t - \frac{1}{a_{i_t i_t}} \boldsymbol{E}_{i_t} \boldsymbol{A} \boldsymbol{x}_t.$$

This update rule is derived from the CD update formula for quadratic objectives, as remarked in Section 2.2. For **(iii)** and **(iv)**, we used `scipy.optimize.minimize_scalar` with default parameters to compute the coordinate-wise updates.

**Measurement.** To evaluate the convergence, we measured $\frac{\|\boldsymbol{x}_k - \boldsymbol{x}^\star\|}{\|\boldsymbol{x}_0 - \boldsymbol{x}^\star\|}$ as our metric. This metric is motivated by Theorems 3.1 and 3.3, which provide convergence bounds in terms of the norm of the iterates. For **(i)** and **(ii)**, the optimal solution is known to be $\boldsymbol{x}^\star = \boldsymbol{0}$, simplifying the metric to $\frac{\|\boldsymbol{x}_k\|}{\|\boldsymbol{x}_0\|}$. For **(iii)** and **(iv)**, we used the BFGS method implemented in `scipy.optimize.minimize` with custom parameters (`gtol=1e-12, maxiter=10000`) to ensure a more accurate approximate optimal solution $\boldsymbol{x}^\star$ before running the RCD and RPCD algorithms.

**Results.** Figures 5-15 illustrate the convergence behavior of RCD and RPCD. Each algorithm was executed 100 times (10 random initial points and 10 runs per initial point) for each combination of $(n, \sigma)$. The plots depict the ratio $\frac{\|\boldsymbol{x}_k - \boldsymbol{x}^\star\|}{\|\boldsymbol{x}_0 - \boldsymbol{x}^\star\|}$ on a log scale. In the plots, solid lines represent the average of the convergence values over all trials at each iteration, and shaded regions indicate the min-max range of the convergence values across all trials at each iteration. The results demonstrate that RPCD consistently converges faster than RCD. For small values of $\sigma$ (e.g., $\sigma = 0.1$), the performance of RCD and RPCD is similar in **(i)**, with RPCD showing only a slight speed advantage. However, for the same small $\sigma$, the performance gap between RCD and RPCD becomes noticeably larger in **(ii)**. This supports Conjecture 4.1 that RPCD is faster than RCD for general quadratic functions and provides additional evidence that **(i)** represents a worst-case instance for RPCD.

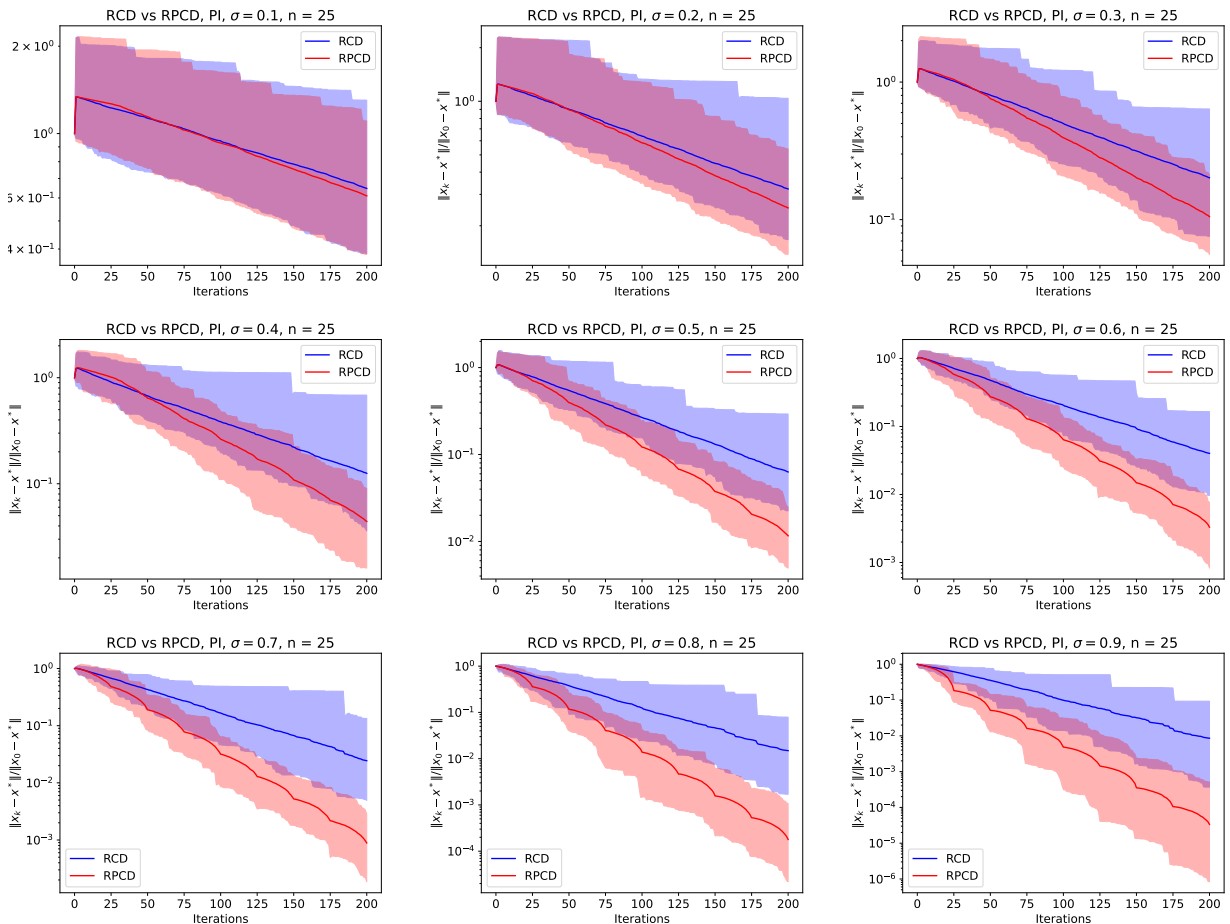

*Figure 5.* RCD vs RPCD, permutation-invariant quadratic functions. $\boldsymbol{A} = \sigma \boldsymbol{I} + (1 - \sigma)\boldsymbol{1}\boldsymbol{1}^{\top}, n = 25, \sigma \in \{0.1, \dots, 0.9\}$. The $y$-axis is in log scale.

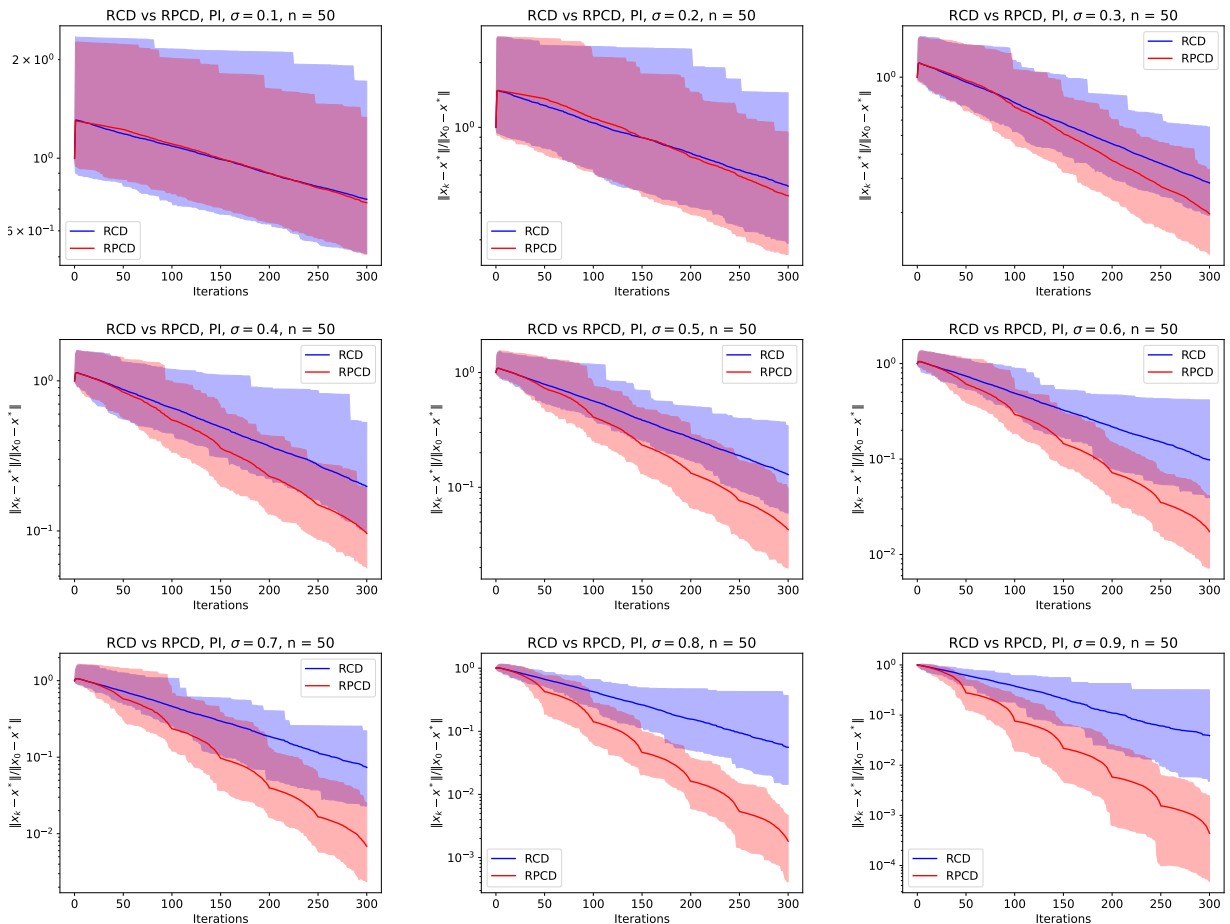

*Figure 6.* RCD vs RPCD, permutation-invariant quadratic functions. $\boldsymbol{A} = \sigma\boldsymbol{I} + (1 - \sigma)\boldsymbol{1}\boldsymbol{1}^\top$, $n = 50$, $\sigma \in \{0.1, \ldots, 0.9\}$. The $y$-axis is in log scale.

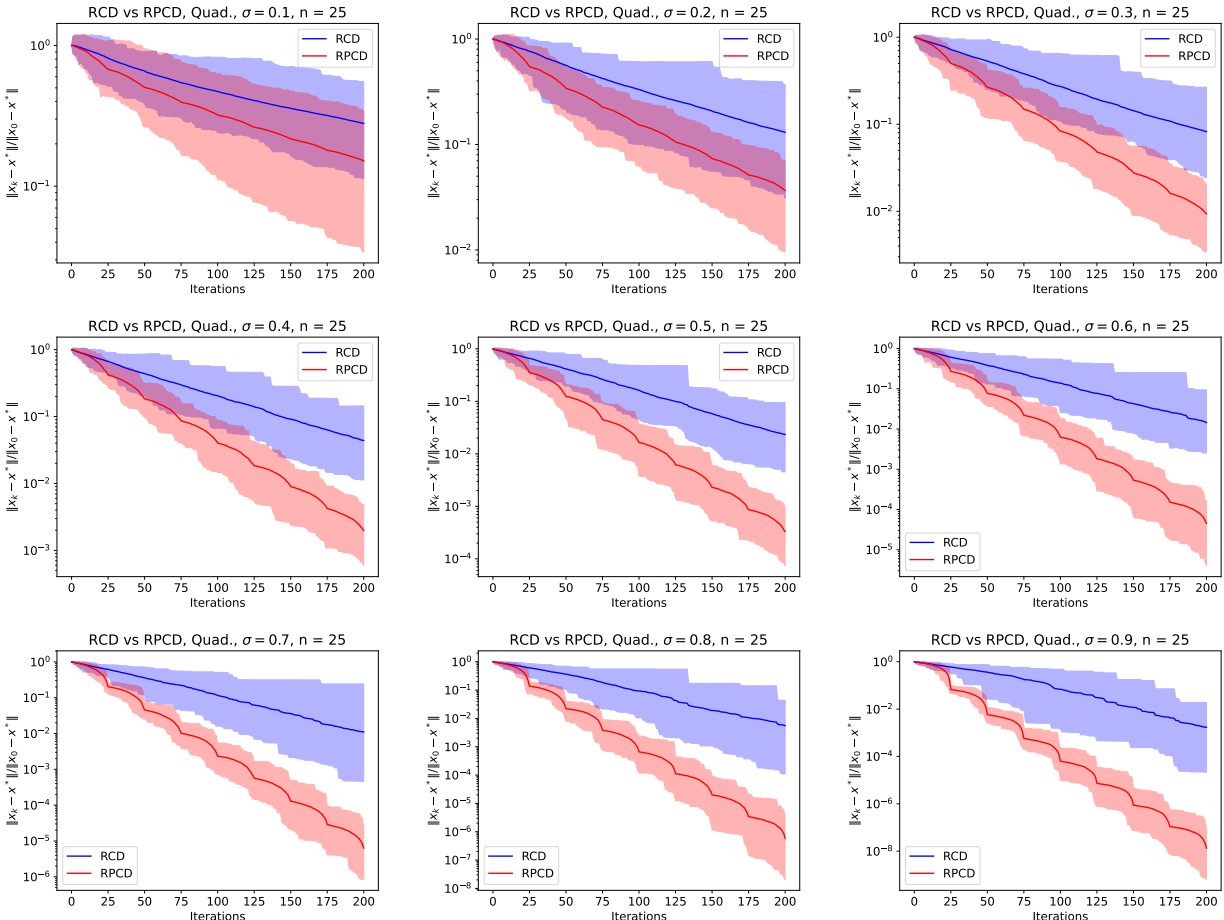

*Figure 7.* RCD vs RPCD, quadratic functions with unit-diagonal Hessians. $\boldsymbol{A}$ is unit-diagonal and $\lambda_{\min}(\boldsymbol{A}) = \sigma$, $n = 25, \sigma \in \{0.1, \dots, 0.9\}$. The $y$-axis is in log scale.

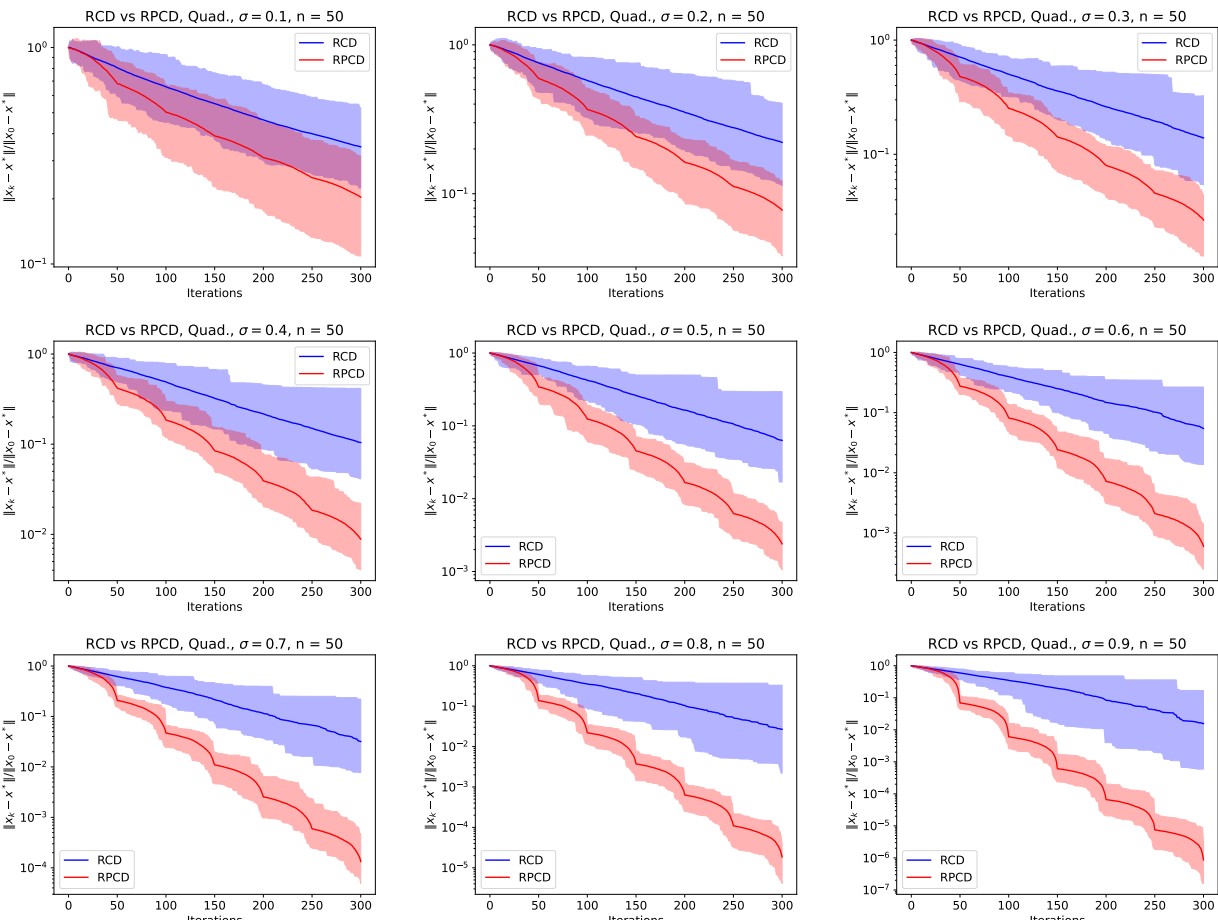

*Figure 8.* RCD vs RPCD, quadratic functions with unit-diagonal Hessians. $\boldsymbol{A}$ is unit-diagonal and $\lambda_{\min}(\boldsymbol{A}) = \sigma$, $n = 50, \sigma \in \{0.1, \ldots, 0.9\}$. The $y$-axis is in log scale.

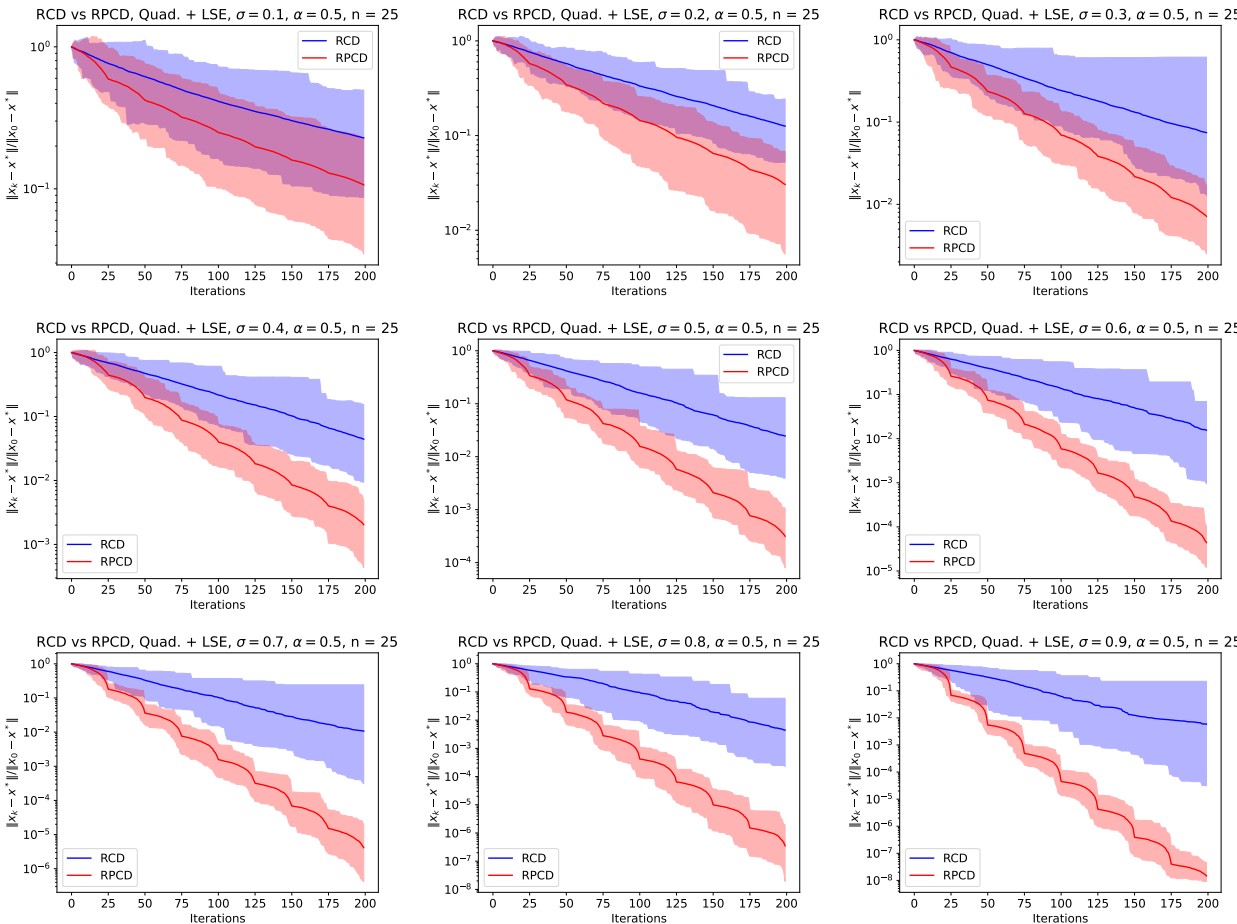

*Figure 9.* RCD vs RPCD, quadratic functions with unit-diagonal Hessians and log-sum-exp term. $\boldsymbol{A}$ is unit-diagonal and $\lambda_{\min}(\boldsymbol{A}) = \sigma$, $n = 25, \sigma \in \{0.1, \ldots, 0.9\}, \alpha = 0.5$. The $y$-axis is in log scale.

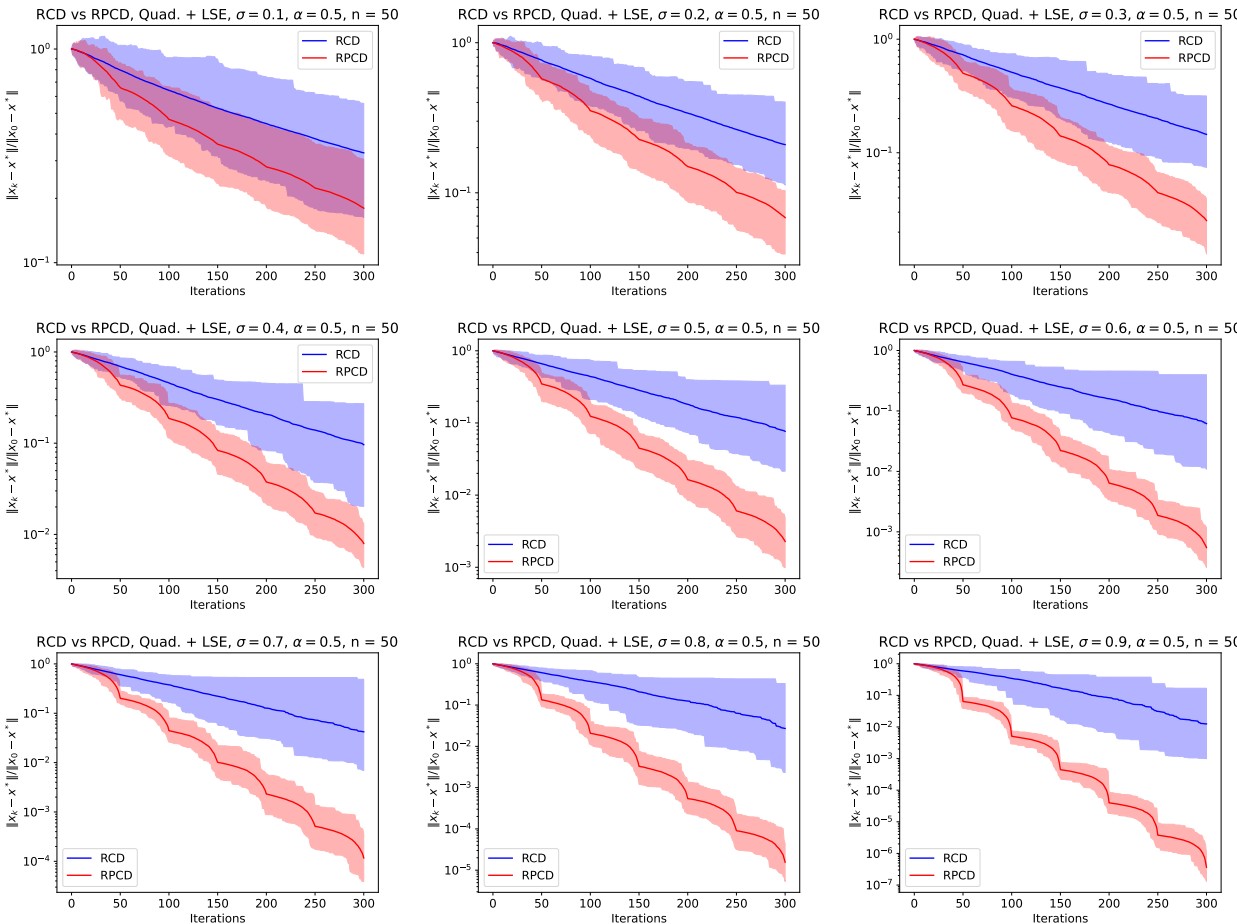

*Figure 10.* RCD vs RPCD, quadratic functions with unit-diagonal Hessians and log-sum-exp term. $\boldsymbol{A}$ is unit-diagonal and $\lambda_{\min}(\boldsymbol{A}) = \sigma$, $n = 50, \sigma \in \{0.1, \ldots, 0.9\}, \alpha = 0.5$. The $y$-axis is in log scale.

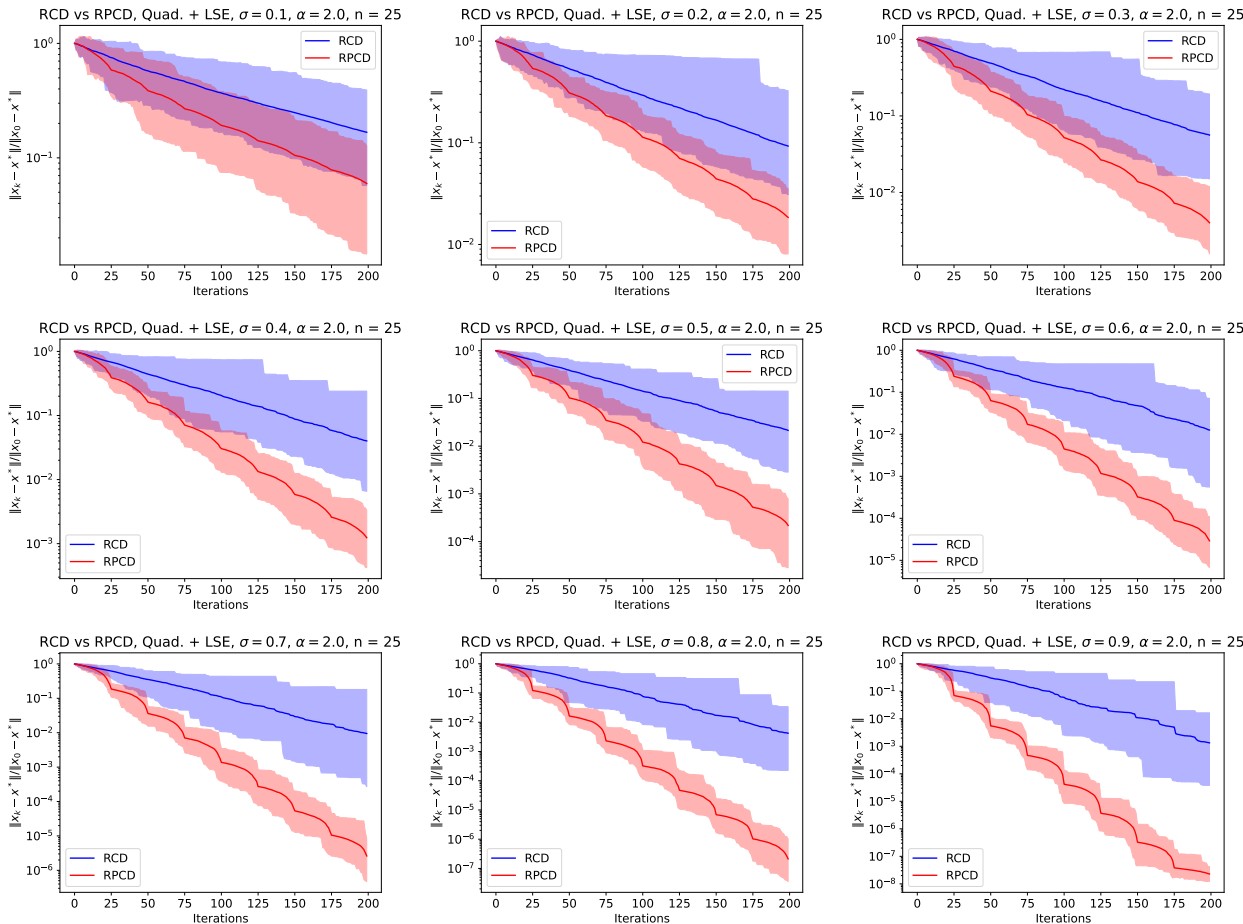

*Figure 11.* RCD vs RPCD, quadratic functions with unit-diagonal Hessians and log-sum-exp term. $\boldsymbol{A}$ is unit-diagonal and $\lambda_{\min}(\boldsymbol{A}) = \sigma$, $n = 25, \sigma \in \{0.1, \ldots, 0.9\}, \alpha = 2.0$. The $y$-axis is in log scale.

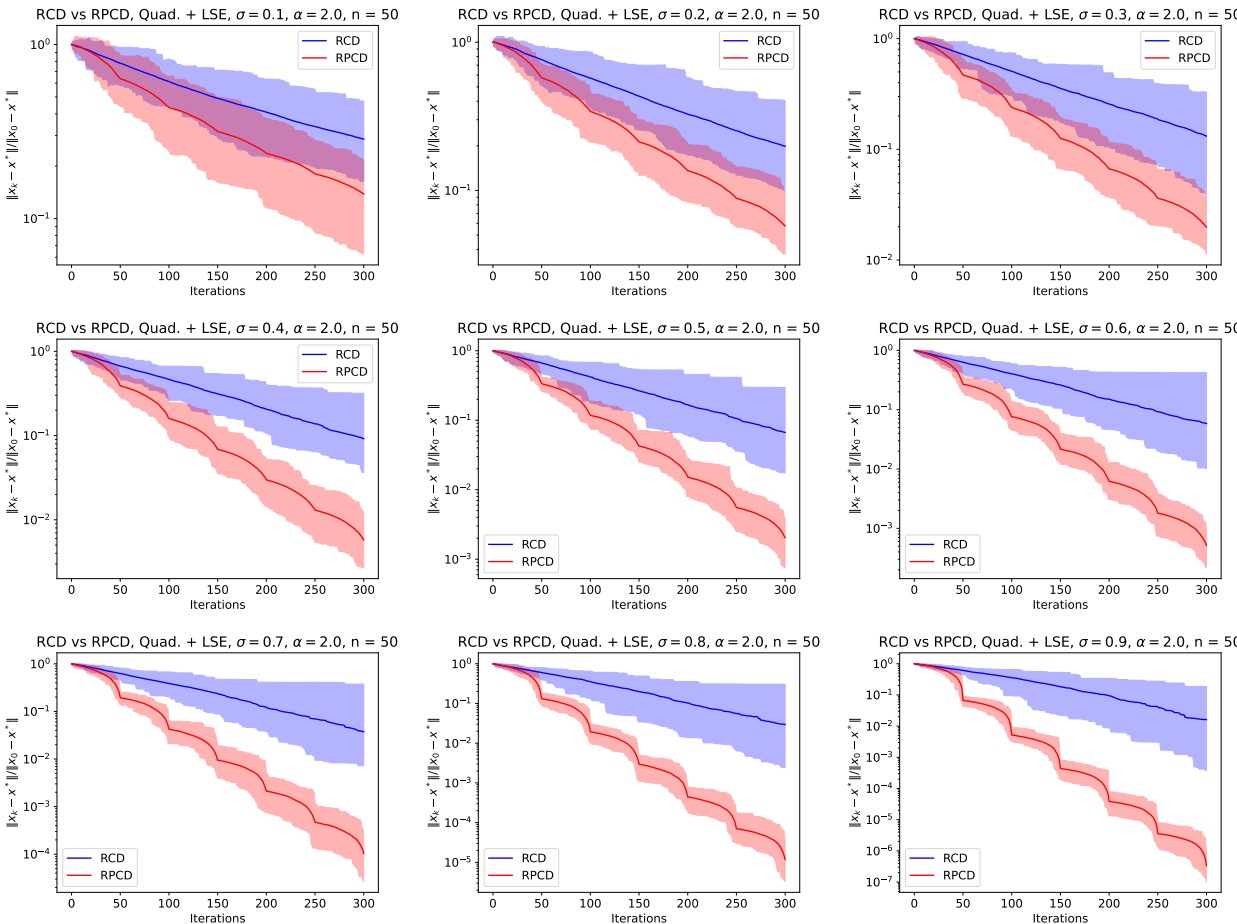

*Figure 12.* RCD vs RPCD, quadratic functions with unit-diagonal Hessians and log-sum-exp term. $\boldsymbol{A}$ is unit-diagonal and $\lambda_{\min}(\boldsymbol{A}) = \sigma$, $n = 50, \sigma \in \{0.1, \ldots, 0.9\}, \alpha = 2.0$. The $y$-axis is in log scale.

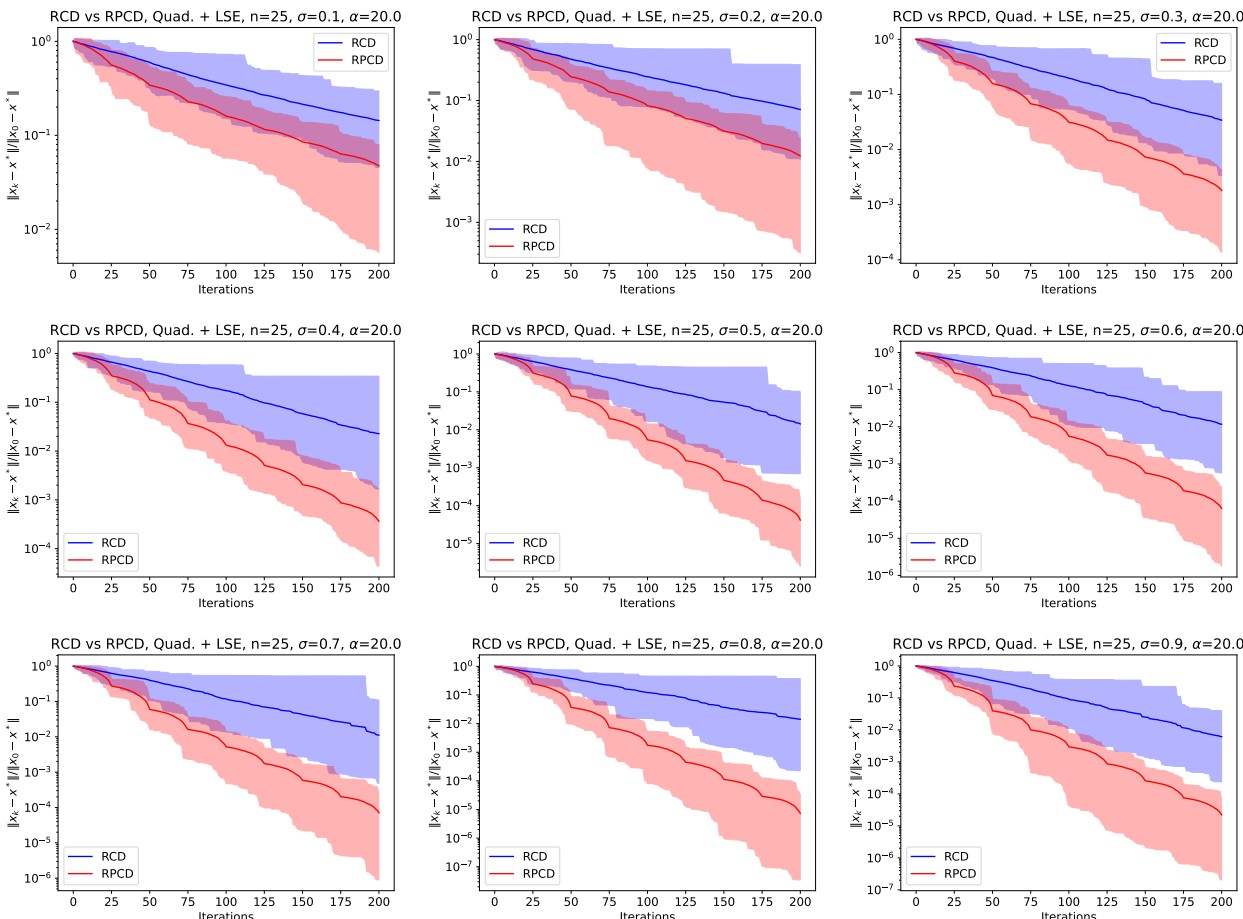

*Figure 13.* RCD vs RPCD, quadratic functions with unit-diagonal Hessians and log-sum-exp term. $\boldsymbol{A}$ is unit-diagonal and $\lambda_{\min}(\boldsymbol{A}) = \sigma$, $n = 25, \sigma \in \{0.1, \ldots, 0.9\}, \alpha = 20.0$. The $y$-axis is in log scale.

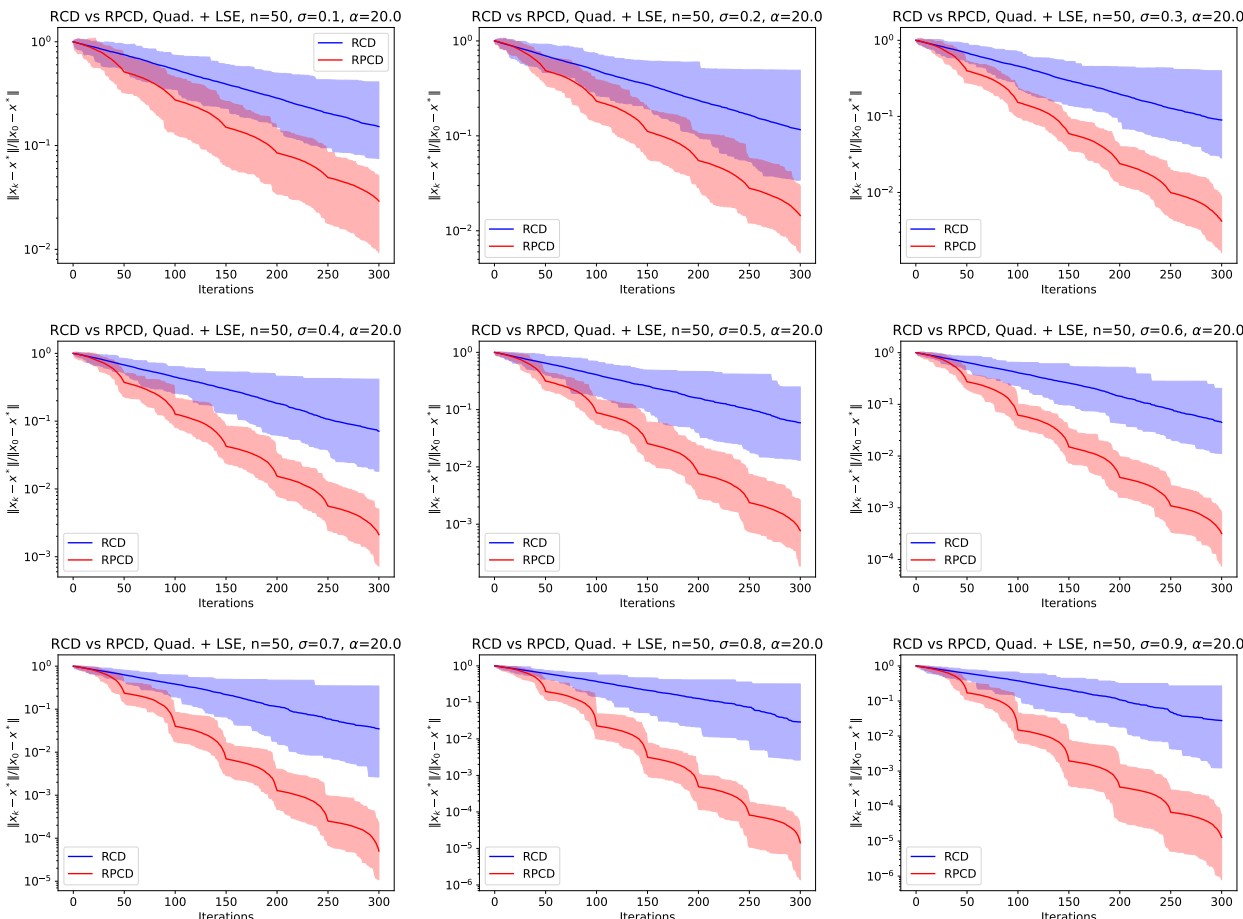

*Figure 14.* RCD vs RPCD, quadratic functions with unit-diagonal Hessians and log-sum-exp term. $\boldsymbol{A}$ is unit-diagonal and $\lambda_{\min}(\boldsymbol{A}) = \sigma$, $n = 50, \sigma \in \{0.1, \ldots, 0.9\}, \alpha = 20.0$. The $y$-axis is in log scale.

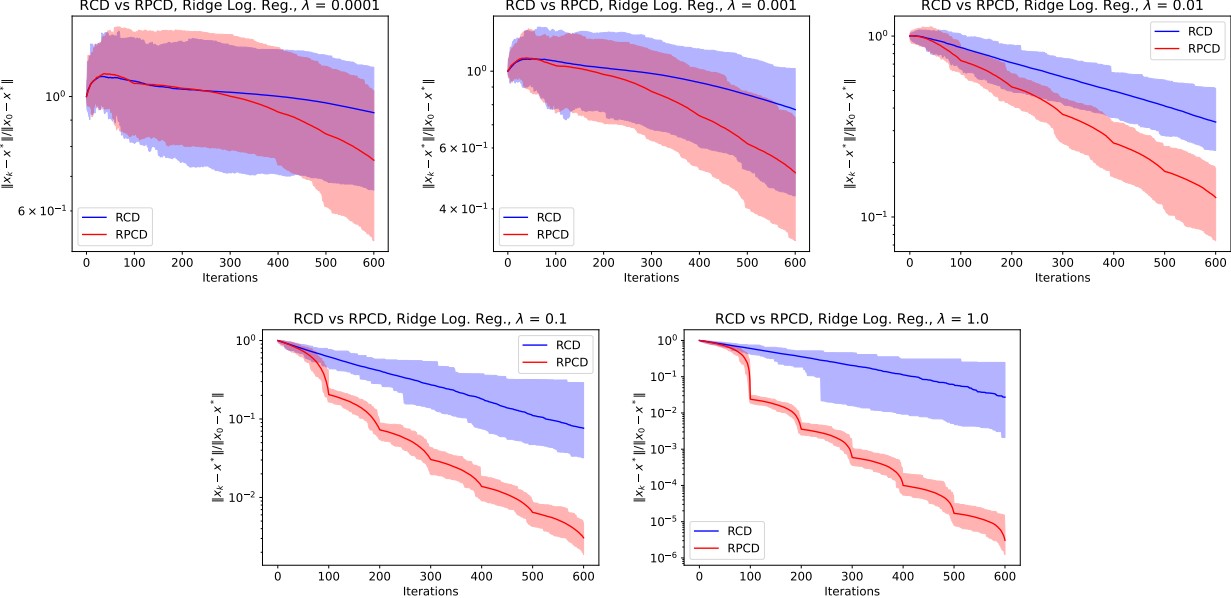

*Figure 15.* RCD vs RPCD, Ridge-regularized logistic regression, $n = m = 100$, $\lambda \in \{0.0001, 0.001, 0.01, 0.1, 1.0\}$. The $y$-axis is in log scale.

# G. Non-Asymptotic Comparison of RCD and RPCD

This section shows that RPCD outperforms RCD even after a finite number of epochs when $A \in \mathcal{A}_\sigma$, complementing the asymptotic results in the main text.

To avoid confusion, we introduce separate notations for RCD and RPCD. We denote by $x_T^{\text{RCD}}$ and $x_K^{\text{RPCD}}$ the iterates after $T$ iterations and $K$ epochs, respectively, both starting from the same initial point $x_0$. The corresponding iteration matrices, denoted by $M_A^{\text{RCD}}$ and $M_A^{\text{RPCD}}$, are both $2 \times 2$ iteration matrices of RCD and RPCD that represent the restriction of $\mathcal{M}_A^{\text{RCD}}$ and $\mathcal{M}_A^{\text{RPCD}}$ to $\text{span}\{I, 11^\top\}$.

## G.1. RCD Lower Bound

From the proof of Theorem 3.4, we have

$$\lambda_{\min}((\mathcal{M}_A^{\text{RCD}})^T(I)) = \alpha_T,$$

where $\alpha_T = e_1^\top (M_A^{\text{RCD}})^T e_1$. Let $v_1$ and $v_2$ be the unit eigenvectors of $M_A^{\text{RCD}}$ associated with eigenvalues $\lambda_1$ and $\lambda_2$, where $\lambda_1 \geq \lambda_2$. Then we can decompose $e_1 = c_1 v_1 + c_2 v_2$, where $c_1^2 + c_2^2 = 1$ since $v_1$ and $v_2$ form an orthonormal basis.

Using this decomposition, we compute

$$
\begin{aligned}
\alpha_T &= e_1^\top (M_A^{\text{RCD}})^T e_1 \\
&= (c_1 v_1 + c_2 v_2)^\top (M_A^{\text{RCD}})^T (c_1 v_1 + c_2 v_2) \\
&= c_1^2 \lambda_1^T + c_2^2 \lambda_2^T \\
&\geq c_1^2 \lambda_1^T.
\end{aligned}
$$

The inequality follows from the fact that $\lambda_2 \geq 0$, as shown in the proof of Theorem 3.4.

We now consider two cases: **(i)** $n > 2$ and **(ii)** $n = 2$.

**(i)** $n > 2$. When $n > 2$, all entries of $M_A^{\text{RCD}}$ are positive. Therefore, we can apply the following lemma, known as the Perron-Frobenius Theorem.

**Lemma G.1** (Horn & Johnson (2012), Theorem 8.2.8.). *Let $A \in \mathbb{R}^{n \times n}$ be a positive matrix, i.e., all entries are strictly positive. Then there exists a unique real vector $x$ such that $Ax = \rho(A)x$ and $x$ is a positive vector, i.e., all entries are strictly positive.*

By this lemma, the eigenvector $v_1 = \begin{bmatrix} v_{11} & v_{12} \end{bmatrix}^\top$ corresponding to the largest eigenvalue $\lambda_{\max}(M_A^{\text{RCD}})$ can be chosen to have strictly positive components.

Now, from the equation $M_A^{\text{RCD}} v_1 = \lambda_1 v_1$, comparing the second coordinates gives

$$\frac{v_{11}}{v_{12}} = \frac{n\lambda_1 - \sigma^2(n-2)}{\sigma^2}.$$

Hence, $v_{11} \geq v_{12}$ is equivalent to $\lambda_1 \geq \left(1 - \frac{1}{n}\right)\sigma^2$.

Now let $p$ be the characteristic polynomial of $M_A^{\text{RCD}}$. One can compute that

$$p\left(\left(1 - \frac{1}{n}\right)\sigma^2\right) = \frac{2(n-1)\sigma^2(\sigma - 1)}{n^2} \leq 0.$$

Since $p$ is a convex quadratic polynomial, the inequality above implies that $\lambda_1 \geq \left(1 - \frac{1}{n}\right)\sigma^2$. Therefore, $v_{11} \geq v_{12}$, and since $c_1 = e_1^\top v_1 = v_{11}$ and $v_{11}^2 + v_{12}^2 = 1$, we conclude that $c_1^2 \geq \frac{1}{2}$.

**(ii)** $n = 2$. In this case,

$$M_A^{\text{RCD}} = \frac{1}{2}\begin{bmatrix} 1 + (1-\sigma)^2 & 0 \\ \sigma^2 & 0 \end{bmatrix},$$

whose eigenvalues are 0 and $\frac{1+(1-\sigma)^2}{2}$. The eigenvector corresponding to the largest eigenvalue is proportional to

$$\boldsymbol{w} = \begin{bmatrix} 1 + (1-\sigma)^2 \\ 1 \end{bmatrix},$$

so the normalized eigenvector is $\boldsymbol{v}_1 = \frac{\boldsymbol{w}}{\|\boldsymbol{w}\|}$. Since $1 + (1-\sigma)^2 \geq \sigma^2$, it follows that $v_{11} \geq v_{12}$, and as in the previous case, we again obtain $c_1^2 \geq \frac{1}{2}$.

Finally, we have

$$\frac{\mathbb{E}[\|\boldsymbol{x}_T^{\text{RCD}}\|^2]}{\|\boldsymbol{x}_0\|^2} \geq \frac{1}{2}\left(1 - \frac{1}{n} + \frac{(1-\sigma)^2}{n}\right)^T.$$

## G.2. RPCD Upper Bound

From the proof of Theorem 3.3, we have

$$\begin{aligned}
\frac{\mathbb{E}\left[\|\boldsymbol{x}_K^{\text{RPCD}}\|^2\right]}{\|\boldsymbol{x}_0\|^2} &= \frac{\boldsymbol{x}_0^\top (\mathcal{M}_A^{\text{RPCD}})^K (\boldsymbol{I}) \boldsymbol{x}_0}{\boldsymbol{x}_0^\top \boldsymbol{x}_0} \\
&\leq \lambda_{\max}((\mathcal{M}_A^{\text{RPCD}})^K(\boldsymbol{I})) \\
&= \boldsymbol{y}^\top (\boldsymbol{M}_A^{\text{RPCD}})^K \boldsymbol{x},
\end{aligned}$$

where $\boldsymbol{x} = \begin{bmatrix} 1 & 0 \end{bmatrix}^\top$ and $\boldsymbol{y} = \begin{bmatrix} 1 & n \end{bmatrix}^\top$. Let $\boldsymbol{v}_1$ and $\boldsymbol{v}_2$ be the unit eigenvectors of $\boldsymbol{M}_A^{\text{RPCD}}$ such that $\|\boldsymbol{v}_i\| = 1$, associated with eigenvalues $\lambda_1, \lambda_2$ with $\lambda_1 \geq \lambda_2$. Then $\boldsymbol{x} = c_1 \boldsymbol{v}_1 + c_2 \boldsymbol{v}_2$ for some $c_1, c_2$ with $c_1^2 + c_2^2 = 1$, because $\boldsymbol{x} = \boldsymbol{e}_1$. We have

$$\begin{aligned}
\boldsymbol{y}^\top (\boldsymbol{M}_A^{\text{RPCD}})^K \boldsymbol{x} &= c_1 \lambda_1^K \boldsymbol{y}^\top \boldsymbol{v}_1 + c_2 \lambda_2^K \boldsymbol{y}^\top \boldsymbol{v}_2 \\
&= \lambda_1^K \left( c_1 \boldsymbol{y}^\top \boldsymbol{v}_1 + c_2 \left(\frac{\lambda_2}{\lambda_1}\right)^K \boldsymbol{y}^\top \boldsymbol{v}_2 \right) \\
&\leq \lambda_1^K \left( |c_1| \|\boldsymbol{y}\| \|\boldsymbol{v}_1\| + |c_2| \|\boldsymbol{y}\| \|\boldsymbol{v}_2\| \left(\frac{\lambda_2}{\lambda_1}\right)^K \right) \\
&\leq \lambda_1^K \|\boldsymbol{y}\| (|c_1| + |c_2|).
\end{aligned}$$

Since $c_1^2 + c_2^2 = 1$, we have $|c_1| + |c_2| \leq \sqrt{2}$. Also, $\|\boldsymbol{y}\| = \sqrt{n^2 + 1}$, so we get $\boldsymbol{y}^\top (\boldsymbol{M}_A^{\text{RPCD}})^K \boldsymbol{x} \leq \lambda_1^K \sqrt{2(n^2 + 1)}$.

## G.3. Comparison of RCD and RPCD

From the above, if $T = nK$, we have

$$\left( \frac{\mathbb{E}\left[\|\boldsymbol{x}_T^{\text{RCD}}\|^2\right]}{\|\boldsymbol{x}_0\|^2} \right)^{\frac{1}{K}} \geq \left(1 - \frac{1}{n} + \frac{(1-\sigma)^2}{n}\right)^n \left(\frac{1}{2}\right)^{\frac{1}{K}}$$

$$\left( \frac{\mathbb{E}\left[\|\boldsymbol{x}_K^{\text{RPCD}}\|^2\right]}{\|\boldsymbol{x}_0\|^2} \right)^{\frac{1}{K}} \leq \max\left\{ \left(1 - \frac{1}{n}\right)^n, \left(1 - \frac{\sigma}{n}\right)^{2n} \right\} \left(\sqrt{2(n^2+1)}\right)^{\frac{1}{K}}.$$

Thus, if we can show that

$$\left(1 - \frac{1}{n} + \frac{(1-\sigma)^2}{n}\right)^n \left(\frac{1}{2}\right)^{\frac{1}{K}} \geq \max\left\{ \left(1 - \frac{1}{n}\right)^n, \left(1 - \frac{\sigma}{n}\right)^{2n} \right\} \left(\sqrt{2(n^2+1)}\right)^{\frac{1}{K}}, \tag{42}$$

then we can conclude that RPCD is faster than RCD. The following is equivalent to (42):

$$\frac{\left(1 - \frac{1}{n} + \frac{(1-\sigma)^2}{n}\right)^n}{\max\left\{ \left(1 - \frac{1}{n}\right)^n, \left(1 - \frac{\sigma}{n}\right)^{2n} \right\}} \geq \left(2\sqrt{2(n^2+1)}\right)^{\frac{1}{K}}.$$

Since the left-hand side is $\geq 1$, for any $\sigma \in (0, 1)$, there exists $K_0$ such that if $K \geq K_0$, then (42) holds.

Therefore, after $K_0$ epochs, RPCD is faster than RCD.

