# OpenReview forum: "Provable Benefit of Random Permutations over Uniform Sampling in Stochastic Coordinate Descent"
_ICML.cc/2025/Conference — ICML 2025 poster_

### Official Review · Reviewer_Mbvp · 2025-03-06

**Overall Recommendation:** 4

**Summary:**

The paper considers coordinate descent with updates performed on iid coordinates (RCD) versus coordinates chosen by random permutation (RPCD).  The permutation updates are empirically known to outperform the iid baseline, but this has lacked theoretical justification.  The main result of the paper proves the RPCD method converges faster than RCD for a class of quadratic objectives with constraints on the quadratic form.

**Claims And Evidence:**

The claims are well supported by simple experiments measuring convergence and solid proofs.

**Essential References Not Discussed:**

N/A

**Experimental Designs Or Analyses:**

Yes

**Methods And Evaluation Criteria:**

The evaluations generally make sense for the setting.  One area somewhat under explained is section 4.1 in particular the algorithm to search for ‘difficult’ PSD matrices and section 4.2 where the log-sum-exp term is included for some of the experiments.  Both sections would benefit from a bit more explanation; why is that algorithm a better candidate for generating hard quadratic programs than any other matrix distribution?  When the authors write that the log-sum-exp terms make the loss less coordinate-friendly, is there a reference or quick explanation they could give for why this should be especially difficult for these algorithms?  Intuitively I can look at this function as very non-separable and guess that coordinate descent would have slower convergence, but the plots in Table 3 with or without the LSE terms appear almost identical.  I should mention I quite like this section for trying to give empirical ways to stress test the authors' hypothesis.

**Other Comments Or Suggestions:**

N/A

**Other Strengths And Weaknesses:**

N/A

**Questions For Authors:**

N/A

**Relation To Broader Scientific Literature:**

The paper seems to improve on earlier bounds at the expense of stricter assumptions.  I can appreciate why the problem is difficult as analyzing RPCD requires evaluating n gradient updates at a time.  The trick of (essentially) constraining the class of Hessians to the two-dimensional space of matrices invariant to conjugation by permutation makes sense, although it is quite limiting for the problem.

**Theoretical Claims:**

The results appear correct, I mainly confirmed the analysis in the appendix up to page 19, although I didn’t go through all the details of the very technical followup chasing polynomial root identities.

---

> ### Author Rebuttal · Authors · 2025-04-01
>
> We appreciate the valuable questions and comments, and we express our gratitude for the positive feedback and recognizing the novelty of our work. We provide detailed responses to the questions below.
>
> ### Methods And Evaluation Criteria
>
> > *The evaluations generally make sense for the setting. One area somewhat under explained is section 4.1 in particular the algorithm to search for ‘difficult’ PSD matrices and section 4.2 where the log-sum-exp term is included for some of the experiments. Both sections would benefit from a bit more explanation; why is that algorithm a better candidate for generating hard quadratic programs than any other matrix distribution? When the authors write that the log-sum-exp terms make the loss less coordinate-friendly, is there a reference or quick explanation they could give for why this should be especially difficult for these algorithms? Intuitively I can look at this function as very non-separable and guess that coordinate descent would have slower convergence, but the plots in Table 3 with or without the LSE terms appear almost identical. I should mention I quite like this section for trying to give empirical ways to stress test the authors' hypothesis.*
>
> - To elaborate on the **Algorithmic Search** part in Section 4.1, this method works by using a scipy optimizer (as in Step 4) to maximize the value $\rho(\mathcal{M}\_{\boldsymbol{A}})$, while Steps 1–3 ensure that the matrix $\boldsymbol{A}$ is generated to be inside the desired function class (unit-diagonal PSD with fixed $\lambda_{\min} = \sigma$). The worst case examples are likely to be in a (Lebesgue) measure zero set, which motivated us to use such an optimization approach rather than sampling-based methods. We will add a more detailed explanation in our next revision.
> - For the quadratic+LSE experiments in Section 4.2, we would like to clarify that our purpose of using the LSE term was to simply construct a non-quadratic example; it is not specially designed as a hard instance for CD algorithms, while we did add the orthogonal matrix $\boldsymbol{Q}$ to see if random permutations are beneficial for the 'non-coordinatewise-separable' cases as well.
> - We also conducted experiments expanding the range of $\alpha$ up to 100 in the quadratic+LSE setting. Across all experiments, RPCD consistently converged faster than RCD. However, convergence slowed with increasing $\alpha$. However, the performance gap between RPCD and RCD narrowed as $\alpha$ increased. (You may also refer to our response to reviewer gqcX.)
>
> We provide a link to the experimental results: https://anonymous.4open.science/r/rcd_rpcd-7416/
>
> ### Relation To Broader Scientific Literature:
>
> > *The paper seems to improve on earlier bounds at the expense of stricter assumptions.*
>
> While we thank the reviewer again for the diligent review and positive response, we would like to carefully point out that our assumptions are not necessarily *stricter* than directly relevant previous works that consider *permutation-based* coordinate descent algorithms.
> Specifically, [LW19] considers a narrower class of problems than we do (as noted in Section 3.3, Comparison with Previous Work). While [WL17] considers a slightly larger class of quadratics allowing small diagonal perturbations on permutation-invariant Hessians, [GOVW18] also relies on the permutation-invariant Hessian assumption, although they considered negative off-diagonal cases.
>
> [LW19] Ching-Pei Lee and Stephen J. Wright. Random permutations fix a worst case for cyclic coordinate descent. IMA Journal of Numerical Analysis, 2019.
>
> [WL17] Stephen J. Wright and Ching-Pei Lee. Analyzing random permutations for cyclic coordinate descent. Mathematics of Computation, 2017.
>
> [GOVW18] Mert Gurbuzbalaban, Asuman Ozdaglar, Nuri Denizcan Vanli, Stephen J. Wright. Randomness and permutations in coordinate descent methods. Mathematical Programming, 2018.

---

> > ### Comment · Reviewer_Mbvp · 2025-04-01
> >
> > Thank you to the authors for answering my questions.  I will keep my positive score.

---

### Official Review · Reviewer_h7Qf · 2025-03-09

**Overall Recommendation:** 2

**Summary:**

Authors reiterate the problem of proving that the coordinate descent method with a random permutation of coordinates (RPCD) is theoretically faster than classical random coordinate descent (RCD). They prove that the asymptotic lower bound on the convergence rate of RCD is worse than the upper bound on the convergence rate of RPCD. Furthermore, they show a strict gap between the convergence rates of RCD and RPCD on the class of quadratic functions with permutation-invariant Hessians.

**Claims And Evidence:**

The claims seem well-supported by the provided theory, but I have not independently checked the proofs.

**Essential References Not Discussed:**

This paper (Xu & Yin, 2015, https://epubs.siam.org/doi/epdf/10.1137/140983938) analyzes an RPCD-like algorithm, which performs descent over a subspace rather than exact minimization within it. The study establishes a \(1/\sqrt{k}\) convergence rate in the convex setting, assuming standard conditions.

**Experimental Designs Or Analyses:**

The experiments seem appropriate for a work with a predominantly theoretical focus.

**Methods And Evaluation Criteria:**

The proposed class of functions appears adequate and is somewhat accepted in other works (Lee & Wright, Random permutations fix a worst case for cyclic coordinate descent)

**Other Comments Or Suggestions:**

-

**Other Strengths And Weaknesses:**

-

**Questions For Authors:**

The paper conjectures that the benefits of RPCD extend beyond quadratic objectives. Have you considered applying your analysis framework to more general convex functions, such as strongly convex but non-quadratic functions?

**Relation To Broader Scientific Literature:**

Other studies analyze the Lipschitz smoothness of the function as the key parameter driving RCD-related methods (https://arxiv.org/pdf/1805.09185, https://arxiv.org/pdf/2102.07245). However, the current paper is more directly comparable to Lee & Wright, 2019 (https://optimization-online.org/wp-content/uploads/2016/07/5562.pdf). As the authors themselves note, their convergence results for RPCD (Equation 10 in the paper) are weaker than those of Lee & Wright (Equation 3.15 in the linked work). In the appendix, the authors claim to improve the convergence rate and extend its applicability to a broader range of values for $\sigma$.

**Theoretical Claims:**

I have not verified the correctness of the provided claims.

---

> ### Author Rebuttal · Authors · 2025-04-01
>
> We are grateful for the reviewer's valuable questions and rich comments. Below, we have put together our responses to the questions.
>
> ### Relation To Broader Scientific Literature
>
> > *As the authors themselves note, their convergence results for RPCD (Equation 10 in the paper) are weaker than those of Lee & Wright (Equation 3.15 in the linked work). In the appendix, the authors claim to improve the convergence rate and extend its applicability to a broader range of values for $\sigma$.*
>
> We would like to clarify that our RPCD upper bound results in the main paper (Theorem 3.3) are presented in a slightly weaker but more concise form solely for the purpose of readbility and easier comparison with the RCD lower bounds, and we have demonstrated in Appendix D that our analysis is not fundamentally weaker than that of previous works.
>
> ### Essential References Not Discussed
>
> Xu & Yin (2015) analyze an RPCD-like algorithm by fixing the update order without loss of generality. This approach does not fully address the distinction between a randomly permuted coordinate selection and a deterministic, fixed-order update. Nevertheless, thanks for the suggestion and we will consider adding discussions in our next revision.
>
> ### Questions
>
> > *The paper conjectures that the benefits of RPCD extend beyond quadratic objectives. Have you considered applying your analysis framework to more general convex functions, such as strongly convex but non-quadratic functions?*
>
> This is a very interesting question. It is true that our framework (or more precisely, the proof techniques) cannot be directly applied to broader function classes, as it all heavily relies on the idea of using matrix operators that act on the 'Hessian-like matrices' of the quadratics.
>
> There are two possible directions to generalize to general non-quadratic objectives:
> * One would be to derive local convergence results for non-quadratics, which we discuss in our response to Reviewer gqcX.
> * Another direction could be to use existing results for other permutation-based algorithms like SGD with random reshuffling (SGD-RR) [MKR20] by basically substituting the finite-sum gradient oracles with the coordinate blocks of the gradients. Unlike SGD-RR, however, CD-type algorithms already enjoy *linear convergence* (even without random permutation) if the function is $\mu$-strongly convex. (This is also related to the analysis of SGD-RR under the interpolation condition by [FTS23], where plugging in $\sigma = 0$ yields linear convergence.) Thus, we cannot benefit from the variance reduction analysis in previous work, and it turns out that directly applying variance reduction analysis from SGD-RR leads to a loose upper bound. It is more likely that the effect of using permutations will be closer to a *preconditioning-like* effect (similar to previous analyses on cyclic CD [SY16]), which is largely different from SGD-RR, and we leave this direction for future investigation.
>
> [MKR20] Konstantin Mishchenko, Ahmed Khaled, Peter Richtárik. Random Reshuffling: Simple Analysis with Vast Improvements. NeurIPS 2020.
>
> [FTS23] Chen Fan, Christos Thrampoulidis, Mark Schmidt. Fast Convergence of Random Reshuffling under Over-Parameterization and the Polyak-Lojasiewicz Condition. ECML PKDD 2023.
>
> [SY16] Ruoyu Sun, Yinyu Ye. Worst-case Complexity of Cyclic Coordinate Descent: $O(n^2)$ Gap with Randomized Version. Mathematical Programming, 2016.

---

### Official Review · Reviewer_M9VV · 2025-03-12

**Overall Recommendation:** 4

**Summary:**

The paper studies the stochastic coordinate descent method for quadratic optimization and focuses on two schemes: uniform sampling versus random permutation. Under the mild unit-diagonal assumption, the authors show that random-permutation coordinate descent (RPCD) converges faster than random coordinate descent (RCD).  Precisely, the authors establish the following results.

1. The authors prove a lower bound for RCD.

2. For a special function class, RPCD is proven to be faster than the lower bound of RCD mentioned above.

3. For the same special function class, RCD admits an even stronger (slower) lower bound.

The authors also discuss how to generalize their results and provide several numerical experiments.

## update after rebuttal

I maintain my score.

**Claims And Evidence:**

All claimed theorems are proved.

**Essential References Not Discussed:**

N/A.

**Experimental Designs Or Analyses:**

The experiments are sufficient and details are also reported.

**Methods And Evaluation Criteria:**

N/A.

**Other Comments Or Suggestions:**

1. I think it's better to clarify that $\mathcal{M}_{\boldsymbol{A}}^{k}$ denotes the function composition in stead of its $k$-th power.

2. For some places, the identity matrix is written as $\boldsymbol{I}$ instead of $\boldsymbol{I}_{n}$ as defined at the beginning of Section 2.

3. It's better to say Theorems 3.1 and 3.4 hold with respect to the Lebesgue measure.

4. It could be better to provide the explicit expression of $\mathcal{M}_{\boldsymbol{A}}^{\mathrm{RCD}}(\cdot)$ since it is computable and will be used many times in the proof.

**Other Strengths And Weaknesses:**

The paper is well-written in general. I can't find any major weaknesses.

**Questions For Authors:**

The paper focuses on asymptotic results. Could the authors say something about non-asymptotic analysis?

**Relation To Broader Scientific Literature:**

The analysis may be useful in more general optimization problems.

**Theoretical Claims:**

I mainly viewed proofs of Theorems 3.1 and 3.4. As far as I can check, they are correct.

---

> ### Author Rebuttal · Authors · 2025-04-01
>
> We deeply appreciate the constructive review and feedback. We thank the reviewer for reading the paper thoroughly in detail, and we hope our response relevantly addresses all points raised in the review.
>
> ### Appendix
>
> > *Line 622: $k\to \infty$ should be $T\to \infty$.*
>
> Thanks for finding the typo. We will fix this in our next revision.
>
> > *Line 720: I can't see the reason why $\mu_{i}<\mu_{1}$ (they possibly equal to each other).*
>
> It is true that we must consider cases when $\mu_i = \mu_1$ and yet the limit remains unchanged. We will fix the details in our next revision as well.
>
> ### Comments
>
> > *I think it's better to clarify that $\mathcal{M}\_{\boldsymbol{A}}^{k}$ denotes the function composition instead of its k-th power. For some places, the identity matrix is written as $I$ instead of $I_{n}$ as defined at the beginning of Section 2.*
>
> We will elaborate on these points in Section 2 in our next revision.
>
> > *It's better to say Theorems 3.1 and 3.4 hold with respect to the Lebesgue measure.*
>
> It is true that the ‘measure zero sets’ are with respect to the Lebesgue measure. We will also add this in our next revision.
>
> > *It could be better to provide the explicit expression of $\mathcal{M}_{\boldsymbol{A}}^{\text{RCD}}$ since it is computable and will be used many times in the proof.*
>
> We deferred the explicit form of $\mathcal{M}_{\boldsymbol{A}}^{\text{RCD}}$ to Line 660 of the Appendix, but we appreciate the feedback and will consider moving this into the main text in our next revision.
>
>
> ### Questions
>
> > *The paper focuses on asymptotic results. Could the authors say something about non-asymptotic analysis?*
>
> This is a very good question. The reason why our results are asymptotic is that they involve matrix powers. Specifically, see Line 705 (Theorem 3.1), Line 943 (Theorem 3.3), and Line 2081 (Theorem 3.4). However, with a more refined analysis, we can determine the minimum number of iterations to guarantee RPCD's faster convergence compared to RCD. In order to obtain non-asymptotic convergence guarantees for some large enough yet finite $K$ or $T$, we must find the non-asymptotic counterparts of parts like Lines 708-714 and Lines 941-948 in our proofs. However, this is not so straightforward as, for instance, the quantity in Lines 716-719 reaches $1$ at the limit but is smaller than 1 for finite $T$. In fact, our RCD lower bound increases as $T \rightarrow \infty$, while our RPCD upper bound decreases as $K \rightarrow \infty$, which necessitates characterizing the "crossing" point when the RPCD upper bound becomes smaller than RCD lower bound after a certain number of iterations; this makes the non-asymptotic analysis harder.
>
> We plan to add these discussions as a remark in our next revision.

---

### Official Review · Reviewer_gqcX · 2025-03-13

**Overall Recommendation:** 4

**Summary:**

This paper investigates the convergence rates of random coordinate descent (RCD) and random permutation coordinate descent (RPCD) for minimizing a class of quadratic functions. The key contributions are: (a) a novel lower bound for RCD's contraction rate on general positive definite quadratic functions and a stronger version on a specific function class (denoted by $\mathcal{A}\_{\sigma}$); (b) an upper bound for RPCD's contraction rate on $\mathcal{A}\_{\sigma}$; (c) showing that the upper bound of RPCD is strictly smaller than the lower bound of RCD on $\mathcal{A}\_{\sigma}$.

## Update after rebuttal

I am satisfied with the further clarifications in the rebuttal, and I decide to keep my score.

**Claims And Evidence:**

Yes, the theoretical claims are supported by proofs and numerical experiments.

**Essential References Not Discussed:**

None.

**Experimental Designs Or Analyses:**

The numerical experiments are mainly usesd to support the theoretical claims and seem to be valid.

**Methods And Evaluation Criteria:**

Yes.

**Other Comments Or Suggestions:**

Here are a few additional comments aiming at helping the author(s) further improve the article:
1. Since the whole analysis focuses on quadratic functions, it may not be very meaningful to introduce the general optimization problem in equation (1). Clearly setting the scope of this paper helps the readers better comprehend the main contributions.
2. Since different theorems are presented for different function classes, I suggest the author(s) making a table that clearly classifies the problems and summarizes the known results for each class. This may help the readers understand which part of the theory is proved and which is unknown.

**Other Strengths And Weaknesses:**

I think this paper is a solid contribution to the understanding of coordinate-descent-type algorithms, and the theoretical results and their implications are well presented.

The main weakness is that the results only hold on a restricted function class, and it is likely that the technical tool developed in this paper may be difficult to generalize to other problems.

**Questions For Authors:**

I have one question mainly on the design of the numerical experiments. Is there any specific reason why you consider the quadratic+LSE problem in Section 4.2? Since the setting (iii) is no longer within the scope of the theory, it may be more interesting to investigate problems that are more commonly seen in machine learning tasks. Also, is there any $\alpha$ that breaks the superiority of RPCD? This may provide some insights on how the benefit of RPCD relies on the quadratic form of the objective function.

**Relation To Broader Scientific Literature:**

Although the results are restricted to a specific function class, the analysis in this paper provides insights on the benefit of RPCD over RCD on a broader class of problems.

**Theoretical Claims:**

I have not checked the full technical details, but the overall correctness seems to be sound.

---

> ### Author Rebuttal · Authors · 2025-04-01
>
> We thank the reviewer for the positive response and meaningful questions. Below, we summarize and respond to your questions one by one.
>
> ### Weaknesses
>
> > *The main weakness is that the results only hold on a restricted function class, and it is likely that the technical tool developed in this paper may be difficult to generalize to other problems.*
>
> While the reviewer’s comment is correct overall, we would like to carefully add that we have empirical evidence that the function class $\mathcal{A}_{\sigma}$ represents the worst case among general quadratics and that there still exist potential ways to extend the proof techniques to a slightly larger subclass of quadratics (see **Appendix E**). As a side note, for deterministic algorithms, it is possible that convergence results on quadratics could also yield *local* convergence rates of non-quadratic functions by considering the Hessian at the optimum (see, e.g., [Bertsekas, 1997, Proposition 4.4.1]). While this motivates us to extend our results to non-quadratic objectives and demonstrate local convergence, directly applying this approach is likely challenging in the stochastic setting. Identifying a suitable operator for analyzing the expected iterate norm is a technical hurdle, and we therefore leave this extension to future work. (We also leave another discussion point regarding non-quadratic functions in our response to reviewer h7Qf.)
>
> [Bertsekas, 1997] Dimitri P Bertsekas. Nonlinear programming. Journal of the Operational Research Society, 48(3):334–334, 1997.
>
>
> ### Comments
>
> > *Since the whole analysis focuses on quadratic functions, it may not be very meaningful to introduce the general optimization problem in equation (1). Clearly setting the scope of this paper helps the readers better comprehend the main contributions.*
>
> Our purpose in starting with the general minimization problem (1) was for a smoother exposition of the introduction section, as some of the previous works we discuss therein consider general convex functions. Nevertheless, we appreciate the feedback and will try to clarify the scope of our results by adding more details in the **Summary of Contributions** paragraph.
>
> > *Since different theorems are presented for different function classes, I suggest the author(s) making a table that clearly classifies the problems and summarizes the known results for each class. This may help the readers understand which part of the theory is proved and which is unknown.*
>
> We tried to add a summary table of our results in the main paper, but had to remove it due to space limitations; we will add this table in our next revision.
>
>
> ### Questions
>
> > *Is there any specific reason why you consider the quadratic+LSE problem in Section 4.2? Since the setting (iii) is no longer within the scope of the theory, it may be more interesting to investigate problems that are more commonly seen in machine learning tasks. Also, is there any $\alpha$ that breaks the superiority of RPCD? This may provide some insights into how the benefit of RPCD relies on the quadratic form of the objective function.*
>
> For the quadratic+LSE experiments in Section 4.2, we would like to clarify that our purpose in using the LSE term was to simply construct a non-quadratic example. Considering the reviewer's question, we also conducted additional experiments on (1) expanding the range of $\alpha$ up to 100 in setting (iii) in our experiments, and (2) a logistic regression task with a ridge penalty, following the setup in [NSLFK15] (with n=m=100). The loss function for this experiment is as follows:
> $$ \min_x \frac{1}{m} \sum_{i=1}^{m} \log(1 + \exp(-b_i a_i^{\top} x)) + \frac{\lambda}{2} ||x||^2, $$ where $a_i$ and $y$ are drawn from the standard normal distribution and $b_i=\text{sign}(a_i^{\top} y)$ but randomly flipped the sign with probability 0.1. Across all experiments, we observed that RPCD consistently converged faster than RCD. Another thing to note is that convergence slowed with increasing $\alpha$ in (1) and decreasing ridge penalty in (2).
>
> We provide a link to the experimental results: https://anonymous.4open.science/r/rcd_rpcd-7416/
>
> [NSLFK15] Julie Nutini, Mark Schmidt, Issam H. Laradji, Michael Friedlander, Hoyt Koepke. Coordinate descent converges faster with the Gauss-Southwell rule than random selection. In Proceedings of the 32nd International Conference on Machine Learning, 2015.

---

> > ### Comment · Reviewer_gqcX · 2025-04-03
> >
> > Thanks for the response. I decide to keep my score.

---

### Decision · Program_Chairs · 2025-05-01

**Decision:**

Accept (poster)

**Comment:**

This paper studies the open question whether RPCD (randomly permuted coordinate descent) is faster than RCD (randomized coordinate descent). It shows that RPCD is faster than RCD for a class of quadratic optimization problems where the Hessians are permutation invariant. More specifically, it establishes rigorously the upper bound of RPCD for this class, and the lower bound of RCD for this class, and by comparing them it reaches the results that RPCD is faster than RCD for this class of problems. A limitation is that the results focus on a specific function class; nevertheless, given the slow progress on this problem over years, the results in this paper are encouraging and may shed light on this difficult and fundamental problem. I recommend accept.